# Nasal *Staphylococcus aureus* carriage promotes depressive behaviour in mice via sex hormone degradation

Guoxiu Xiang[1,6], Yanan Wang [1,6], Kaiji Ni[2,6], Huoqing Luo[3], Qian Liu [1], Yan Song[1], Ping Miao[1], Lei He [1], Ying Jian[1], Ziyu Yang[1], Tianchi Chen[1], Ke Xu[1], Xia Sun[2], Zhen Shen[1], Chenfeng Ji[2], Na Zhao[1], Mengxin He[2], Yan Pan[2], Yanli Luo [2]✉, Ji Hu [3]✉, Michael Otto [4]✉ & Min Li [1,5]✉

The human microbiome has a pronounced impact on human physiology and behaviour. Despite its unique anatomical connection to the brain, the role of the nasal microbiome in neurological diseases is understudied. Here, using human data and experiments in mice, we show that nasal *Staphylococcus aureus* is linked to depression. Nasal microbiome analyses revealed a positive correlation between depression scores and *S. aureus* abundance among patients with depression and healthy controls. Metabolomics of the nasal cavity showed decreased sex hormones, estradiol and testosterone in patients with depression versus controls. Nasal microbiota transplants from patients reproduced depression-like behaviour in mice with differential abundance of *S. aureus*. Further homology and mutational analysis uncovered an *S. aureus* sex hormone-degrading enzyme, 17b-hydroxysteroid dehydrogenase (Hsd12), which degraded testosterone and estradiol in mice, leading to lower levels of dopamine and serotonin in the murine brain. These findings reveal a nasal commensal that influences depressive behaviour and provides insights into the nose–brain axis.

Depression is a complex disorder whose pathogenesis is multifactorial[1]. It causes considerable personal suffering, an increased risk of suicide, cardiac disease and other comorbidities, as well as a substantial economic burden[2]. At the molecular level, depression is often accompanied by an abnormal abundance of cytokines, neurotrophic factors and sex hormones, leading to changes in neurotransmitter concentrations at the synapses[3]. Steroid sex hormones in particular serve as potent endogenous neuromodulators and have been shown to exert a substantial impact on the pathophysiology of depression, in large part through their modulation of dopamine and serotonin biosynthesis[4].

In recent years many human morbidities have been linked to a dysbiosis of the colonizing microbiota[5]. There is now also profound evidence that the gut microbiota influences brain function. This includes depression[6], which may happen via alteration of the levels of dietary neurotransmitter precursors[7] or degradation of sex hormones[8,9].

It remains poorly understood if bacterial colonization or infection of body sites other than the gut impacts brain function. Hosang and colleagues reported that the lung microbiome influences brain auto-immunity[10] and Wang and colleagues reported that *Porphyromonas gingivalis*, an oral pathogen, impacts depressive

[1]Department of Laboratory Medicine, Renji Hospital, School of Medicine, Shanghai Jiao Tong University, Shanghai, China. [2]Department of Psychological Medicine, Renji Hospital, School of Medicine, Shanghai Jiao Tong University, Shanghai, China. [3]School of Life Science and Technology, ShanghaiTech University, Shanghai, China. [4]Pathogen Molecular Genetics Section, Laboratory of Bacteriology, Division of Intramural Research, National Institute of Allergy and Infectious Diseases, US National Institutes of Health, Bethesda, MD, USA. [5]School of Nursing, Shanghai Jiao Tong University, Shanghai, China. [6]These authors contributed equally: Guoxiu Xiang, Yanan Wang, Kaiji Ni. ✉e-mail: luoluoyanli@163.com; huji@shanghaitech.edu.cn; motto@niaid.nih.gov; rjlimin@shsmu.edu.cn

behaviour in mice[11]. Whether and how the nasal microbiome impacts brain function has remained virtually unaddressed. This is surprising given that the close vicinity of the nasal system to the brain and the fact that the intranasal route allows direct entry of drugs and biologics to the central nervous system indicate that metabolites produced by nose-colonizing bacteria may reach and affect the brain[12,13].

In this study we analysed the nasal microbiome of people suffering from depression compared with that of healthy controls. *Staphylococcus aureus* emerged as the predominant bacterium associated with depression. We show that *S. aureus* colonization promotes depressive behaviour in mice via an enzyme that degrades sex hormones, leading to decreased levels of serotonin and dopamine in the brain. These findings establish a role of the nasal microbiome in the pathogenesis of depression and identify *S. aureus*, a nasal commensal in a considerable subset of the human population[14], as responsible for the underlying mechanism.

## Nasal *S. aureus* carriage is associated with depression

We analysed nasal microbiome samples from 100 patients with untreated depression and 118 healthy controls. Other than for measures of anxiety and depression, there were no significant demographic or clinical differences between the two groups, including the serum concentrations of hormones and cytokines known to potentially affect anxiety and depression[15] (Supplementary Table 1). The microbiome diversity (α-diversity) was higher in samples from healthy individuals than those with depression and the microbiota of the two groups were significantly[16] dissimilar (β-diversity; Fig. 1a,b), indicating a significant association of the nasal microbiome composition with depression. Following analysis using multivariate logistic regression models adjusting for several covariates including sex, age, body mass index, education, income, psychosocial and biochemical determinants, the α-diversity remained significantly different (Supplementary Table 2).

Using a method that allows species-level determination[17,18], we discovered that among the most abundant genera or species, only the genus *Staphylococcus* and specifically the species *S. aureus* were significantly more abundant in individuals with depression compared with healthy controls (Fig. 1c,d and Extended Data Fig. 1a,b). Culture-based analysis substantiated that assessment (Extended Data Fig. 1c,d). There was also a significant difference in absolute *S. aureus* abundance, both when males and females were analysed separately and in the combined groups (Fig. 1e). Moreover, Patient Health Questionnaire-9 (PHQ-9) and Generalized Anxiety Disorder-7 (GAD-7) scores were significantly[16] associated with *S. aureus* abundance in females and males (Fig. 1f and Extended Data Fig. 1e–g). These findings showed that *S. aureus* abundance in the nasal microbiome is positively associated with depression.

*S. aureus* colonizes the nose (as well as, in a largely correlated fashion, other body sites) only in about a quarter to a third of the human population[14]. We found that there was a significantly higher abundance of *S. aureus* nasal carriers among patients with depression than healthy individuals (Fig. 1g), suggesting that *S. aureus* carriage is a risk factor for depression.

Finally, sequence type analysis did not reveal that specific *S. aureus* strains are associated with depression, with sequence types 398 and 15 (ST398 and ST15, respectively) dominating in both groups, suggesting that any potential causal relationship is probably not strain-specific (Extended Data Fig. 1h).

## Nasal *S. aureus* induces depressive behaviour in mice

To analyse experimentally whether the nasal microbiome impacts depression, we transferred nasal microbiome samples from healthy individuals and patients with depression into the nasal cavities of mice (one donor to one mouse, randomly selected but always male to male and female to female). We then used established mouse models for the subsequent analysis of depressive behaviour (Fig. 2a). We first ascertained that the model set-up appropriately measured the impact of the transferred microbiomes without cofounders and the mice did not show signs of disease[19] (Extended Data Fig. 2a). Application of antibiotics to the nostrils resulted in a decrease of only approximately two orders of magnitude of colony-forming units (CFUs) and significant changes in the nasal, but not lung or gut microbiomes, indicating that any observed phenotypes are exclusively caused by nasal microbiome changes, as intended (Extended Data Fig. 2b–k). Following transplantation, the nasal microbiomes of the two groups mice (transplants from healthy individuals versus from patients with depression) differed significantly, reflecting the differences observed in humans (Extended Data Fig. 2l–n). Finally, the nasal microbiomes changed significantly after transplantation and recovered to bacterial abundance levels and a composition comparable to that before the experiment within a course of 14 days after the last transplantation (Extended Data Fig. 2b,o–q).

We observed significantly increased levels of anxiety-like behaviour in the open field test (OFT) and depression-like behaviour in forced swimming and tail suspension tests (FST and TST, respectively) in mice that received transplants from patients with depression (Fig. 2b and Extended Data Fig. 3a). Furthermore, the OFT, FST and TST values were significantly and at 'fair' levels[16] correlated with the levels of depression and anxiety of the transplant donors, as measured by PHQ-9 and GAD-7 clinical scales, respectively (Extended Data Fig. 3b). These findings strongly indicated that the composition of the nasal microbiome influences depressive behaviour. Notably, in a multivariate analysis, mice that received microbiota transplants from depressed patients showed significantly higher *S. aureus* relative abundance compared with mice that received microbiota transplants from healthy controls (Fig. 2c).

To provide direct experimental evidence for a causal relationship between *S. aureus* carriage and depression, we colonized the noses of mice with *S. aureus* and then measured anxiety and depressive behaviour separately in male and female mice (Fig. 2d–g). To demonstrate a specific role of *S. aureus*, we performed additional comparisons to mice that received the equal numbers of *Staphylococcus epidermidis*, a less virulent relative of *S. aureus* that also colonizes the nose[20,21]. The two bacterial species achieved similar colonization levels in our mouse model, allowing direct comparison (Extended Data Fig. 4a). No changes in lung and nose cytokine levels, histological appearance of lung tissue and nasal epithelia, abundance of olfactory macrophages and neutrophils or expression of the olfactory marker protein were observed between the control and *S. aureus* or *S. epidermidis* nasally colonized mice (Extended Data Fig. 4b–h). Even shortly (6 h) after the last application of the bacteria, there were no differences in cytokine expression of the lungs and only minor changes in the transcriptome 3 h after the last application of the nasal mucosa, which are expected and reflective of physiological immune activation associated with commensal colonization (Extended Data Fig. 5a–c). Moreover, no differences in weight or temperature were noted between the groups and no signs of disease were observed (Extended Data Fig. 5d–f). Together, these controls substantiated that our model is a model of asymptomatic nasal colonization. Notably, *S. aureus* colonized the nose at >1 × 10³ CFU over the entire course of the experiment and was undetectable in the lungs, whereas intestinal colonization[14] was transient, indicating that the phenotypes observed in this experiment are due to nasal colonization rather than colonization at other body sites (Extended Data Fig. 5g).

Female, but not male, mice colonized with *S. aureus* exhibited increased anxiety and depressive behaviour (Fig. 2e and Extended Data Fig. 6a,b). When male mice were additionally treated with a protocol of chronic unpredictable mild stress (CUMS; Fig. 2f), the results were similar to those observed in female mice (Fig. 2g). These findings indicate that *S. aureus* nasal carriage promotes depression and does so more strongly in female than male mice.

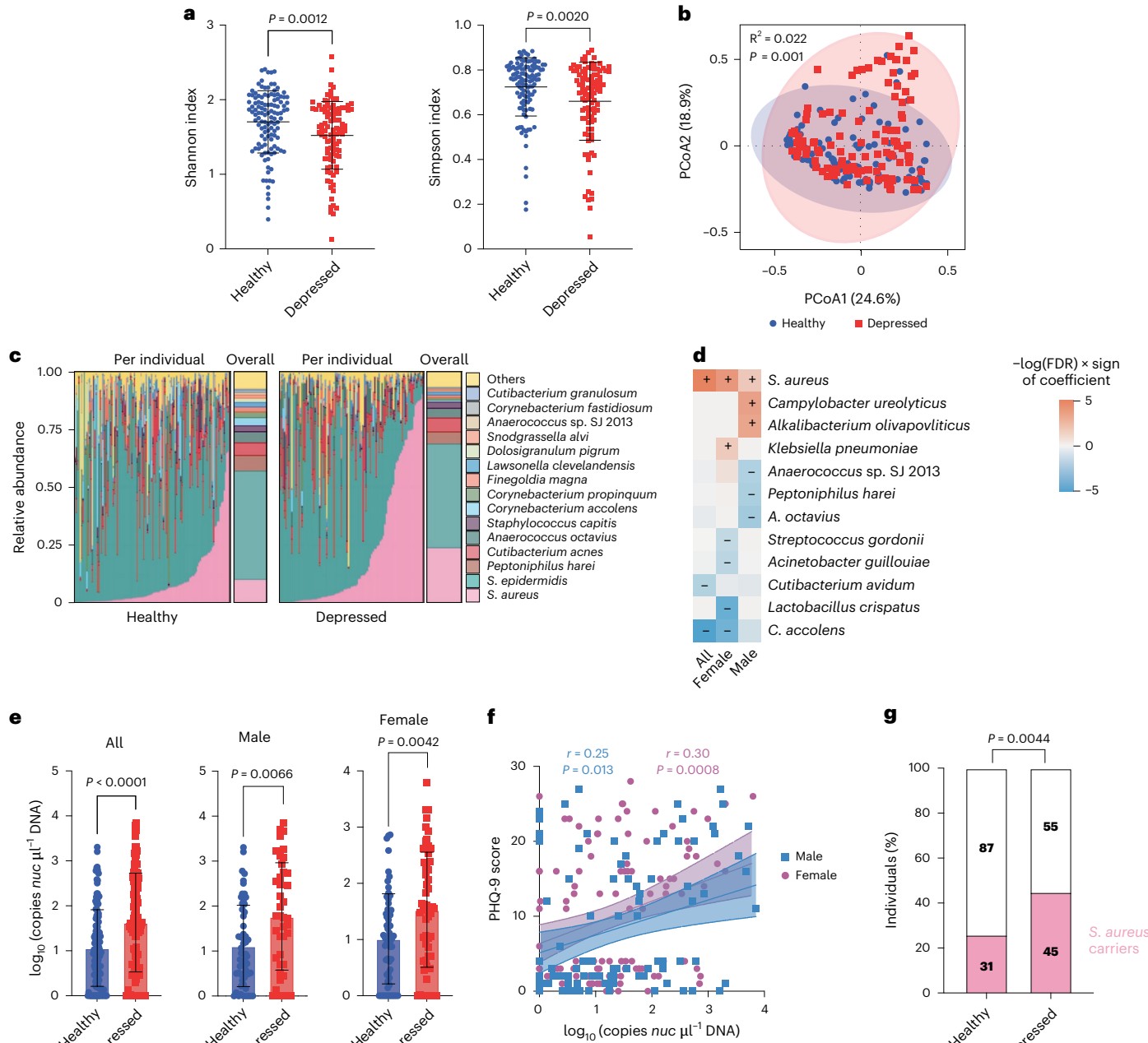

**Fig. 1 | Nasal *S. aureus* carriage is associated with depression. a–g**, Results from 16S rRNA microbiome sequencing. **a**, The Shannon (left) and Simpson (right) methods were used to calculate the α-diversity in the nasal microbiomes of healthy individuals and (*n* = 118) and patients with depression (*n* = 100) at the species level. **b**, β-Diversity comparison of the composition of the nasal microbiomes of the two study cohorts at the species level. Shaded areas show the 95% confidence interval. **c**, Per-individual and overall relative abundances of the most abundant species in the nasal microbiomes of healthy individuals (*n* = 118) and individuals with depression (*n* = 100). **d**, Multivariate analysis adjusted for age, sex, body mass index, education, income, family relationships, adverse childhood experiences, available supportive friends and batch covariates. Fields marked with + and − signs represent significant associations (adjusted false discovery rate (FDR) < 0.05). A genus-level analysis is shown in Extended Data Fig. 1a,b. **e**, Actual abundance of *S. aureus* in the two cohorts (left) as well as among the males (middle) and females (right) among those cohorts.

**f**, Correlation of *S. aureus* abundance with PHQ-9 depression score in males and females among all individuals. The line shows simple linear regression; *r*, correlation coefficient. Shaded areas show the 95% confidence intervals. An analysis of the GAD-7 anxiety score is provided in Extended Data Fig. 1f. **e,f**, The *S. aureus* abundance was determined as the log-transformed number of *nuc* copies per µl of DNA. **g**, *S. aureus* carrier status in the two study cohorts. An individual was defined as a *S. aureus* carrier if at least one of 24 randomly selected bacterial clones in the culture-based analysis was found to be *S. aureus* using matrix-assisted laser desorption ionization–time-of-flight mass spectrometry (MALDI-TOF-MS) analysis. The number of individuals in the different groups are shown in the bars. Statistical analysis was performed using a two-sided Mann–Whitney test (**a,e**), permutational multivariate analysis of variance (PERMANOVA; **b**), Spearman's correlation (**f**) or Fisher's exact test (**g**). Data are the mean ± s.d.

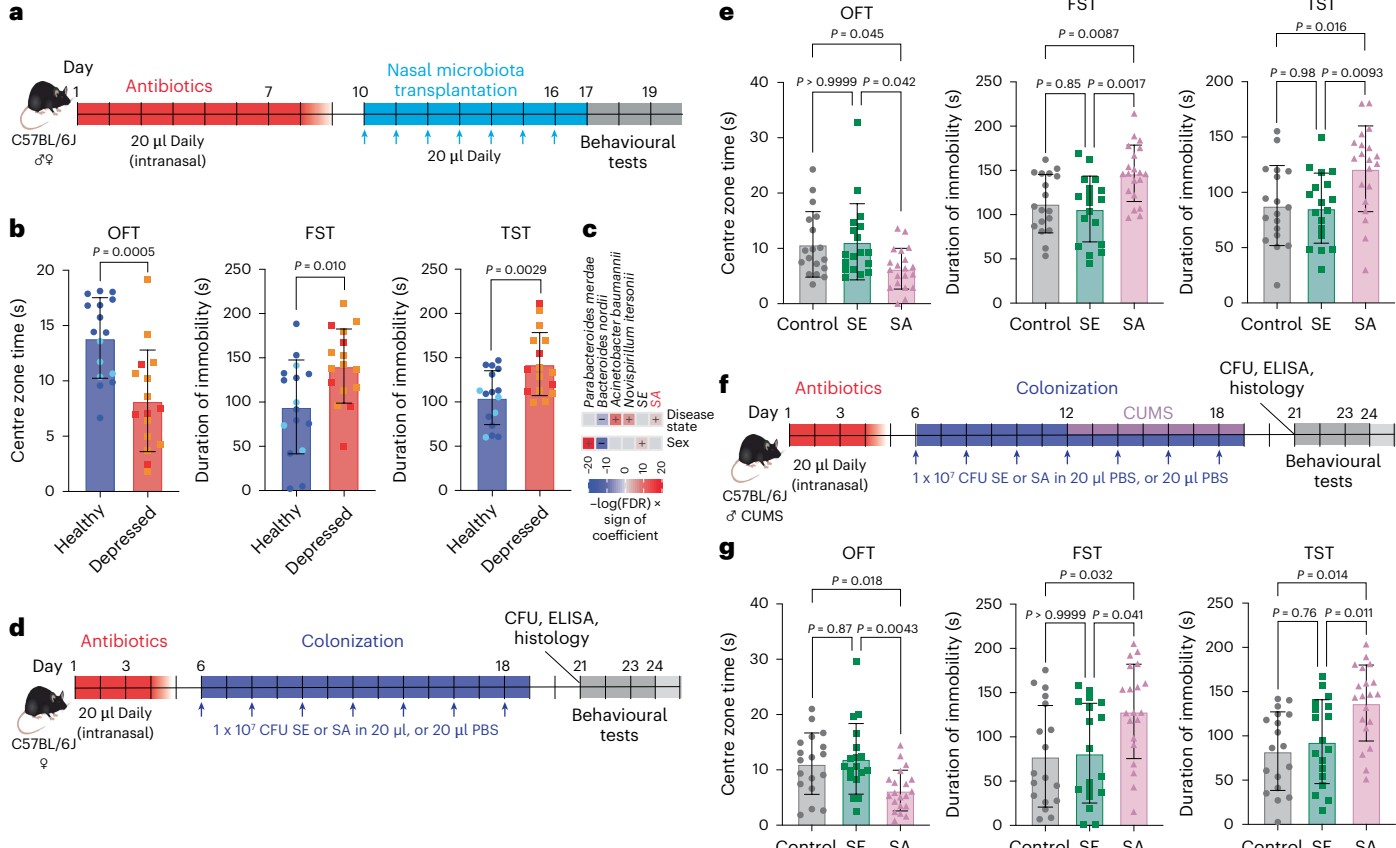

**Fig. 2 | Nasal *S. aureus* carriage induces depressive behaviour in mice. a**, Set-up of the experiment to investigate the impact of nasal microbiome transplantation on anxiety and depression-like behaviour in mice. **b**, Anxiety-like behaviour of mice in the OFT test (left) as well as depression-like behaviour in FST (middle) and TST (right) tests. Male and female mice received nasal transplants from male and female donors, respectively (one donor per one mouse); *n* = 16 mice (*n* = 8 male and *n* = 8 female; sex-specific analysis in Extended Data Fig. 3a) per group. Light blue and orange symbols depict *S. aureus*-positive (by culture analysis) transplanted microbiomes. **c**, Multivariate analysis (covariate-adjusted analysis using MaAsLin2) adjusted for sex. **d**, Set-up of the experiment to investigate anxiety and depression-like behaviour in female mice nasally colonized with *S. aureus* or *S. epidermidis* and control mice. **e**, Anxiety-like behaviour of mice in the OFT (left) as well as depression-like behaviour in FST (middle) and TST (right) tests. Data for the corresponding experiment with

male mice in Extended Data Fig. 6. **f**, Set-up of the experiment to investigate anxiety and depression-like behaviour in male mice nasally colonized with *S. aureus* or *S. epidermidis* and control mice with CUMS treatment. Bacteria were administered seven times at 48-h intervals, including during the seven-day CUMS protocol. **g**, Anxiety-like behaviour of mice in the OFT (left) as well as depression-like behaviour in the FST (middle) and TST (right) tests. **e**,**f**, *n* = 18 (control and SE) or 20 (SA) mice per group. Data are the mean ± s.d. Statistical analysis was performed using a two-sided unpaired Student's *t*-test (**b**), or two-sided one-way analysis of variance (ANOVA) or Kruskal–Wallis test (depending on the normality of distribution in the groups) and Tukey's and Dunn's post-tests, respectively (**e**,**g**). Colonization levels and evidence for absence of disease and inflammation in Extended Data Figs. 4 and 5. SA, *S. aureus*; SE, *S. epidermidis*. Credit: mouse illustrations in **a**,**d**,**f**, creazilla under a Creative Commons license CC0 1.0.

## Reduced nasal sex hormone levels are associated with depression in mice

To explain our findings mechanistically, we hypothesized that an increased abundance of *S. aureus* alters metabolism in the nose, which could modulate brain physiology and behaviour. We therefore performed tandem mass spectrometry (MS/MS)-based metabolomic profiling of nasal metabolites in randomly selected individuals from both groups (*n* = 40; population characteristics of those subsets in Supplementary Table 3).

According to a Bray–Curtis distance-based evaluation of principal coordinate analysis (PCoA; Fig. 3a) and an orthogonal partial least squares-discriminant analysis (OPLS-DA; Fig. 3b), there was a significant difference between the nasal metabolomes of healthy individuals and patients with depression. Significantly altered pathways predominantly included pathways related to steroid hormone biosynthesis (Fig. 3c). Furthermore, nasal testosterone and estradiol levels were significantly decreased in patients with depression compared with healthy individuals (Fig. 3d,e), whereas the serum levels of these hormones were similar (Extended Data Fig. 6c). Moreover,

the nasal estradiol and testosterone levels were significantly and at 'fair' to 'moderate' degrees[16] correlated with the anxiety and depression scores (Fig. 3f,g and Extended Data Fig. 6d). These findings demonstrated that differential concentrations of sex hormones in the nose are associated with depression, suggesting that depression may be impacted by nasal microbiome members via alteration of sex hormone levels.

## Nasal *S. aureus* degrades sex hormones and is associated with decreased dopamine and serotonin levels in the brain

In humans testosterone is converted to androstenedione and estradiol to estrone by 17β-hydroxysteroid dehydrogenases (HSDs; Fig. 4a). We hypothesized that *S. aureus* produces a similar activity. In accordance with the idea that *S. aureus* degrades sex hormones in the nose, nasal estradiol and testosterone in humans were at a 'fair' degree[16] negatively correlated with the abundance of *S. aureus*, with the correlation being stronger for estradiol (Fig. 4b). Only *S. aureus* was associated with increases in metabolites belonging to steroids and their derivatives

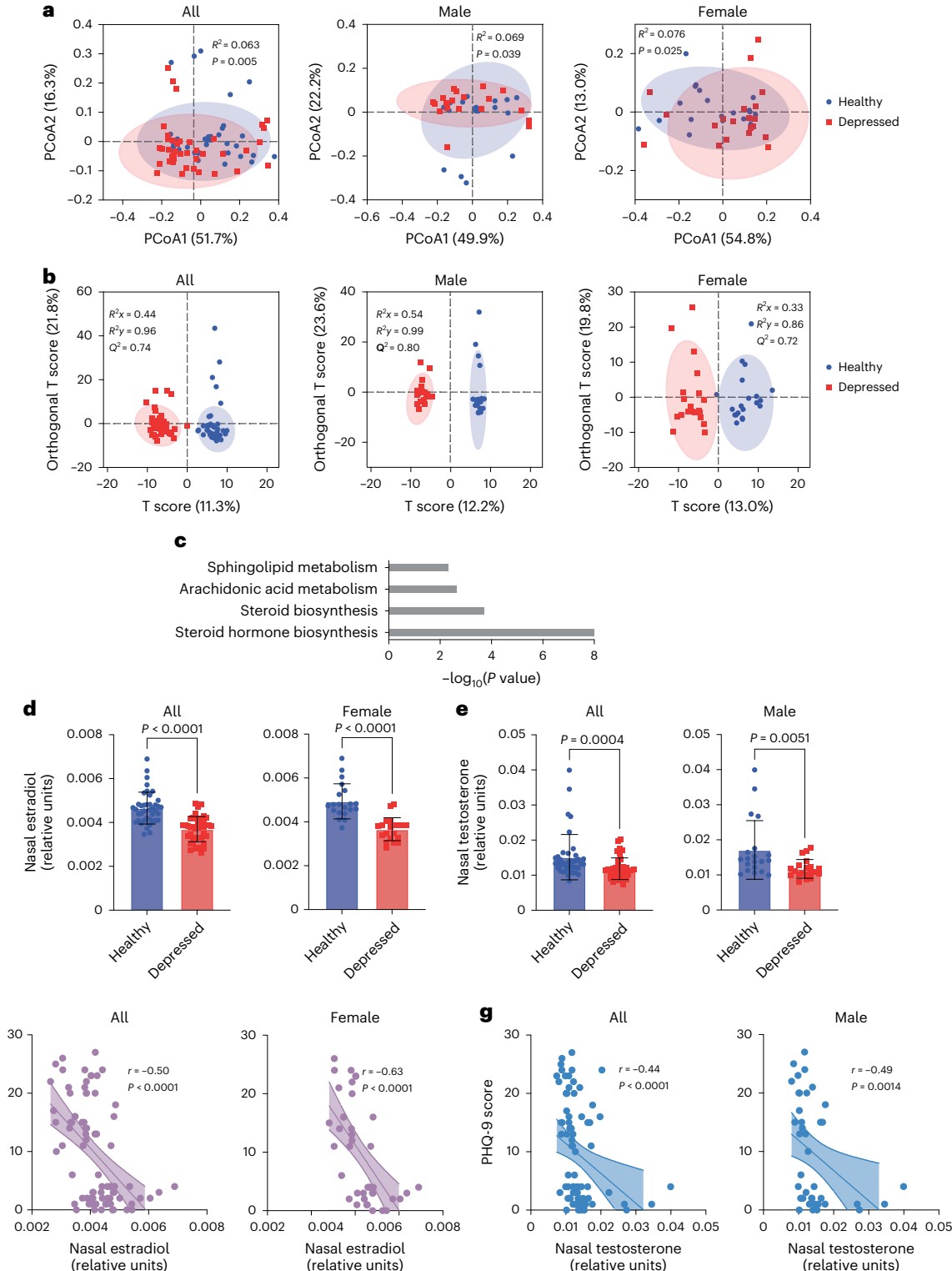

**Fig. 3 | Reduced nasal sex hormone levels are associated with depression.**
**a**,**b**, The nasal metabolomes of individuals in the two cohorts ($n$ = 40 each; randomly selected) were analysed using PCoA with Bray–Curtis distance-based PERMANOVA. $R^2$ proportion of variance explained by the principal coordinate (**a**) and OPLS-DA analyses, with the model fit parameters ($R^2x$, $R^2y$) and predictive ability ($Q^2$) shown (**b**). **c**, Significantly altered metabolic pathways by KEGG enrichment. **d**,**e**, Nasal estradiol (**d**) and testosterone (**e**) levels in the same randomly selected cohort subsets. Data are the mean ± s.d. Serum levels are provided in Extended Data Fig. 6c. **f**,**g**, Correlation between the levels of nasal estradiol (**f**) and testosterone (**g**), and PHQ-9 depression scores. GAD-7 anxiety score analysis in Extended Data Fig. 6d. **a**,**b**,**f**,**g**, Lines show simple linear regression and shaded areas the 95% confidence intervals. Statistical analysis was performed using a two-sided Mann–Whitney test (**d**,**e**) or Spearman's correlation (**f**,**g**).

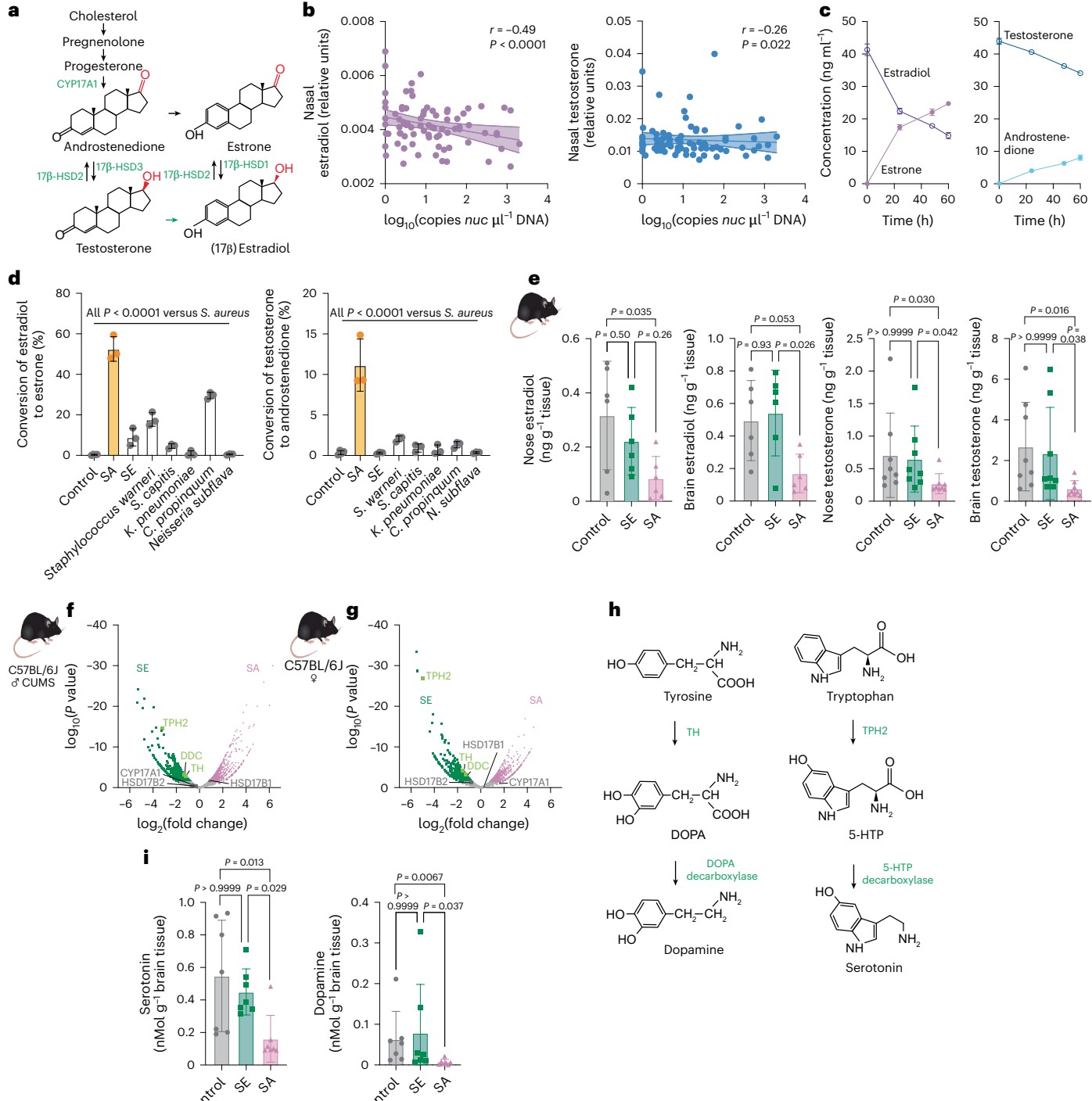

**Fig. 4 | Nasal *S. aureus* degrades sex hormones and is associated with decreased dopamine and serotonin levels in the brain. a,** Testosterone and estradiol biosynthesis. Note the HSD-catalysed conversion of androstenedione/testosterone and estradiol/estrone. Enzyme names are in green font; hydroxyl and keto groups affected by the HSD enzymes are in red. **b,** Correlation between nasal testosterone (right) and estradiol (left) concentrations, and nasal *S. aureus* abundance in combined selected cohort subsets (*n* = 80). The line shows the simple linear regression. Shaded areas show 95% confidence intervals. **c,** In vitro conversion of estradiol to estrone (left) and testosterone to androstenedione (right) by *S. aureus* culture filtrate within 24 h (*n* = 3 per group). **d,** Conversion of estradiol to estrone (left) and testosterone to androstenedione (right) by culture filtrates from different bacteria. Tryptone soya broth (TSB) control, *n* = 4; *n* = 3 for all other groups. **e,** Nasal and midbrain levels of testosterone and estradiol in mice colonized with *S. aureus* or *S. epidermidis*, and in controls. Estradiol was measured in female mice (*n* = 6 per group) and testosterone in male mice

(*n* = 8 per group). **f,g,** Transcriptome analysis of midbrain gene expression in male and female mice nasally colonized with *S. aureus* compared with *S. epidermidis*. Red, increased in *S. aureus*-colonized mice; green, increased in *S. epidermidis*-colonized mice. Dopamine and serotonin biosynthesis enzymes are labelled; *n* = 3 per group. **e–g,** Experimental set-up as depicted in Fig. 2d,f. **h,** Dopamine and serotonin biosynthesis. Enzyme names are in green font. **i,** Concentrations of serotonin (left) and dopamine (right) in the midbrains of *S. aureus*-colonized, *S. epidermidis*-colonized and control mice; *n* = 7 mice (*n* = 4 male and *n* = 3 female) mice per group. The GABA and glutamate concentrations are in Extended Data Fig. 9f. **c–e,i,** Data are the mean ± s.d. Statistical analysis was performed using a two-sided one-way ANOVA or a Kruskal–Wallis test, depending on the normality of distribution in the groups, and Tukey's and Dunn's post-tests, respectively (**d,e,i**), or Spearman's correlation (**b**). SA, *S. aureus*; SE, *S. epidermis*. Steroid formulae in **a** and **h** created with BioRender.com. Credit: mouse illustrations in **e–g**, creazilla under a Creative Commons license CC0 1.0.

in a multivariate metabolomics analysis (Supplementary Fig. 1). Furthermore, *S. aureus* had the ability to convert estradiol to estrone and testosterone to androstenedione in vitro, with the estradiol-converting activity stronger than the testosterone-converting activity (Fig. 4c). Although some other nose-colonizing bacteria that we tested exhibited similar activities in vitro, these were significantly less pronounced than in *S. aureus* (Fig. 4d). All tested *S. aureus* sequence types exhibited similar activities (Extended Data Fig. 7a). In the mouse nasal colonization model, *S. aureus* colonization led to significantly reduced levels of nasal estradiol and testosterone compared with controls and *S. epidermidis* colonization (Fig. 4e), whereas the levels of sex hormones in serum remained unchanged (Extended Data Fig. 7b). *S. aureus* also had a similar effect on sex hormone levels in the brain, substantiating our hypothesis about the impact of the nasal microbiome, and particularly *S. aureus* on, brain metabolites. The changed levels of sex hormones in the brain were not due to differences in de novo biosynthesis, as the levels of the precursors pregnenolone and progesterone were unchanged (Extended Data Fig. 7c). Furthermore, transcriptomic analysis of midbrain samples of mice nasally colonized with *S. aureus* showed that expression of the CYP17A1 and HSD genes involved in sex hormone biosynthesis were unchanged between the mice that were nasally colonized with *S. aureus* and those colonized with *S. epidermidis* as controls (Fig. 4f,g).

Estradiol and testosterone are known to increase the levels of dopamine and serotonin in the midbrain[22,23], a region that is known to play a crucial role in emotional regulation[24]. These neurotransmitters have a well-known impact on depression. Dopamine is synthesized from tyrosine via dihydroxyphenylalanine (DOPA) in two steps by tyrosine hydroxylase (TH) and DOPA decarboxylase (DCC). Serotonin (5-hydroxytryptamine, 5-HT) is synthesized from tryptophan via 5-hydroxytryptophan (5-HTP) by tryptophan hydrolase (TPH2) and 5-HTP decarboxylase (Fig. 4h). The transcriptomic analysis also revealed that TPH2, TH and DOPA decarboxylase transcripts were significantly less abundant in both male and female mice than in their *S. epidermidis*-colonized counterparts (Fig. 4f,g), indicating that sex hormone degradation by *S. aureus* in the nose affects dopamine and serotonin levels in the brain via downregulation of their biosynthesis. Changes in the expression of these genes were verified by quantitative PCR with reverse transcription of midbrain tissue samples (Extended Data Fig. 7d). In accordance with these findings, Kyoto Encyclopedia of Genes and Genomes (KEGG) pathway enrichment analysis revealed significant changes in related metabolic pathways (for example, tyrosine and tryptophan metabolism; Extended Data Fig. 7e). Importantly, we then verified that decreased biosynthesis results in decreased serotonin and dopamine concentrations in the brains of mice nasally colonized with *S. aureus* (Fig. 4i). In contrast, the concentrations of other neurotransmitters were not affected (Extended Data Fig. 7f).

## An *S. aureus* HSD degrades sex hormones and induces depressive behaviour in mice

To elucidate the mechanism by which *S. aureus* degrades sex hormones, we searched the genome of the *S. aureus* nasal isolate that we used in this study and whose genome we sequenced for putative *hsd* genes. To that end, we searched for members of the short-chain dehydrogenase/reductase (SDR) family, which have physiological roles in steroid hormone, prostaglandin and retinoid metabolism[25] (Supplementary Table 4). Excluded from further analysis were genes with known different functions. We then expressed the selected putative *hsd* genes in *Escherichia coli* and tested for the ability to degrade estradiol and testosterone. Only one expression strain, expressing the putative *hsd* gene number 12 (*hsd12*), showed a strong capacity to degrade estradiol and testosterone. This identified the *hsd12* gene as responsible for sex hormone degradation in *S. aureus* and as a previously unknown 17β-HSD, isozymes of which are used by the host to maintain homeostasis of

active estradiol and testosterone[26] (Fig. 5a,b). The *hsd12* gene, annotated in the National Center for Biotechnology Information (NCBI) as an SDR family oxidoreductase, was present in all *S. aureus* isolates obtained in this study as well as highly conserved in published *S. aureus* sequences. It is in a region with predominantly genes without known functions (Extended Data Fig. 8a).

To link the observed *S. aureus*-mediated phenotypes directly to the *hsd12* gene, we created an *S. aureus hsd12*-deletion mutant. Deletion of *hsd12* did not change in vitro growth in different media and had only a very minor influence on the expression of other genes in the genome by RNA-sequencing (RNA-Seq) analysis (Extended Data Fig. 8b,c). Colonization with the *hsd12*-deletion strain in female and CUMS-treated male mice led to phenotypes indistinguishable from those observed in the controls but significantly different from those observed in mice colonized with the isogenic wild-type *S. aureus* strain (Fig. 5c–l). We also verified that bacterial colonization did not lead to changes in weight and temperature (Extended Data Fig. 9a,b), the mice did not show signs of illness (Extended Data Fig. 9c), and inflammation and cytokine and histology analyses did not show significant differences between the *S. aureus* and isogenic *hsd12*-deletion strains (Extended Data Fig. 9d–g). These findings indicated that the *hsd12* gene is mechanistically responsible for the link between *S. aureus* nasal colonization with depression.

## Depression-inducing capacity of a mouse-adapted *S. aureus* strain

To rule out that the observed impact of *S. aureus* on depressive behaviour is strain-specific, we also performed key experiments using a strain (JSNZ)[27] of sequence type 88 (ST88) that has been reported to be mouse-adapted[27]. *S. aureus* ST88 degraded estradiol and testosterone similarly to the ST398 strain that we used generally in this study (Fig. 6a). We created an *hsd*-deletion mutant of ST88, which did not show any growth defects or considerable changes in gene expression, excepting the *hsd* gene (Extended Data Fig. 8d,e), and exhibited markedly reduced capacity to degrade estradiol and testosterone (Fig. 6b).

We then set up a mouse colonization model that replicated the general set-up of the model used with strain ST398 (Fig. 2d,f) but only using two nasal instillations with a strongly reduced bacterial amount seven days apart—a change prompted by the reported better overall mouse colonization capacity of that strain[27] (Fig. 6c). Strain ST88 colonized the antibiotic-treated noses over the time course of the experiment like ST398 and like ST398 did not colonize the lungs but showed much higher capacity to colonize the gut (Extended Data Fig. 10a; compare with Extended Data Fig. 5g). Although the use of a mouse-adapted strain might better reflect a 'natural' colonization scenario; this means that results obtained with ST88 should be interpreted with the caveat that they might not exclusively be due to the nose–brain but also the gut–brain axis. ST88 colonization did not affect weight, temperature, disease or cytokine expression in the noses and lungs, and deletion of *hsd* did not affect colonization (Extended Data Fig. 10b–e). In most of the behavioural tests, *S. aureus* ST88 increased depressive behaviour, whereas application of ST88Δ*hsd* led to phenotypes indistinguishable from the control (Fig. 6d). These results showed that the promotion of depressive behaviour in mice by the Hsd enzyme of *S. aureus* is not strain-dependent.

Finally, using strain ST88, we tested whether *S. aureus* degrades sex hormones artificially introduced into the noses of mice. To that end, we first eliminated the effects of physiological hormonal fluctuations via a bilateral ovariectomy in female mice and bilateral orchidectomy in male mice, and subsequently used a nasal hydrogel for sustained release of estradiol or testosterone (Fig. 6e). We used a hydrogel as it enables better brain targeting and avoids rapid renal or intestinal excretion, which we demonstrated with fluorescein isothiocyanate (FITC; Extended Data Fig. 10f). Furthermore, we ascertained that

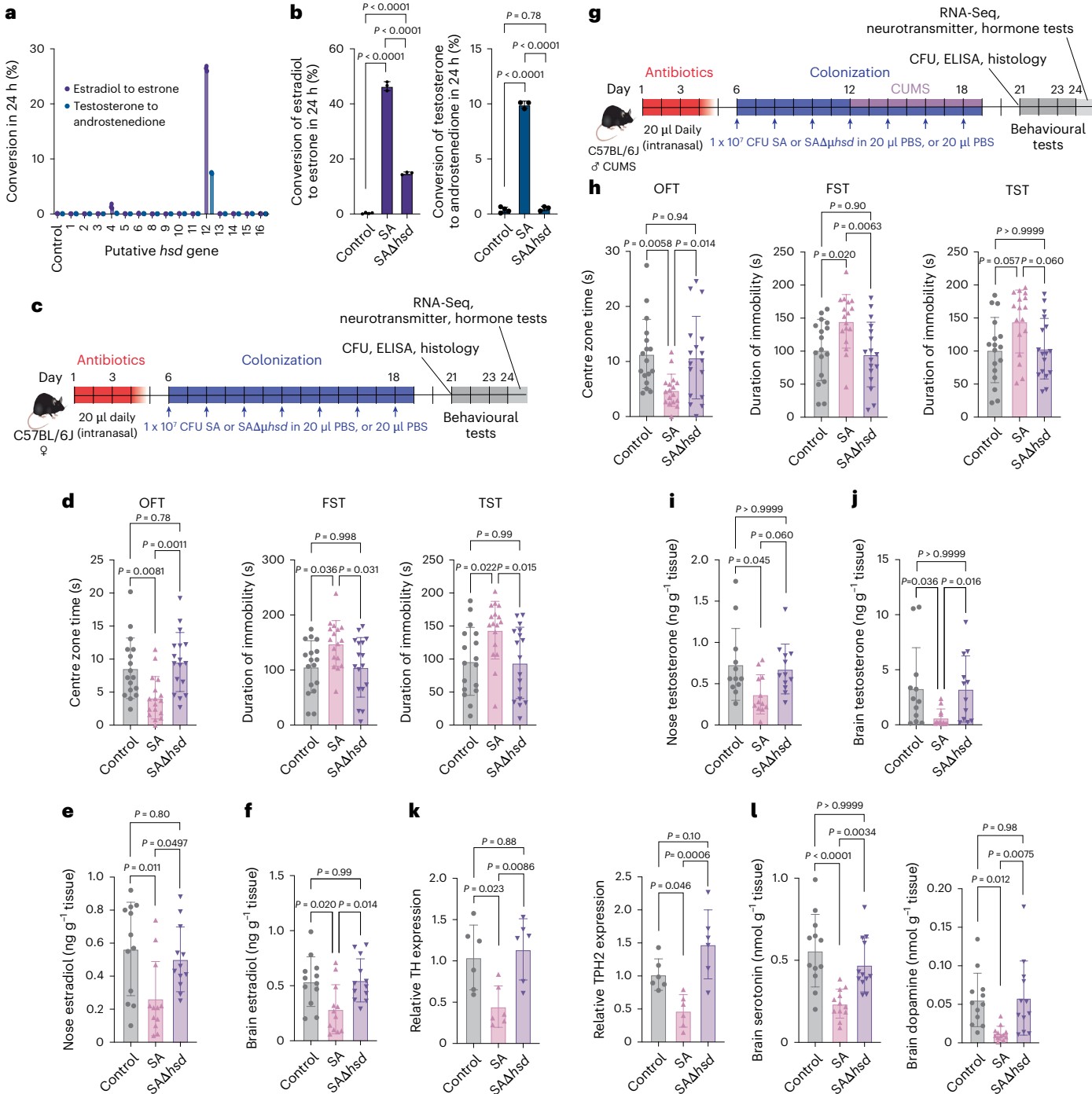

**Fig. 5 | An *S. aureus* HSD degrades sex hormones and induces depressive behaviour in mice. a**, Conversion of testosterone to androstenedione and estradiol to estrone by putative *S. aureus hsd* genes expressed in *E. coli*. The putative *hsd7* gene was not included as it could not be expressed, probably due to its large size (*n* = 3 per group). **b**, Conversion of testosterone to androstenedione and estradiol to estrone by culture filtrates from *S. aureus* wild type and its *hsd12* isogenic mutant. TSB control, *n* = 4; other groups, *n* = 3. **c**, Set-up of the model to test for induction of anxiety and depression-like behaviour in female mice by nasal colonization with *S. aureus* wild type, its *hsd12* isogenic mutant and controls. **d**, Anxiety-like behaviour of female mice in the OFT (left) as well as depression-like behaviour in FST (middle) and TST (right) tests (*n* = 17 per group). **e,f**, Nose (**e**) and midbrain (**f**) estradiol levels in female mice (*n* = 12 per group). **g**, Set-up of the model to test for induction of anxiety

and depression-like behaviour in male CUMS-treated mice by nasal colonization with *S. aureus* wild type, its *hsd12* isogenic mutant and controls. **h**, Anxiety-like behaviour of male mice in the OFT (left) as well as depression-like behaviour in FST (middle) and TST (right) tests (*n* = 17 per group). **i,j**, Nose (**i**) and midbrain (**j**) testosterone levels in male mice (*n* = 12 per group). **k**, Relative expression of TH and TPH2 genes in midbrain samples (*n* = 6 per group; *n* = 3 females and *n* = 3 CUMS-treated males). **l**, Midbrain serotonin (left) and dopamine (right) levels (*n* = 12 per group; *n* = 6 females and *n* = 6 CUMS-treated males). Statistical analysis was performed using a two-sided one-way ANOVA or Kruskal–Wallis test, depending on normality of distribution in the groups, and Tukey's and Dunn's post-tests, respectively. Data are the mean ± s.d. SA, *S. aureus*; SAΔ*hsd*, *S. aureus hsd12* isogenic mutant. Credit: mouse illustrations in **c**,**g**, creazilla under a Creative Commons license CC0 1.0.

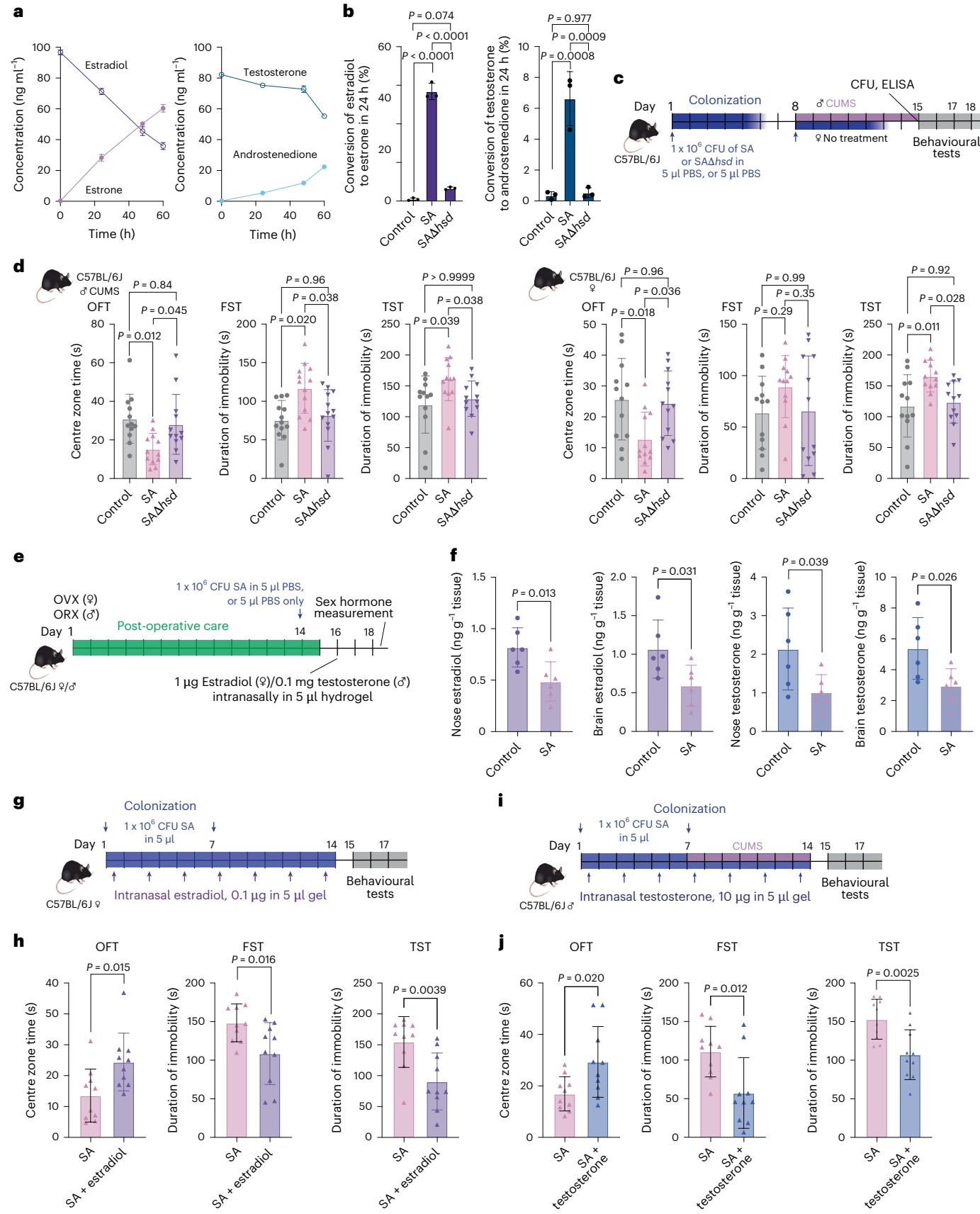

**Fig. 6 | Depression-inducing capacity of a mouse-adapted _S. aureus_ strain.**
**a**, In vitro sex hormone conversion by _S. aureus_ ST88 culture filtrate (_n_ = 3 per group). **b**, Sex hormone conversion by culture filtrates from _S. aureus_ ST88 wild type and _hsd12_ isogenic mutant (_n_ = 3 per group). **c**, Set-up of model to test for the induction of anxiety and depression-like behaviour in mice by nasal colonization with _S. aureus_ wild type, its _hsd12_ isogenic mutant and controls. **d**, Anxiety-like behaviour in the OFT (left) as well as depression-like behaviour in FST (middle) and TST (right) tests (_n_ = 12 per group). **e**, Set-up of the model to measure degradation of applied sex hormones to the noses of mice. ORX, orchidectomy; OVX, ovariectomy. **f**, Detected sex hormone levels (estradiol in female and testosterone in male mice) with and without _S. aureus_ application (_n_ = 6 per group). **g**, Set-up of the model to test for depression and anxiety-like

behaviour in female mice after application of sex hormones (in the background of _S. aureus_ colonization). **h**, Anxiety-like behaviour in the OFT (left) as well as depression-like behaviour in FST (middle) and TST (right) tests (_n_ = 10 per group). **i,j**, Corresponding set-up (**i**) of the experiment in **g** and results in male mice (**j**; _n_ = 10 per group). Statistical analysis was performed using a two-sided one-way ANOVA with Tukey's (**b**,**d**) post-test with the exception of the male mice TST in **d** where a two-sided Kruskal–Wallis test with Dunn's post-test was used, or and unpaired two-sided Student's _t_-test (**f**,**h**,**j**) with the exception of female TST and OFT and male FST in **h**,**j** where a two-sided Mann–Whitney test was used. Data are the mean ± s.d. SA, _S. aureus_ ST88 wild type; SAΔ_hsd_, _S. aureus hsd12_ isogenic mutant. Credit: mouse illustrations in **c**–**e**,**g**,**i**, creazilla under a Creative Commons license CC0 1.0.

---

the gel releases loaded estradiol and testosterone rapidly and virtually completely over a period of two days (Extended Data Fig. 10g). Application of _S. aureus_ together with estradiol- or testosterone-loaded hydrogel significantly decreased nasal and brain estradiol and testosterone levels (Fig. 6f). Finally, we showed that hydrogel-mediated nasal application of sex hormones significantly alleviated anxiety and depressive behaviour induced by _S. aureus_ application (Fig. 6h–k). In addition to confirming the impact of nasal sex hormones on depression and anxiety-related phenotypes, these findings provide further direct evidence for _S. aureus_-mediated degradation of sex hormones in the nasal cavity and the impact of this mechanism on depression-like behaviour.

## Discussion

In this study we show that the nasal microbiome influences depression dependent on the presence of _S. aureus_ in the nose. We demonstrate that _S. aureus_ degrades the sex hormones testosterone and estradiol by expressing a previously unknown 17β-HSD isozyme that is encoded by a gene we named _hsd12_, which led to decreased concentration of dopamine and serotonin in the brain in mouse models. Although we found that some other nasal commensal bacteria also had some sex hormone-degrading activity in vitro, their abundance was not increased in humans suffering from depression.

Peripheral bacterial infection[28–30] as well as the composition of the gut microbiome[6–9,31] have been shown to impact neurological disease. However, a role of the nasal microbiome in neurological diseases has remained speculative or based merely on correlative evidence[12,32]. Our discovery of a mechanism that causes an impact of the nasal microbiome on depression provides previously unavailable evidence for a key role of the nose–brain axis in the development of neurological disease.

The field of microbial endocrinology is rapidly advancing[33,34], with several recent studies illustrating hormone conversion by gut bacteria[8,9,35,36]. By identifying an enzyme in the nasal microbiome that converts steroid sex hormones, our research indicates the presence of isozymes involved in steroid metabolism within host-associated environments beyond the intestine.

Both male and female sex hormones have been implicated in determining depressive behaviour[22,37] and there are distinct sex-dependent differences in the incidence of depression[38]. Accordingly, we observed a stronger correlation between nasal estradiol levels, compared with testosterone, with depressive behaviour in humans and more pronounced induction of depressive behaviour by _S. aureus_ in female mice compared with males. Our finding that _S. aureus_ degrades the female sex hormone estradiol much more rapidly than the male sex hormone testosterone may explain these differences, and _S. aureus_ nasal colonization may thus contribute to the observed sex-dependent difference in depression prevalence.

The impact of commensal _S. aureus_ in the nasal microbiome on depression is categorically different from that of infection on neurological disease that is inflammation-associated[28–30]. Asymptomatic _S. aureus_ presence in the nose, however, represents a known risk factor

for _S. aureus_ infection[14,39]. Our study adds the neurological disease depression to the list of morbidities for which _S. aureus_ represents a risk factor, and _S. aureus_ nasal decolonization could thus also have an impact on the incidence of depression.

Our study has limitations. We investigated a relatively small human cohort and relied predominantly on mechanistic studies in mice to support our conclusions. Our cohort of young adults without comorbidities cannot represent populations with socioeconomic adversity and adolescent or elderly depression. Although we matched key confounders, unmeasured factors may contribute to residual confounding. Furthermore, no method of _S. aureus_ detection is entirely accurate, which is why we included several different methods in our study. Finally, it is not entirely clear how the bacteria benefit from sex hormone-degrading enzymes. Potentially, they utilize them as carbon sources or to eliminate estradiol-mediated stress[40].

In conclusion, our discovery of a significant impact of the nasal microbiome on a widespread neurological disease highlights the importance of the nose–brain axis and nasal bacterial colonization for such diseases.

## Methods

### Inclusion and ethics

The research complied with all applicable ethical regulations. The human clinical study was approved by the ethics committee of Renji Hospital, Shanghai Jiao Tong University School of Medicine, Shanghai, China (approval number KY2022-139-B). All animal procedures followed the ethical guidelines outlined in the Guide for the Care and Use of Laboratory Animals proposed by the Institute for Laboratory Animal Research of the National Academy of Sciences and all protocols were approved by the Animal Welfare Committee of Renji Hospital, Shanghai Jiao Tong University School of Medicine, Shanghai, China (approval number RJ2023-025B).

### Human participants

Healthy volunteers were recruited from the Physical Examination Center of Renji Hospital. Patients who had been newly diagnosed with depression were recruited from the Department of Psychological Medicine of Renji Hospital. All participants were ethnically Han Chinese. Each participant was provided with a detailed written questionnaire by the research physicians. Informed consent was obtained from all human research participants. Patients with depression and healthy volunteers were enrolled between November 2022 and November 2024. All participants were between 18 and 44 years of age. The patients with depression were diagnosed according to the _Diagnostic and Statistical Manual of Mental Disorders_ (DSM-IV)[41] by clinical psychologists and had no other psychiatric disorders or family history of any psychiatric disorders. The severity of depression and anxiety was evaluated using PHQ-9 and GAD-7 scores, respectively. The healthy controls were physically healthy and lacked any neurological illness or related family history, with PHQ-9 and GAD-7 scores both below four. Other exclusion criteria for both control and depression groups included nasal or oral diseases, previous use of any

type of antidepressant medication within the past three months, use of antimicrobial drugs within the past four weeks, thyroid dysfunction, diabetes, hypertension, autoimmune diseases, tumours and pregnancy or lactation.

Due to the minimal risks involved in the sampling process of this study, only non-economic compensation measures were adopted. Professional psychologists were provided for participants when they experienced significant discomfort or needed immediate psychological support during the research process.

### Sample collection

Two nasal swab samples were collected from the nasal cavity of each participant using Copan swabs. With the participant's head tilted slightly backward, a trained physician gently inserted the swab along the nasal cavity to a depth of 2–3 cm without endoscopic guidance or local anaesthesia. The swab was rotated firmly against the mucosal surface for 15–20 s to ensure adequate specimen collection before slowly withdrawing. For microbiota analysis, the collected swabs were immediately immersed in 1 ml sterile saline. All collected swabs were maintained at 4 °C and processed within 4 h of collection. For metabolome analysis, the collected swabs were immediately stored at −80 °C without sterile saline. We collected 119 serum samples from participants who agreed to provide blood samples. After collection, the serum samples were immediately aliquoted and stored at −80 °C.

### Culture-based analysis

Nasal swabs were vortexed in 1 ml sterile saline for 2 min. Aliquots of 100 μl from each swab were diluted, plated on 5% sheep blood agar and incubated at 37 °C for 24 h. Twenty-four random colonies were isolated and identified by matrix-assisted laser desorption ionization–time-of-flight mass spectrometry (MALDI-TOF-MS, Bruker Daltonics) in each sample.

### Full-length 16S rRNA gene sequencing

Nasal swabs were immersed in 1 ml sterile saline and vortexed for 2 min. Each sample (500 μl volume) was centrifuged at 13,000$g$ and 4 °C for 10 min. The pellets were resuspended in 180 μl Buffer ATL (QIAamp DNA mini kit, QIAGEN, 51306) with 5 μl lysozyme (50 mg ml$^{-1}$; Sigma, L6876) and 5 μl lysostaphin (1 mg ml$^{-1}$; Sigma, L4402), and incubated at 37 °C for 30 min. Next, DNA was extracted according to the manufacturer's instructions. The full-length *16S* rRNA gene was amplified using the primers 27F (5′-AGRGTTYGATYMTGGCTCAG-3′) and 1492R (5′-RGYTACCTTGTTACGACTT-3′). The amplicons were then sequenced on the PacBio platform. For negative controls, a full sequencing protocol was applied to a sterile swab.

### Amplicon sequence analysis

Sequence analysis was performed using the DADA2 workflow in the QIIME2 software pipeline[42]. Initially, primers and adaptors were removed and sequences with a quality score of less than three or expected errors greater than two were filtered out, retaining sequences with lengths between 800 and 1,800 base pairs (bp). Further processing with DADA2 included the removal of duplicate sequences, learning error models, inferral of amplicon sequence variants (ASVs) and removal of chimaeras to obtain an ASV feature table. To mitigate potential contamination effects, all ASVs detected in the negative controls were removed from subsequent analyses (Supplementary Data 1). Batch correction was performed with MMUPHin[43]. Only ASV sequences present in at least two samples were retained to eliminate spurious features. The taxonomy of ASV sequences was analysed using RDP Classifier version 2.13 against the NT_16S (v20221012) database. Taxonomic α-diversity was estimated using Shannon and Simpson indices. The β-diversity between groups was measured using PCoA based on Bray–Curtis dissimilarity and compared using the PERMANOVA method with the vegan R (v2.6-4) package. Microbiome taxonomic

abundance data were analysed using general linear models by MaAsLin2 (ref. 44), adjusted for demographic variables (age, sex and body mass index), socioeconomic factors (education and income), family relationships, adverse childhood experiences, social support (availability of supportive friends) and technical factors (batch). The *P* values were adjusted for multiple comparisons using FDR based on the Benjamini–Hochberg method. Differentially abundant genera and species were determined using FDR < 0.05. Absolute quantification of *S. aureus* genomic copies was achieved through quantitative PCR analysis utilizing a standard curve developed via successive 1:10 dilutions of linearized plasmid DNA templates containing cloned *nuc* sequences with pre-determined copy number.

### Whole-genome sequencing

A single *S. aureus* isolate per specimen was randomly selected for whole-genome sequencing. Genomic DNA was isolated from the bacterial cell pellets using a Bacterial DNA kit (OMEGA) according to the manufacturer's instructions. Paired-end libraries with insert sizes of 150 bp were prepared following Illumina's standard genomic DNA library preparation procedure. The qualified Illumina paired-end library was used for Illumina NovaSeq 6000 sequencing (150 bp × 2; Shanghai BIOZERON Co., Ltd). The raw paired-end reads were filtered using fastp v0.12.5 and de novo assembly was performed using SPAdes v3.15.4. The resulting scaffolds were annotated using Prokka v1.14.6·. Multilocus sequence typing was performed using mlst v2.23.0 (https://github.com/tseemann/mlst) based on whole-genome sequencing data.

### Non-targeted metabolomics

Extract solution (1,000 μl; 3:1 methanol:water containing isotope-labelled internal standard mixture) was added to the nasal swab sample and the mixture was sonicated for 10 min in an ice-water bath. The samples were incubated for 1 h at −40 °C and centrifuged at 13,800$g$ and 4 °C for 15 min. The resulting supernatant was transferred to a fresh glass vial for analysis. Liquid chromatography with MS/MS (LC–MS/MS) analyses were performed using an ultra-high-performance liquid chromatography (UHPLC) system (Vanquish, Thermo Fisher Scientific) with a UPLC HSS T3 column (2.1 mm × 100 mm; 1.8 μm) coupled to an Orbitrap Exploris 120 mass spectrometer (Orbitrap MS, Thermo Fisher Scientific). The mobile phase consisted of 5 mmol l$^{-1}$ ammonium acetate and 5 mmol l$^{-1}$ acetic acid in water (A) and acetonitrile (B). The auto-sampler temperature was 4 °C and the injection volume was 2 μl. The mass spectrometer was used to acquire MS/MS spectra based on information-dependent acquisition mode in the control of the acquisition software (Xcalibur, Thermo Fisher Scientific). Raw data were converted to the mzXML format using ProteoWizard and processed using the XCMS R package (v3.22.0) for peak detection, extraction, alignment and integration[45]. An in-house MS2 database (BiotreeDB) was used for metabolite annotation. The cutoff for peak annotation was set at 0.3. The identification level of metabolites was annotated based on the method by Alseekh et al.[46], and metabolites identified at the MS2 level with definitive KEGG compound assignments were used for further analysis. The raw peak areas of metabolites were normalized using internal standards. Differentially abundant metabolites were determined based on the variable importance for the projection value calculated by OPLS-DA analysis and FDR using the ropls R package (v1.30.0)[47], with a cutoff of variable importance for the projection > 1 and FDR < 0.05. KEGG pathway enrichment analysis was performed using MetaboAnalyst 6.0 (ref. 48). HAllA-based multivariate analysis (v0.8.20) was conducted to explore associations between species showing differential abundance in MaAsLin2 analysis and metabolites meeting statistical significance criteria[49].

### Animals

Specific-pathogen-free C57BL/6J mice were purchased from GemPharmatech and bred in-house under specific-pathogen-free conditions.

The vendor specifically guaranteed that the mice were free of *S. aureus*. The mice were provided with food and water ad libitum and housed at consistent ambient temperature ($22 \pm 1\,°C$) and humidity ($50 \pm 5\%$) with a 12-h light–dark cycle. The mice were maintained on a γ-gamma-irradiated standard rodent diet (Suzhou Shuangshi Laboratory Animal Feed Technology Co., Ltd). The diet contained corn, wheat bran, soybean meal, fish meal, alfalfa meal, calcium hydrogen phosphate, sodium chloride, vitamin premix and mineral premix, with the following nutritional composition: crude protein, $\geq 200\,g\,kg^{-1}$; crude fat, $\geq 40\,g\,kg^{-1}$; crude ash, $\leq 80\,g\,kg^{-1}$; crude fibre, $\leq 50\,g\,kg^{-1}$ and moisture content, $\leq 100\,g\,kg^{-1}$.

### Animal experiments

**Nasal microbiota transplantation.** Nasal microbiota transplantation was performed in male and female C57BL/6J mice that were about 6 weeks old. Briefly, 10 µl antibiotic cocktail ($0.5\,g\,l^{-1}$ ampicillin, $0.5\,g\,l^{-1}$ metronidazole and $0.25\,g\,l^{-1}$ vancomycin) was instilled daily into each mouse's nostril (total of 20 µl per mouse) for seven days. Nasal microbiota transplantation was performed three days after the last antibiotic instillation. Male mice received nasal transplants from male donors and female mice from female donors (one donor per mouse). Donor nasal microbiota samples were randomly selected from the healthy and depression cohorts, and were not expanded through cultivation before inoculation. Each mouse was inoculated with the microbiome of a different donor. The nasal microbiota sample was dropped into the nostrils (10 µl per nostril; 20 µl total per mouse) of antibiotic-treated mice every day for seven days to improve microbial community stability. Four mice were housed in one cage. Behavioural assessments were performed afterwards. To evaluate the bacterial load, noses were collected at different time points, homogenized in 0.5 ml cold PBS, and homogenates were serially diluted and plated on 5% sheep blood agar plates for enumeration.

DNA was extracted according to the manufacturer's instructions (EZNA soil DNA kit, Omega). The V3–V4 region of the 16S rRNA gene was amplified using the primers 338F (5′-ACTCCTACGGGAGGCAGCAG-3′) and 806R (5′-GGACTACHVGGGTWTCTAAT-3′). The amplicons were then sequenced on the Illumina platform. Sequence analysis was performed using the DADA2 workflow in the QIIME2 software pipeline. The taxonomy of ASV sequences was analysed using the SILVA v138.2 database. Taxonomic α-diversity was estimated using Shannon and Simpson indices. The β-diversity between groups was measured using PCoA based on Bray–Curtis dissimilarity and compared using the PERMANOVA method with the vegan R (v2.6-4) package. Microbiome taxonomic abundance data were analysed using general linear models by MaAsLin2.

**Bacterial culture and nasal colonization.** *S. aureus* strain P24-2 (ST398) or *S. epidermidis* strain P24-1, which were isolated from the nasal microbiome of patient 24 in the depression cohort, were cultured in TSB at $37\,°C$ for 8 h, harvested by centrifugation at $13,000g$ and $4\,°C$ for 2 min, and then resuspended in sterile PBS. Each nostril of male and female C57BL/6J mice (approximately 6 weeks old) received 10 µl of an antibiotic cocktail ($0.5\,g\,l^{-1}$ ampicillin, $0.5\,g\,l^{-1}$ metronidazole and $0.25\,g\,l^{-1}$ vancomycin; 20 µl total per mouse). After three days of daily antibiotic pretreatment, the mice were subjected to treatment with 10 µl bacterial solution in each nostril (total of 20 µl per mouse containing approximately $1 \times 10^7$ CFU) once every two days for seven total treatments. For *S. aureus* strain JSNZ (ST88), mice without antibiotic pretreatment directly received two treatments of 2.5 µl bacterial solution seven days apart (total of 5 µl per mouse containing approximately $1 \times 10^6$ CFU). The mice were euthanized at pre-determined intervals throughout the experimental period. Nose and lung tissues were collected and homogenized in 0.5 ml cold PBS. Caeca and faeces were collected and homogenized in 5 ml cold PBS. For CFU counts, homogenates were serially diluted and plated on 5% sheep blood agar plates, ChromAgar *Staph aureus* selective plates, or selective plates containing $0.5\,mg\,ml^{-1}$ streptomycin (for ST88), which were then incubated at $37\,°C$ for 24 h. *S. aureus* and *S. epidermidis* colonies were identified using MALDI-TOF-MS (Bruker Daltonics).

**Chronic unpredictable mild stress.** Male C57BL/6J mice were exposed to one of the following various low-intensity social/environmental stressors sequentially for a total duration of seven days based on the method by Yu and colleagues[50]. The stressors were: (1) food deprivation for 24 h, (2) overnight illumination for one night, (3) absence of sawdust in the cage for 24 h, (4) moistened sawdust for 24 h, (5) water deprivation for 24 h, (6) physical restraint for 6 h and (7) 45° cage-tilt along the vertical axis for 3 h.

**Hormone replacement therapy.** To eliminate the effects of physiological hormonal fluctuations, a bilateral ovariectomy was performed on female C57BL/6J mice and a bilateral orchidectomy on male C57BL/6J mice (all approximately 6 weeks old). After 14 days of postoperative care and no antibiotic pretreatment, 5 µl JSNZ bacterial solution (containing about $1 \times 10^6$ CFU) was instilled into the right nostrils of the mice. The next day, the mice received hormone replacement therapy using a nasal hydrogel for sustained release of estradiol or testosterone[51]. The female (ovariectomy) mice received 1 µg estradiol in 5 µl in situ nasal hydrogel. The male (orchidectomy) mice received 0.1 mg testosterone in 5 µl in situ nasal hydrogel.

To study the effects of intranasal administration of pure sex hormones, female mice colonized with JSNZ (ST88) received 0.1 µg estradiol or vehicle in 5 µl in situ nasal hydrogel every two days, and CUMS-treated male mice colonized with JSNZ (ST88) received 10 µg testosterone or vehicle in 5 µl in situ nasal hydrogel every two days.

**Nasal hydrogel.** The thermosensitive in situ nasal hydrogel was prepared as follows. Solubilization of estradiol, testosterone or FITC was achieved through hydroxypropyl-β-cyclodextin (HP-β-CD) complexation, sonication (40 kHz, ice bath) in 2% HP-β-CD in PBS (pH 6.5), followed by magnetic stirring ($500g$, $25\,°C$) and purification via centrifugation. PLGA-PEG-PLGA and CS-PEG polymers were dissolved in chilled ultrapure water and acetate buffer (pH 5.0), respectively, and allowed to electrostatically self-assemble (2:1 vol/vol) with drug-loaded HP-β-CD under continuous stirring ($4\,°C$ for 6 h). The gel was reconstituted by blending complexes with 20% chilled poloxamer 407 (1:9 vol/vol)[52], which is a thermosensitive molecule widely used in mucosal delivery. The gelation time was about 30 s at $37\,°C$. In vitro release studies employed dynamic dialysis (molecular weight cutoff, 3.5 kDa) in the synthetic nasal medium SNM3 (pH 6.5)[53] containing 0.5% HP-β-CD, with sustained drug release monitored over 48 h using LC–MS/MS. Anaesthetized mice received unilateral intranasal administration of 5 µl thermosensitive hydrogel (containing about 10 µg FITC) or control liquid formulation (FITC dissolved in 2% HP-β-CD/PBS, pH 6.5). The animals were euthanized at pre-determined intervals post administration, with major organs excised for fluorescent imaging using an IVIS in vivo imaging system.

**Behavioural tests.** Behavioural tests were performed based on the previously reported methods[50,54].

*Tail suspension test.* The TST test was performed by wrapping the tail of each mouse with tape, with the tip sticking out by about 1 cm, and then suspending the mouse head-downwards at a height of 15 cm above the floor. The animals were videotaped from the front for 6 min. The total duration of immobility within the last 4 min was analysed using the VisuTrack software (Shanghai XinRuan Information Technology Co., Ltd.). Duration of immobility was defined as the time when animals did not seem to struggle.

*Forced swimming test.* The FST test was performed by placing individual mice in a water-filled cylinder (diameter of 12 cm and a height of 25 cm; water temperature of 25 ± 1 °C). The animals were videotaped from the front for 6 min. The time of immobility during the last 4 min was counted using the VisuTrack software. It was defined as the time when the animals remained floating or motionless with the only observed movements being those necessary for keeping their balance in the water.

*Open field test.* The OFT test was performed by placing the mice in the centre of an arena (50 cm × 50 cm × 40 cm). The animals were videotaped from above for 6 min. The time spent in the central area during the last 4 min was counted using the VisuTrack software. Additional data related to all OFT experiments are in Supplementary Data 2.

**Cytokine concentration assessment.** Mice nose and lung tissues were collected and homogenized in 0.5 ml RIPA solution (containing 0.1 mM phenylmethylsulfonyl fluoride). After centrifugation at 4 °C and 10,000$g$ for 15 min, the supernatants were collected and assessed using a bicinchoninic acid assay for protein quantification (Sangon Biotech). Concentrations of interleukin-6 and 1β, tumour necrosis factor-α and CXCL1 were determined using an enzyme-linked immunosorbent assay kit (CUSABIO).

**Histopathology.** Mouse noses and lungs were fixed in 4% paraformaldehyde at 4 °C for three days. The noses were then decalcified in 0.12 mol l$^{-1}$ EDTA solution (pH 7.4) for 7–14 days at room temperature. Hematoxylin and eosin staining was then performed following standard procedures.

Nasal tissues were collected four days after the final colonization (before behavioural testing) and fixed in 4% paraformaldehyde at 4 °C for 72 h. Following EDTA decalcification (0.12 M, pH 7.4; 7–14 days with solution replacement every 48 h), the tissues were dehydrated through graded ethanol, cleared in xylene and paraffin-embedded for 5-µm sectioning. Immunofluorescence protocols included sequential dewaxing with eco-friendly agents, EDTA-based heat-induced epitope retrieval (98 °C for 20 min), blocking with 10% donkey serum and incubation with primary antibodies—anti-IBA1 (guinea pig polyclonal antibody; Oasis Biofarm, OB-PGP049) or anti-OMP (rabbit polyclonal antibody; Bioss, bs-19568R) at a dilution of 1:100—and the fluorescent secondary antibodies Alexa Fluor 594 goat anti-guinea pig IgG (Oasis Biofarm, G-GP594; 1:200) and Alexa Fluor 488 donkey anti-rabbit IgG (Thermo Fisher Scientific, A21206; 1:400). Nuclei were counterstained with 4′,6-diamidino-2-phenylindole (1:500), followed by mounting with antifade medium and fluorescence microscopy analysis.

Lung tissues were harvested four days after the final colonization (before behavioural testing) and fixed in 4% paraformaldehyde at 4 °C for 72 h. Paraffin sections underwent sequential dewaxing with eco-friendly agents and graded ethanol, followed by antigen retrieval using either EDTA-based pressure cooking (1,200 W, 1.5 min boiling) or microwave-mediated retrieval (medium power, 12 min). Endogenous peroxidase activity was quenched with 3% $H_2O_2$ before blocking with 10% goat serum. Anti-LY6G primary antibody (rabbit monoclonal antibody; Abcam, ab238132, EPR22909-135; 1:500) was applied overnight at 4 °C, followed by corresponding horseradish peroxidase-conjugated secondary antibody (goat anti-rabbit IgG; Abcam, ab205718; 1:2,000) at 37 °C for 45 min. Immunoreactivity was visualized using 3,3′-diaminobenzidine chromogen with reaction monitoring, counterstained with haematoxylin for 1 min, and differentiated with acid-alcohol. The sections were dehydrated, cleared and mounted with eco-friendly medium for bright-field microscopy analysis.

**Sickness scores.** Sickness scores were determined according to Huet et al.[19], where eight parameters (fur aspect, activity, posture, behaviour, respiration, chest sounds, eyes and body weight) were given a score between one, representing the minimum (healthy), and four, representing the maximum. Total scores thus ranged from eight for entirely healthy mice to 32.

**Neurotransmitter detection.** Midbrain tissues were collected, weighed and homogenized in 80 µl extract solvent (0.1% formic acid in acetonitrile; precooled to −20 °C)/20 µl water. The samples were kept at −20 °C overnight and then centrifuged at 13,800$g$ and 4 °C for 15 min. The 80-µl supernatant samples were incubated with 40 µl of 100 mmol l$^{-1}$ sodium carbonate solution and 40 µl 2% benzoyl chloride in acetonitrile solution for 30 min. The samples were then centrifuged at 13,800$g$ and 4 °C for 15 min after the addition of 10 µl internal standard. Next, 40-µl aliquots of the supernatants were added to 20 µl water and transferred to an auto-sampler vial for UHPLC–MS/MS analysis (Waters ACQUITY Premier or SCIEX Triple Quad 6500+ MS). The mobile phase A was 0.1% formic acid and 1 mmol l$^{-1}$ ammonium acetate in water, and the mobile phase B was acetonitrile. SCIEX Analyst Work Station Software (version 1.6.3) was employed for multiple-reaction-monitoring data processing.

### Anaesthesia
Mice received nasal antibiotics, microbiome transplants or bacteria and were euthanized by cervical dislocation under anaesthesia using isoflurane inhalation.

### Serum cytokine and thyroid hormone detection
Serum cytokine concentrations were measured using a cytokine detection kit (BD Biosciences). Serum concentrations of thyroid-stimulating hormone (TSH), free triiodothyronine (FT3) and free thyroxine (FT4) were determined using a Roche Cobas fully automated electrochemiluminescence immunoassay system.

### Steroid hormone detection
Steroid hormones were detected using LC–MS/MS. For serum or supernatant samples of bacterial late-exponential-phase cultures (TSB), 200 µl of the samples was mixed with 20 µl internal standard solution and 400 µl methanol, followed by vortexing for 30 s and centrifugation for 10 min at 13,800$g$ and 4 °C. Next, 600 µl water was added to 500-µl supernatant aliquots, followed by vortexing for 30 s and centrifugation under the same conditions. A 950-µl aliquot of the obtained supernatant was further purified with solid phase extraction cartridges (Agela). The cartridges were washed with 200 µl methanol and then equilibrated with 200 µl water. After sample application loading, the cartridges were washed with 200 µl of 10% acetonitrile in water (vol/vol) and 200 µl hexane. The cartridge was then rinsed with 40 µl of 90% acetonitrile in water (vol/vol) and 60 µl water was added to the eluent. All samples were shaken for 3 min, after which the mixed solution was subjected to UHPLC–MS/MS analysis (Waters ACQUITY UPLC/Xevo TQ-S MS). The mobile phase A was 0.5 mmol l$^{-1}$ ammonium fluoride in water, and the mobile phase B was methanol. Waters MassLynx V4.1 was employed for multiple-reaction-monitoring data acquisition and processing.

For tissue samples, nose and midbrain tissues were collected, weighed and homogenized in 200 µl water. After the addition of 20 µl internal standard solution and 400 µl methanol, the samples were processed as described for the serum samples.

### Degradation of estradiol and testosterone
Nasal bacteria were cultured in TSB medium containing 100 ng ml$^{-1}$ estradiol or testosterone at 37 °C. After 24 h of cultivation, the supernatant was collected, and testosterone and estradiol concentrations were determined using LC–MS/MS. For time-dependent degradation of testosterone and estradiol by *S. aureus* supernatant, the same conditions were used and the sampling time points were set to 0, 24, 48 and 60 h.

## RNA-sequencing

Nasal mucosal tissues and midbrain samples were carefully isolated using microscopy forceps. Total RNA was extracted using TRIzol reagent according the manufacturer's instructions (Invitrogen) and genomic DNA was removed using DNase I (TaKaRa). RNA-seq transcriptome libraries were prepared following RNA preparation with a TruSeqTM RNA sample preparation kit (Illumina). Paired-end libraries were sequenced by Illumina NovaSeq 6000 sequencing (150 bp × 2; Shanghai BIOZERON Co., Ltd). The raw paired-end reads were trimmed and quality-controlled using Trimmomatic v0.39. Next, clean reads were separately aligned to the reference genome with orientation mode using the hisat2 v2.2.1 software. FeatureCounts v2.0.3 was used to count each gene read. Differentially expressed genes were determined using edgeR v3.42.4 based on FDR and fold change, with a cutoff of FDR < 0.05 and $|\log_2(\text{fold change})| > 1$. KEGG pathway enrichment analysis was performed using clusterProfiler v4.8.2. For bacteria, RNA was prepared from bacteria cultured to logarithmic growth phase and RNA-seq was performed as described above.

## Quantitative real-time PCR

Total RNA was extracted from the midbrain using an RNeasy kit (Qiagen) according to the manufacturer's instructions. The extracted RNA was reverse-transcribed into cDNA using a PrimeScript RT reagent kit with gDNA Eraser (TaKaRa). Real-time PCR was performed using the Hieff UNICON Universal Blue qPCR SYBR Green Master Mix (Yeasen) on a 7500 real-time PCR system (Applied Biosystems). Gene expression was calculated using the $2^{-\Delta\Delta CT}$ method.

## SDR family identification using hidden Markov models

The SDR superfamily encompasses multiple Pfam entries: PF00106, PF01073, PF01370, PF05368, PF08659 and PF1356 (ref. 25). Hidden Markov models of these Pfam profiles were extracted from the Pfam database and hmmer v3.0 was used to scan protein sequences of the P24-2 (ST398) *S. aureus* genome for proteins belonging to these Pfam protein families. Proteins with an *E*-value of less than $1 \times 10^{-5}$ were recorded as positive hits. The sequences of putative SDR superfamily proteins were compared using blastp with the Uniprot database to exclude candidates with known other functions.

## Heterologous expression of putative SDR proteins in *E. coli*

The plasmid pET28a was used as an overexpression vector to express SDR proteins from *S. aureus* in *E. coli*. Genomic DNA of *S. aureus* P24-2 was used as a template to amplify putative SDR genes via PCR (primers in Supplementary Table 5). Amplified genes were cloned into the pET28a plasmids and the resulting plasmids were transformed into *E. coli* BL21 (DE3). The recombinant *E. coli* strains were cultured in 3 ml Luria–Bertani medium containing 100 ng ml⁻¹ testosterone or estradiol at 37 °C. When the optical density at 600 nm reached a value of 0.6–0.8, 0.5 mM isopropyl β-D-1-thiogalactopyranoside was added to the medium to induce protein expression. After 24 h of cultivation, the supernatant was collected, and the concentrations of testosterone and estradiol and their degradation products androstenedione and estrone, respectively, were determined using LC–MS/MS.

## Construction of *S. aureus hsd12*-deletion mutant

The *hsd12*-deletion mutants were generated in the strains *S. aureus* P24-2 (ST398) and JSNZ (ST88) using the pKOR1 allelic replacement strategy. Briefly, sequences of approximately 1 kb flanking the *hsd12* gene were amplified by PCR. The recombinant plasmids pKOR1-HSD12 (for deletions in ST88 and ST398) were constructed using a ligation-independent cloning method and transformed into *E. coli* Top10 cells as the cloning host. The plasmids purified from *E. coli* Top10 cells were electroporated into *S. aureus* P24-2 and JSNZ using *S. aureus* RN4220 as an intermediary host. Allelic replacement was induced by temperature shift. The pKOR1 transformants were

selected by plating on tryptic soy agar containing 10 µg ml⁻¹ chloramphenicol and incubation at 30 °C, followed by incubation in TSB containing 10 µg ml⁻¹ chloramphenicol at 43 °C to allow plasmid integration into the chromosome. Non-plasmid-carrying mutants were selected by plating on tryptic soy agar containing 1 µg ml⁻¹ anhydrotetracycline. Successful deletion of *hsd12* was verified by PCR. To verify the growth behaviour of the *hsd12*-deletion strain (SAΔ*hsd*), the same amount ($1 \times 10^6$ CFU ml⁻¹) of mid-logarithmic phase *S. aureus* P24-2 and its *hsd12*-deletion mutant were cultured in 200 µl of fresh TSB medium in 96-well plates. The plates were incubated at 37 °C in a BioTek Synergy H1 multi-mode microplate reader with continuous shaking for 14 h. The optical density at 600 nm was measured every 15 min.

## Statistics and reproducibility

GraphPad Prism version 10.2.0 for Macintosh was used for simple statistical analyses including comparisons of two or more groups and correlation analyses. Two-group comparisons were performed with unpaired two-sided Student's *t*-tests or Mann–Whitney tests. Comparisons of three or more groups were performed using a one-way or two-sided ANOVA, or a Kruskal–Wallis test, as appropriate, depending on the assessments of normal distribution using the Shapiro–Wilk test, with Tukey's and Dunn's post-tests, respectively. Spearman's correlation was used to assess the association between two quantitative variables, interpreted according to Akoglu et al.[16] and Chan et al.[55], and Fisher's exact tests or $\chi^2$ tests were used to assess contingency, as indicated. Principal coordinate analyses were performed and evaluated using the vegan R (v2.6-4) package. Further tests are described in their specific method sections. Each independent experiment was performed with at least three biological replicates. Values were expressed as the mean ± s.d., unless otherwise indicated in the figure legend.

Where indicated, randomization was performed using the randomization functions of Microsoft Excel 2019, which randomly assigns a number between zero and one to every sample, after which the numbers are sorted in ascending order and the top samples of the desired *n* selected. Animals were randomly assigned to experimental groups using computer-generated random numbers following the completion of an acclimation period. Experimental conditions (including bacterial administration sequence and behavioural testing order) were randomized across subjects.

No statistical method was used to pre-determine sample size. The sample sizes were similar to those reported in previous publications[9,56]. No data were excluded from the analyses except when using the described data filtering processes. The investigators were not blinded to allocation during experiments and outcome assessment, except for histological analysis, where slides were examined independently by a histopathologist who was blinded to the treatment.

## Availability of materials

All unique biological materials are available from M.L. (rjlimin@shsmu.edu.cn) on request.

## Reporting summary

Further information on research design is available in the Nature Portfolio Reporting Summary linked to this article.

## Data availability

Raw microbiome sequencing and transcriptome data have been deposited in the NCBI's Sequencing Read Archive (SRA) database under Bioproject number PRJNA1138490. Raw metabolomics data have been deposited in the MetaboLights database under accession number MTBLS10742. The nucleotide sequences of putative SDR proteins identified in this study have been deposited in NCBI's GenBank database under accession numbers PQ067567 to PQ067586 and PQ106784 to PQ106786. Source data are provided with this paper.

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

## Acknowledgements

This work was supported by the National Key Research and Development Program (grant numbers 2023YFC2306200, W2411070 and 2022YFC2603800 to M.L.), the National Natural Science Foundation of China (grant numbers 82172325 to M.L., 82302595 to P.M. and 82472288 to Y.W.), the Action Plan for Strengthening Public Health System in Shanghai (grant number GWVI-11.1-11 to M.L.), the Shanghai Pujiang Program (grant number 23PJD052 to P.M.) and the Intramural Research Program of the National Institute of Allergy and Infectious Diseases (project number AI000904 to M.O.). The contributions of the NIH author were made as part of his official duties as an NIH federal employee, are in compliance with agency policy requirements and are considered Works of the US Government. However, the findings and conclusions presented in this paper are those of the authors and do not necessarily reflect the views of the NIH or the US Department of Health and Human Services.

## Author contributions

K.N., G.X., H.L., X.S., P.M., C.J., M.H. and Y.P. obtained nasal swabs and serum samples. G.X., L.H., Y.J., Z.Y., T.C., K.X. and N.Z. cultured bacteria and identified bacterial species. Q.L. and G.X. performed 16S rRNA sequencing. Z.S. and G.X. performed whole-genome sequencing. G.X. and Y.S. performed LC–MS/MS tests. G.X. and Y.W. performed animal studies and RNA-Seq. G.X. and H.L. performed behaviour tests. G.X. constructed heterologous expression *E. coli* and *S. aureus* gene-deletion strains. Y.L., J.H., M.O. and M.L. planned, and M.L. supervised, experiments. M.O., G.X. and M.L. analysed the data. M.O. wrote the paper.

## Competing interests

The authors declare no competing interests.

## Additional information

**Extended data** is available for this paper at https://doi.org/10.1038/s41564-025-02120-6.

**Correspondence and requests for materials** should be addressed to Yanli Luo, Ji Hu, Michael Otto or Min Li.

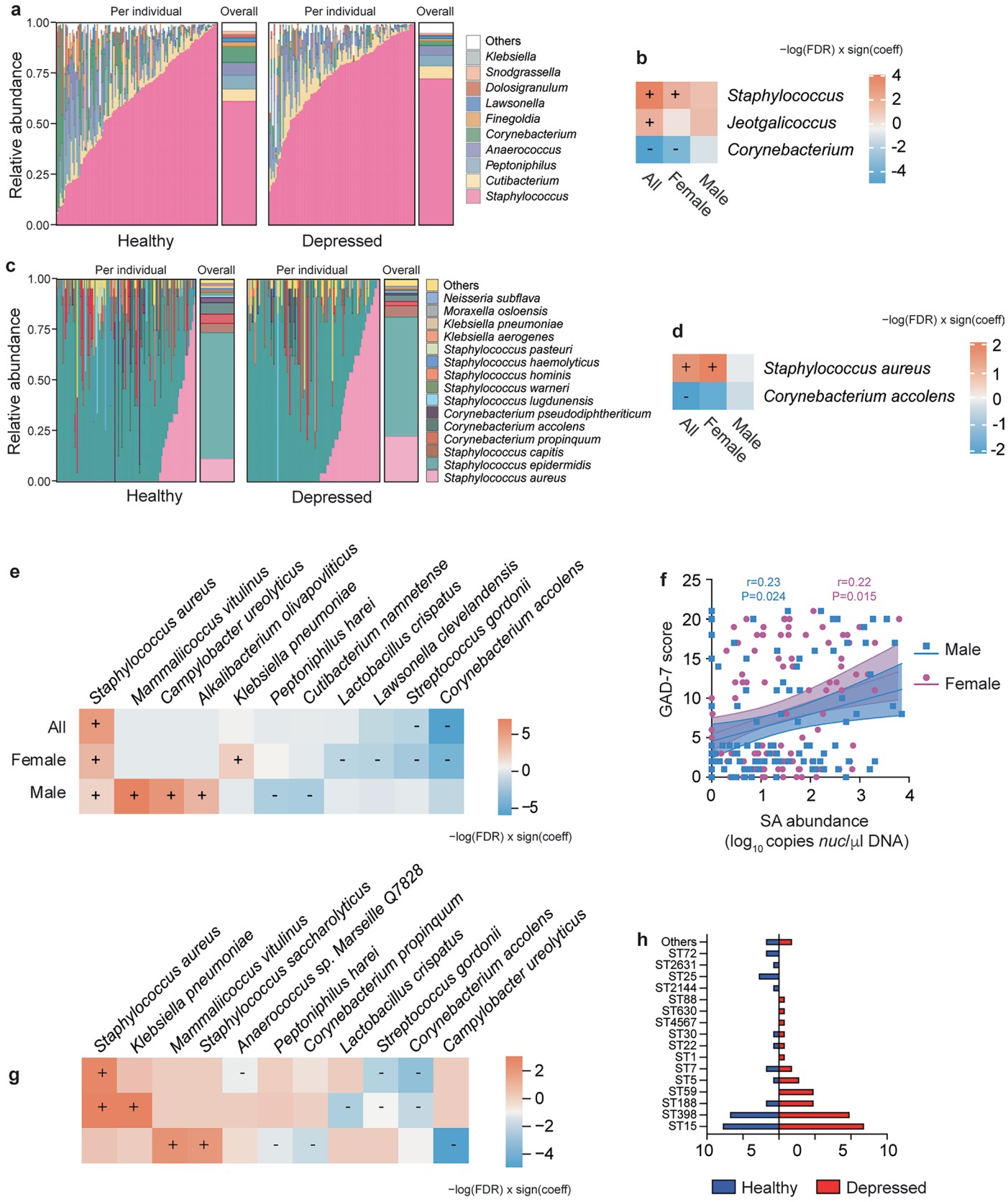

**Extended Data Fig. 1 | See next page for caption.**

**Extended Data Fig. 1 | Nasal microbiome analysis in healthy versus depressed cohorts – additional data. a**, Per-individual and overall relative abundances of the most abundant genera in the nasal microbiomes of healthy ($n$ = 118) and depressed ($n$ = 100) cohorts as per 16S rRNA sequencing. **b**, Corresponding multivariate analysis adjusted for age, sex and batch covariates. **c**, Per-individual and overall abundances of the most abundant species as per culture-based analysis. **d**, Corresponding multivariate analysis adjusted for age, sex and batch covariates. **e**, Multivariate analysis of bacterial abundance and PHQ-9 scores adjusted for age, sex, BMI, education, income, family relationships, adverse childhood experiences, available supportive friends, and batch covariates. **f,g**, Multivariate analysis and correlation plot of bacterial and *S. aureus* (SA) abundance, respectively, and GAD-7 anxiety scores using the same analytical approach as applied to the PHQ-9. Statistical analysis is by Spearman correlation (**f**). The lines show simple linear regression. Shaded areas show 95% confidence intervals. **h**, Sequence-type distribution of randomly selected isolates (one per colonized individual; $n$ = 31, healthy cohort; $n$ = 45, depressed cohort). In multivariate plots (**b,d,e,g**), fields marked with "+" and "-" signs represent significant associations (FDR < 0.05).

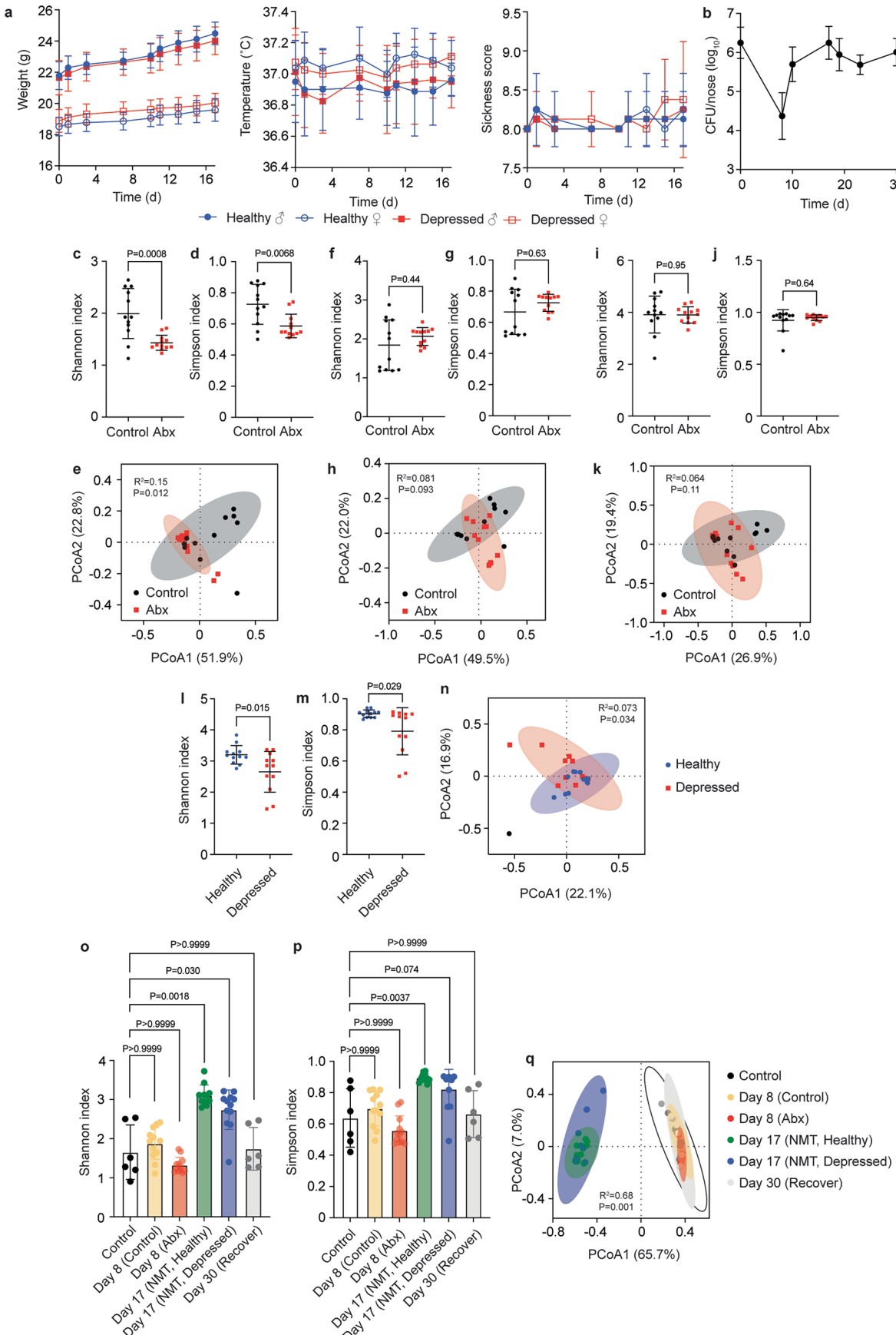

Extended Data Fig. 2 | See next page for caption.

**Extended Data Fig. 2 | Nasal transplant experiment–additional data 1.**
**a**, Weight, temperature and sickness scores over the time of the experiment.
**b**, CFU over the course of the experiment. **c,d**, α-Diversity analyses of the nasal microbiomes of antibiotic-treated (Abx) versus control mice that did not receive antibiotic, measured at day 8 (see Fig. 2a). **e**, β-Diversity analysis of the nasal microbiomes of antibiotic-treated (Abx) versus control mice that did not receive antibiotic, measured at day 8 (see Fig. 2a). **f–h**, Corresponding analyses for the lung microbiomes. **i–k**, Corresponding analyses for the intestinal (fecal)
microbiomes. **l–n**, α- And β-diversity analyses of mouse nasal microbiomes receiving nasal transplants from healthy versus depressed people, measured at day 17 (see Fig. 2a). **o–q**, α- And β-diversity analyses of mouse nasal microbiomes over the course of the experiment in comparison. $n = 12$/group (**c–n**); $n = 6$/group (**a,o–q**); $n = 12$ (**b**). Statistical analyses are by unpaired, two-sided $t$-tests (**c,i,l**), two-sided Mann–Whitney tests (**e–g,j,m**), two-sided Kruskal–Wallis tests versus control (**o,p**) and PERMANOVA (**d,h,k,n**). Shaded areas show 95% confidence intervals (**d,h,k,n,q**). All error bars show the mean ± standard deviation (SD).

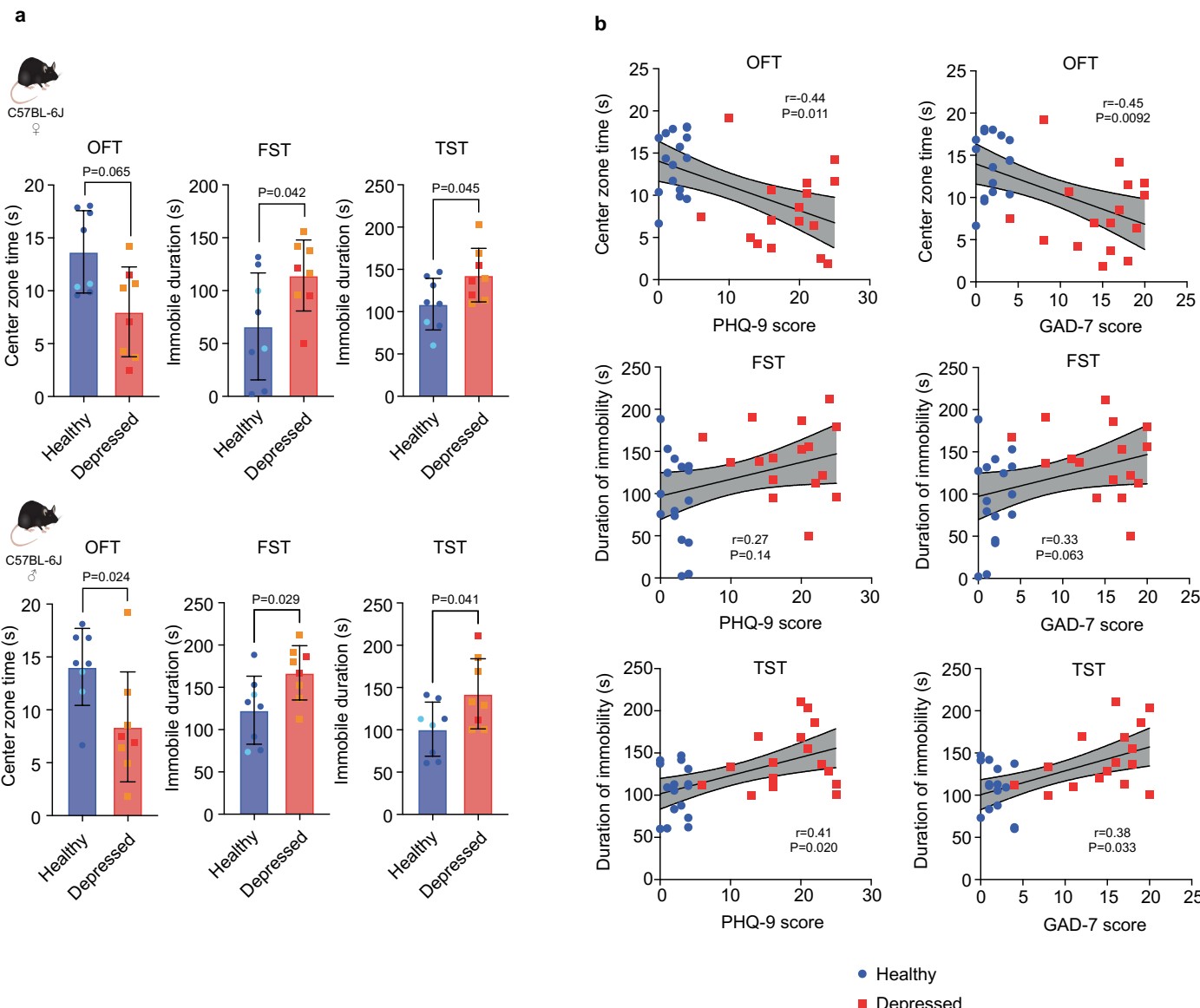

● Healthy
■ Depressed

**Extended Data Fig. 3 | Nasal transplant experiment–additional data 2.**
**a**, Sex-specific behavioural analyses. Male mice received nasal transplants
from male and female from female donors. *n* = 8/group. Statistical analysis is by
unpaired, two-tailed *t*-tests except for female OFT, two-sided Mann–Whitney
test. Error bars show the mean ± standard deviation (SD). **b**, Correlation between

anxiety and depression scores in humans with anxiety and depressive-like
behaviour tests in nasal transplant-recipient mice. *n* = 16/group (*n* = 8 male,
*n* = 8 female). Statistical analysis is by Spearman correlation. Lines show simple
linear regression. Shaded areas show 95% confidence intervals. Credit: mouse
illustrations in **a**, creazilla under a Creative Commons license CC0 1.0.

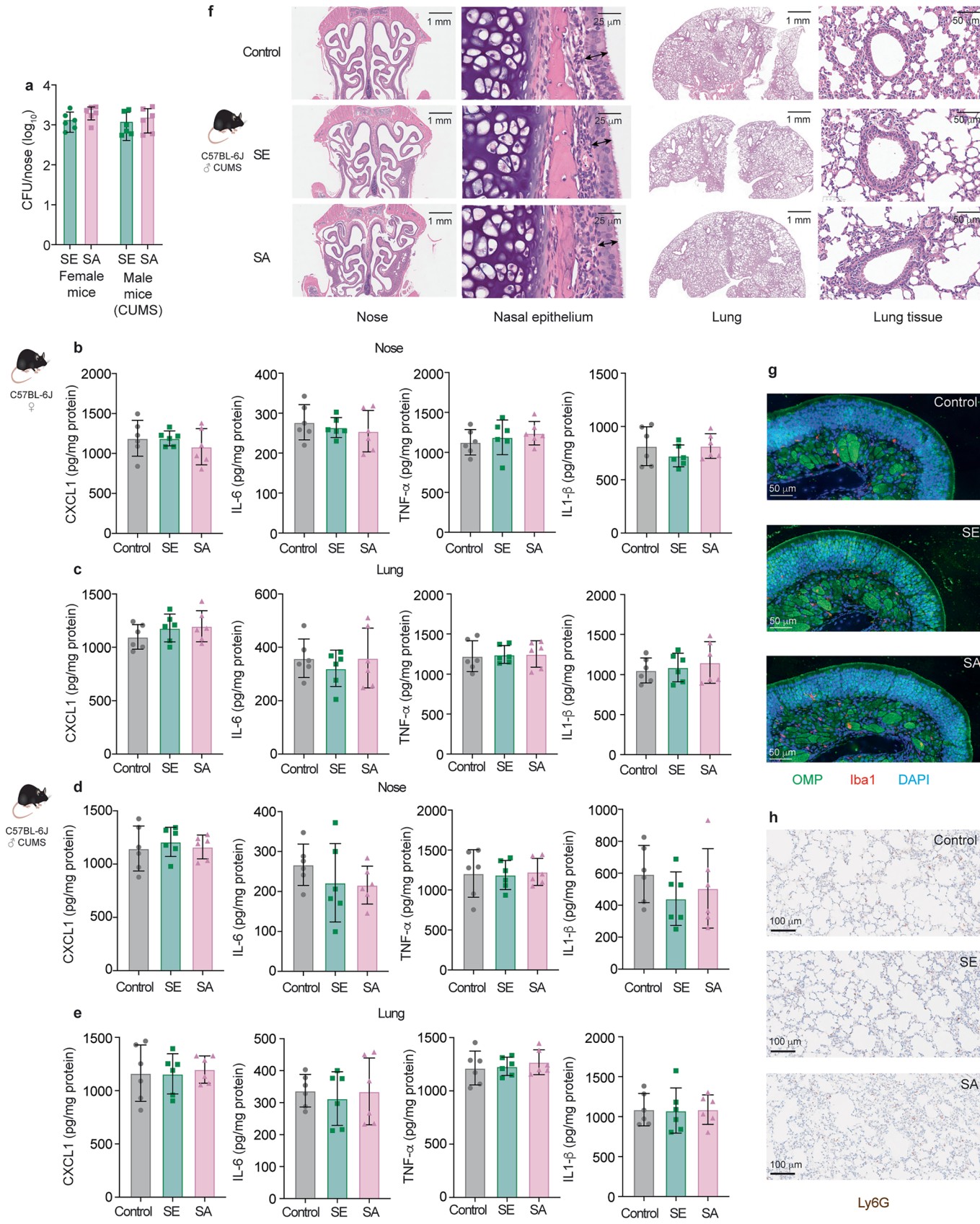

**Extended Data Fig. 4 | See next page for caption.**

**Extended Data Fig. 4 | Mouse colonization experiment–additional data 1.**
**a**, Colonizing CFU in the nose at the time of euthanasia (day 21). $n$ = 6/group.
**b,c**, Nose and lung cytokine concentrations in *S. aureus* (SA) nasally colonized,
*S. epidermidis* (SE) nasally colonized, and control female mice at the time of
euthanasia. $n$ = 6/group. **d,e**, Nose and lung cytokine concentrations in *S. aureus*
nasally colonized, *S. epidermidis* nasally colonized, and control male (CUMS-
treated) mice at the time of euthanasia. $n$ = 6/group. **f**, Nose and lung histology
in *S. aureus* nasally colonized, *S. epidermidis* nasally colonized, and control male

(CUMS-treated) mice at the time of euthanasia. **g**, Histological analysis of
OMP and Iba1 expression in the olfactory epithelium at the time of euthanasia
(day 21). **h**, Ly6G expression in the olfactory epithelium at the time of euthanasia
(day 21). Statistical analysis is by two-sided one-way ANOVA with Tukey's
post-tests (**b**–**e**) and two-sided, unpaired $t$-tests (**a**). There were no significant
differences. All error bars show the mean ± standard deviation (SD). Credit:
mouse illustrations in **b,d,f**, creazilla under a Creative Commons license CC0 1.0.

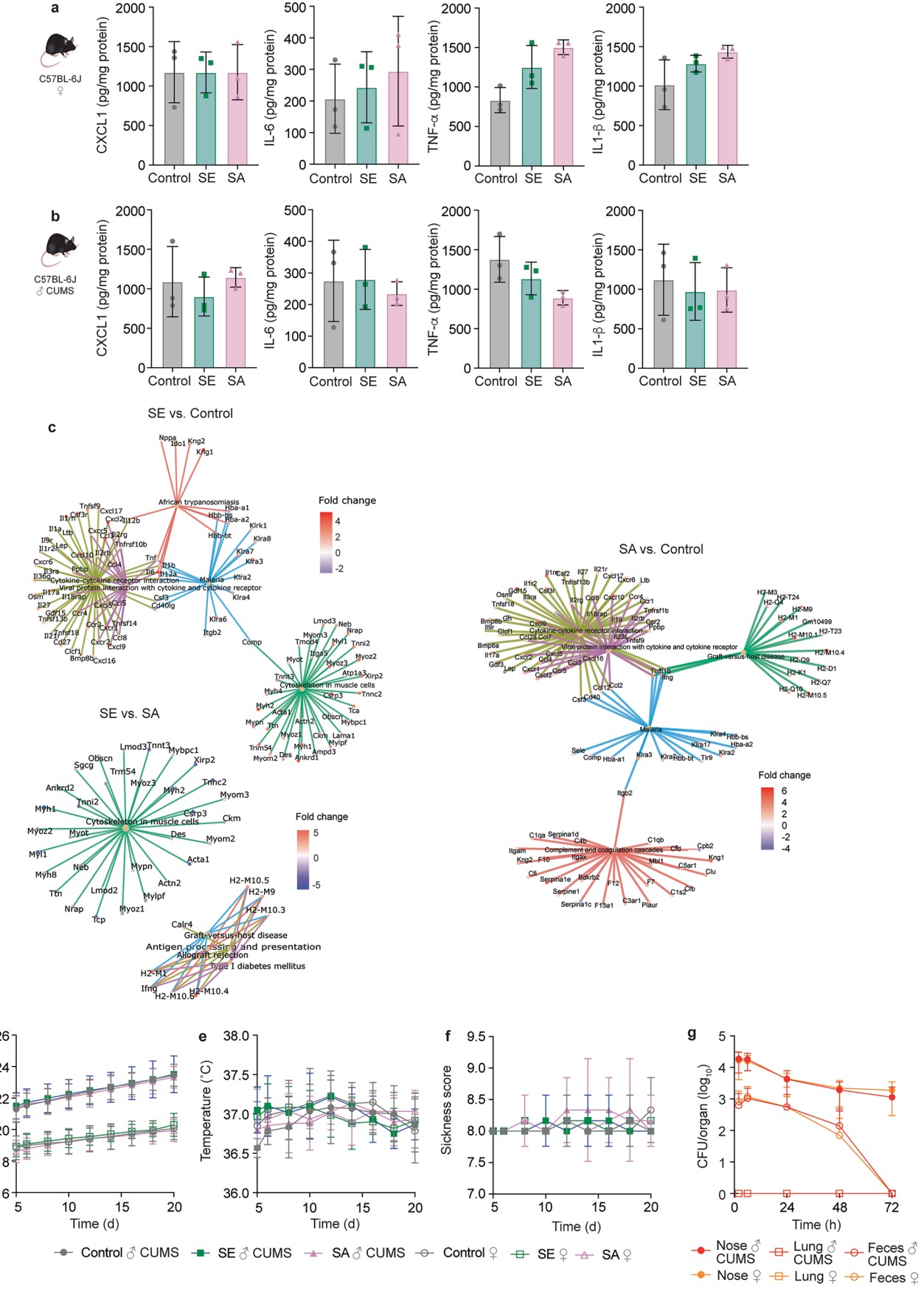

**Extended Data Fig. 5 | See next page for caption.**

**Extended Data Fig. 5 | Mouse colonization experiment–additional data 2.**
**a**,**b**, Lung cytokine concentrations 6 h post the last bacterial application (day 18) in female (**a**) and CUMS-treated male mice (**b**). $n$ = 3/group. Statistical analysis is by two-sided 1-way ANOVAs with Tukey's post-tests. There were no significant differences. **c**, Cnet plots of transcriptomic changes (5 top enriched KEGG pathways) in the nasal mucosa 3 h post the last bacterial application (day 18).

$n$ = 3/group (male CUMS-treated mice). **d**–**f**, Weight, temperature and sickness scores over the time of the experiment. $n$ = 6/group. **g**, Bacterial loads in the noses, lungs, and faeces measured starting 2 h past the time of the last bacterial application. $n$ = 6/group. All error bars show the mean ± standard deviation (SD). SA, *S. aureus*; SE, *S. epidermidis*. Credit: mouse illustrations in **a**,**b**, creazilla under a Creative Commons license CC0 1.0.

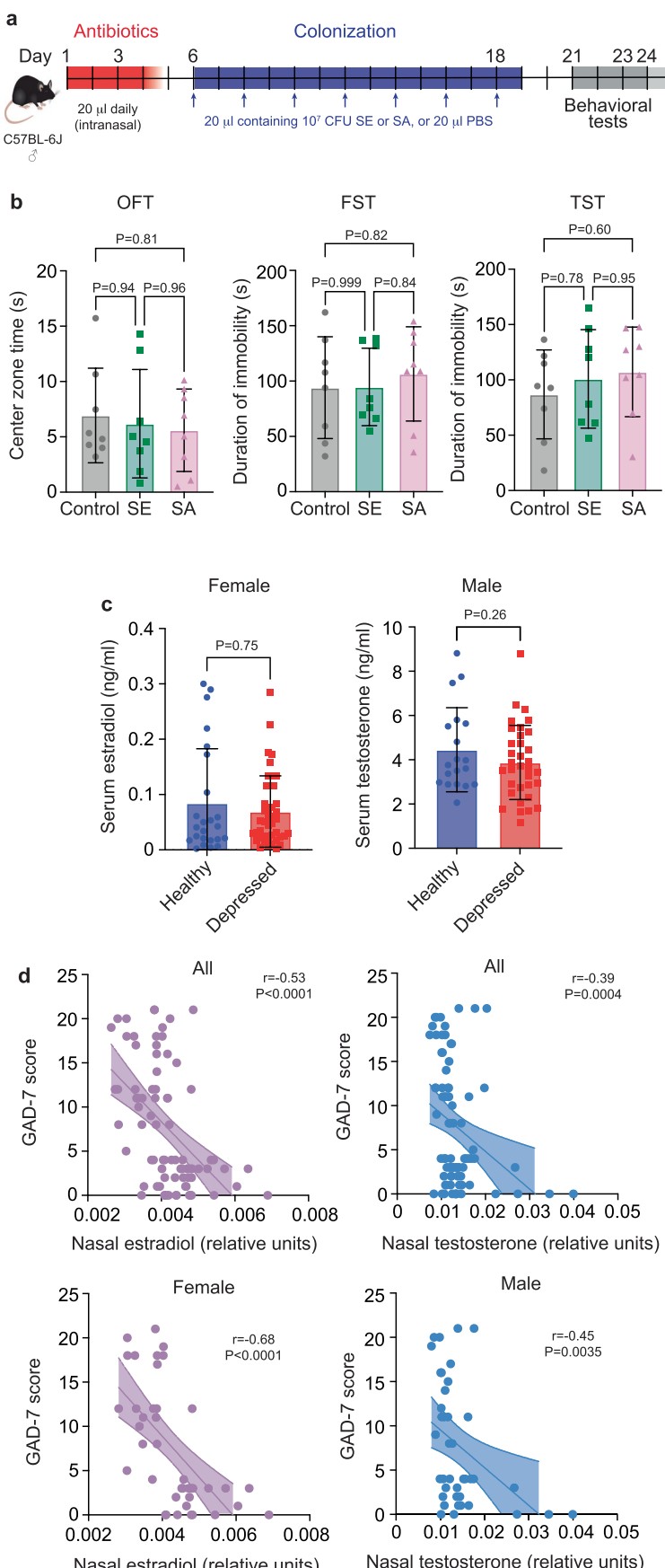

**Extended Data Fig. 6 | See next page for caption.**

**Extended Data Fig. 6 | Induction of anxiety and depression-like behaviour in male mice and metabolome and sex hormone measurement in healthy and depressed cohorts–additional information. a**, Set-up of the experiment (as in Fig. 2d, but using male mice). **b**, Anxiety-like behaviour of mice in OFT and depression-like behaviour in FST and TST tests. $n$ = 8/group. SA, *S. aureus*; SE, *S. epidermidis*. **c**, Serum estradiol and testosterone levels in female and male cohorts, respectively. $n$ = 19 (male, healthy), $n$ = 34 (male, depressed), $n$ = 24 (female, healthy), $n$ = 42 (female, depressed). Serum levels were determined in all individuals who consented to having blood drawn. **d**, Correlation of nasal testosterone or estradiol levels with GAD-7 anxiety scores. Statistical analysis is by two-sided one-way ANOVA with Tukey's post-tests (**b**), two-sided Mann–Whitney tests (**c**: serum estradiol), two-sided, unpaired $t$-test (**c**: serum testosterone) and Spearman correlation (**d**). Error bars show the mean ± standard deviation (SD) (**b**,**c**). Lines show simple linear regression and shaded areas show 95% confidence intervals (**d**). Credit: mouse illustrations in **a**,**b**, creazilla under a Creative Commons license CC0 1.0.

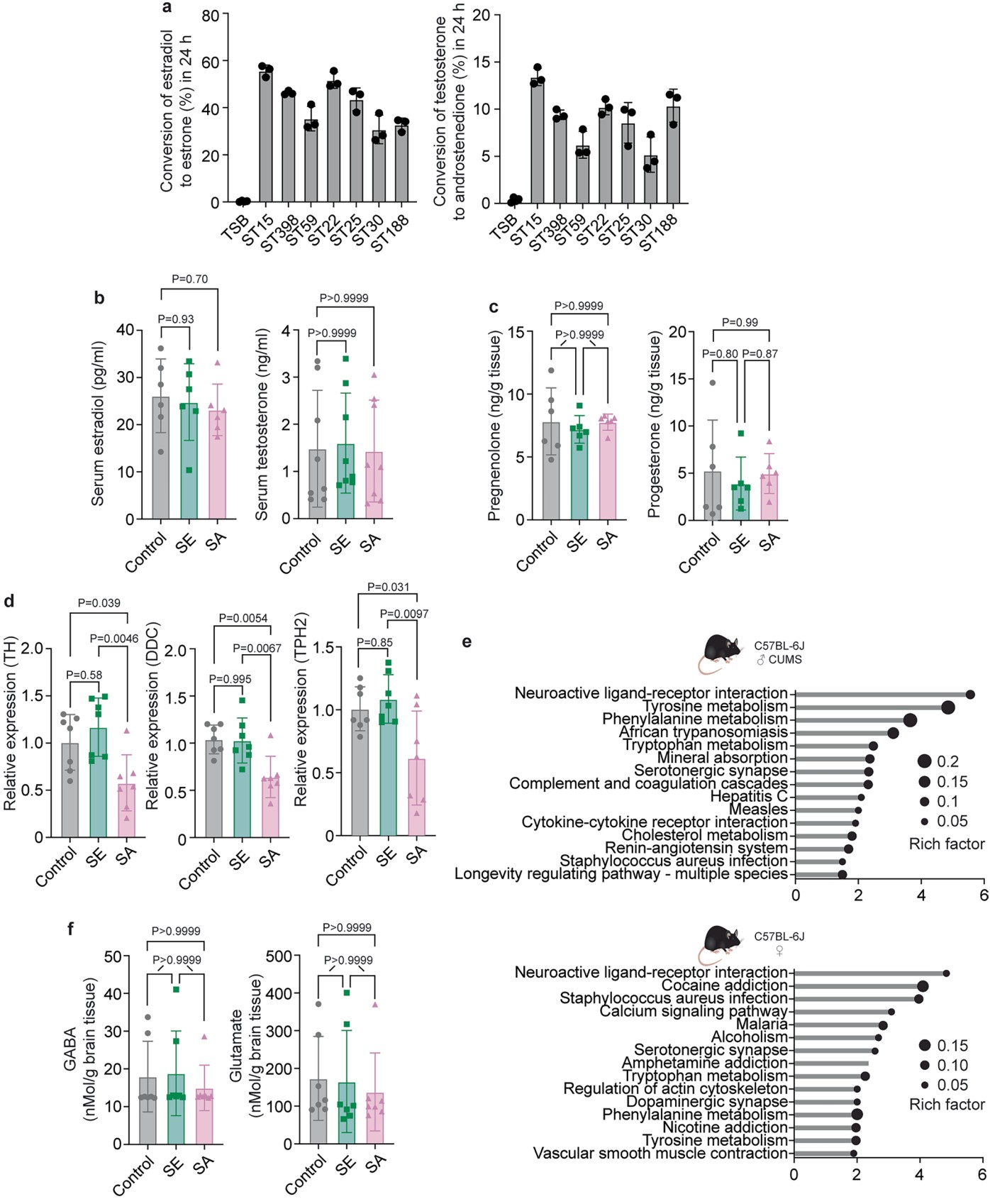

**Extended Data Fig. 7 | See next page for caption.**

**Extended Data Fig. 7 | Nasal *S. aureus* degrades sex hormones and increases dopamine and serotonin levels in the brain – additional information.**
**a**, Conversion of estradiol and testosterone to estrone and androstenedione, respectively, by randomly selected isolates of main isolate STs. $n = 3$/group.
**b**, Serum estradiol and testosterone levels in mice colonized with *S. aureus* (SA) or *S. epidermidis* (SE), or in controls. $n = 6–8$/group ($n = 8$, CUMS-treated male; $n = 6$, female). **c**, Pregnenolone and progesterone midbrain levels. $n = 6$/group ($n = 3$, CUMS-treated male; $n = 3$, female). **d**, Expression of serotonin and dopamine biosynthesis genes in the midbrain of mice colonized with

*S. aureus* or *S. epidermidis*, or in controls. $n = 7$/group ($n = 4$, CUMS-treated male; $n = 3$, female). **e**, Results of KEGG pathway enrichment. **f**, GABA and glutamate midbrain concentrations in *S. aureus*-colonized, *S. epidermidis*-colonized, and control mice. $n = 7$/group ($n = 4$, CUMS-treated male; $n = 3$, female). Statistical analysis is by two-sided one-way ANOVAs or Kruskal–Wallis tests, depending on normality of distribution in the groups, and Tukey's and Dunn's post-tests, respectively (**b**–**d**,**f**). All error bars show the mean ± standard deviation (SD). Credit: mouse illustrations in **e**, creazilla under a Creative Commons license CC0 1.0.

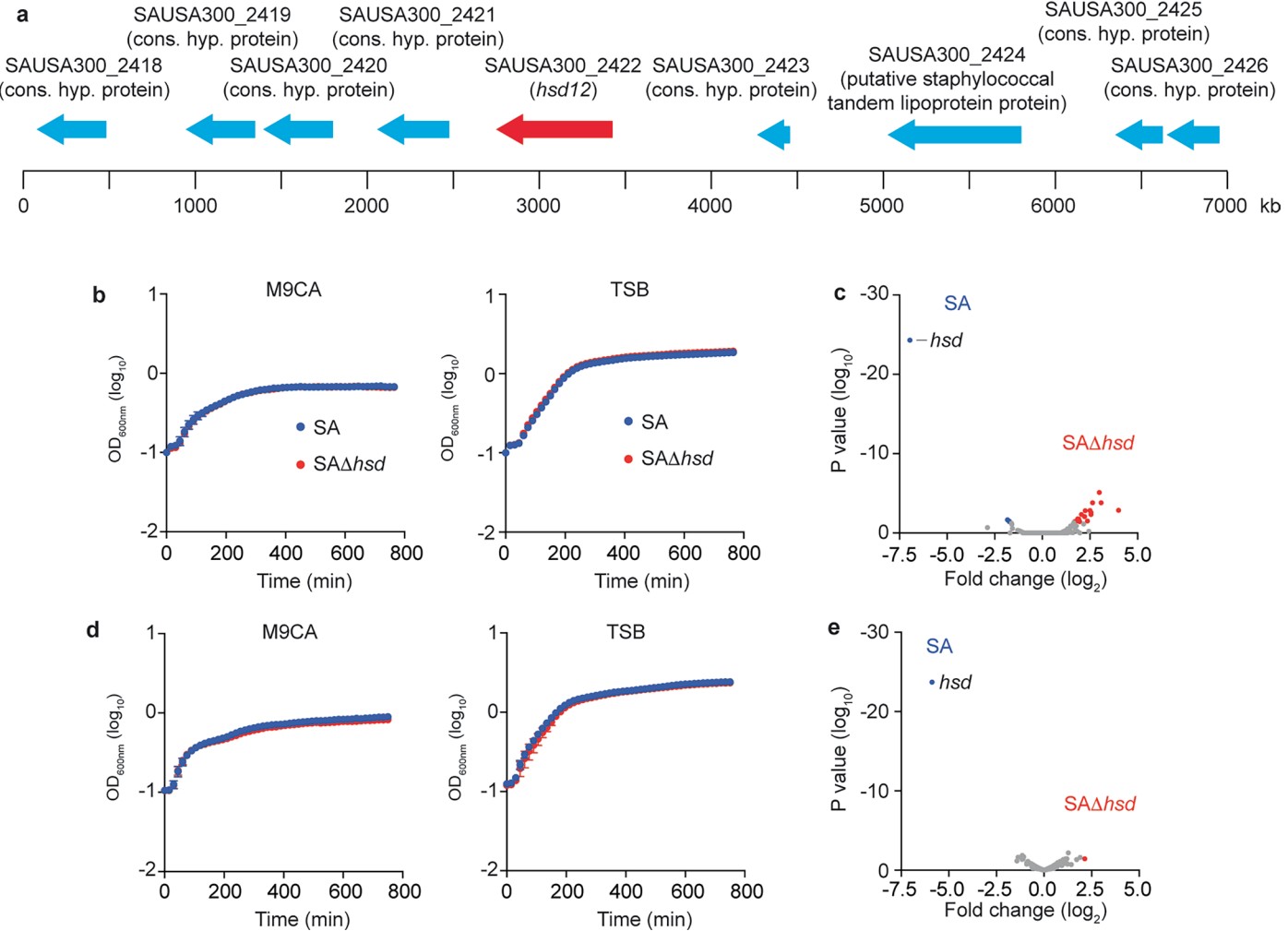

**Extended Data Fig. 8 | Location and deletion of the *S. aureus hsd12* gene.**
**a**, Genetic context of the *hsd12* gene, shown for *S. aureus* strain USA300 FPR3757.
**b**, Growth comparison of *S. aureus* wild-type (SA) and isogenic *hsd12* deletion
(SAΔ*hsd*) strains of strain ST398 in two different media. *n* = 4/group. **c**, RNA-Seq
transcriptomic comparison of gene expression in ST398 wild-type versus

isogenic *hsd12* deletion strains grown in TSB to logarithmic growth phase.
Significantly differentially expressed genes are coloured (red, up in Δ*hsd*; blue,
up in wild-type). **d**,**e**, Corresponding analyses for ST88 wild-type versus isogenic
*hsd12* deletion strains. *n* = 4/group. All error bars show the mean ± standard
deviation (SD).

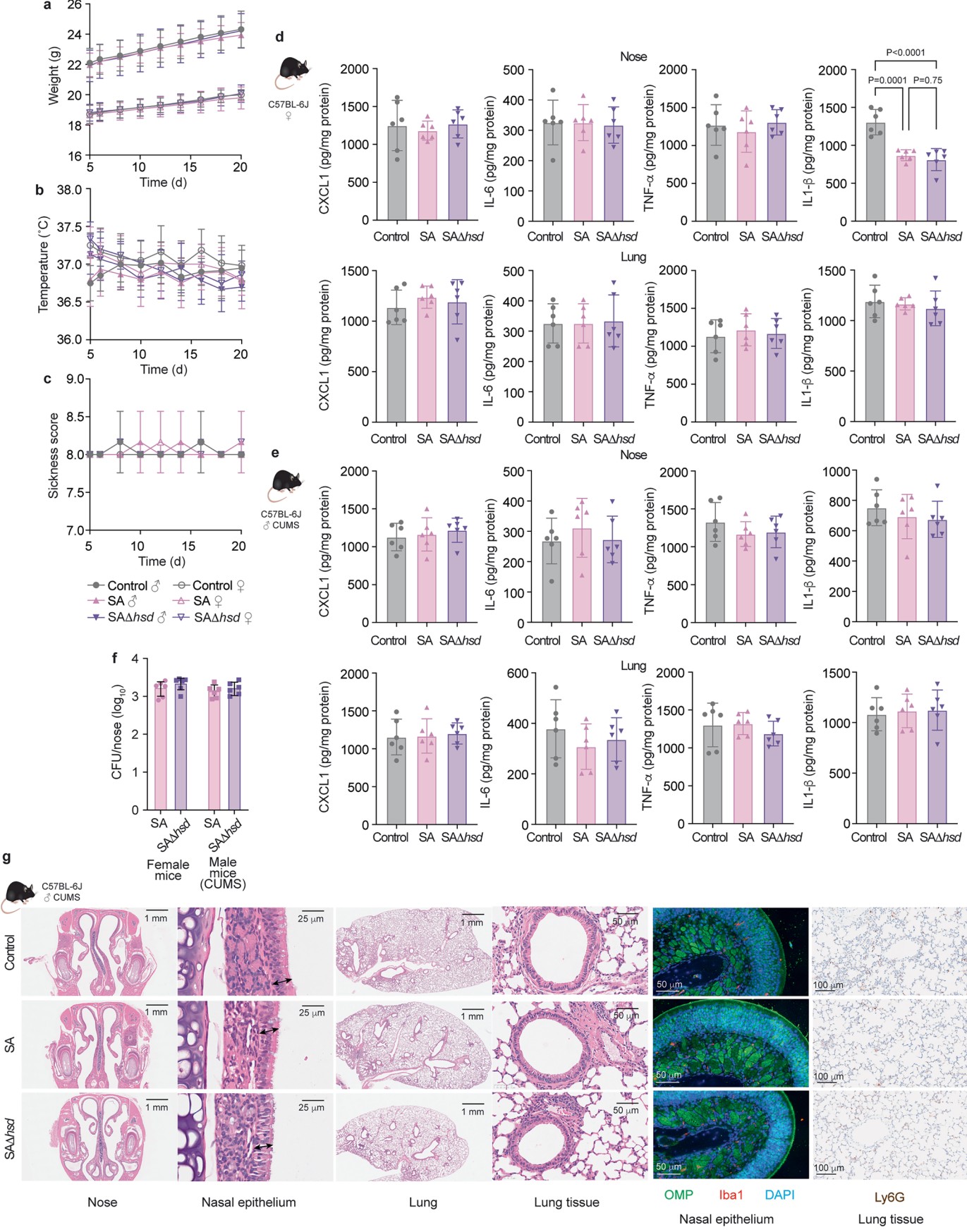

**Extended Data Fig. 9 | See next page for caption.**

**Extended Data Fig. 9 | Absence of inflammation and histology differences in mice nasally colonized by *S. aureus* wild-type and isogenic *hsd12* (ST398) deletion strains. a–c**, Weight, temperature and sickness scores over the time of the experiment. *n* = 6/group. **d**, Nose and lung cytokine concentrations in *S. aureus* (SA) nasally colonized, *S. aureus* Δ*hsd* (SAΔ*hsd*) nasally colonized, and control female mice. *n* = 6/group. **e**, Nose and lung cytokine concentrations in *S. aureus* nasally colonized, *S. aureus* Δ*hsd* nasally colonized, and control (CUMS-treated) male mice. *n* = 6/group. **f**, Colonizing CFU in the nose at the time of euthanasia. *n* = 6/group. **g**, Nose and lung histology in *S. aureus* nasally colonized, *S. aureus* Δ*hsd* nasally colonized, and control male (CUMS-treated) mice. Statistical analysis is by two-sided one-way ANOVA with Tukey's post-tests (**d**,**e**) and two-sided, unpaired *t*-tests (**f**). There were no statistically significant differences except for where marked. All error bars show the mean ± standard deviation (SD). Credit: mouse illustrations in **d**,**e**,**g**, creazilla under a Creative Commons license CC0 1.0.

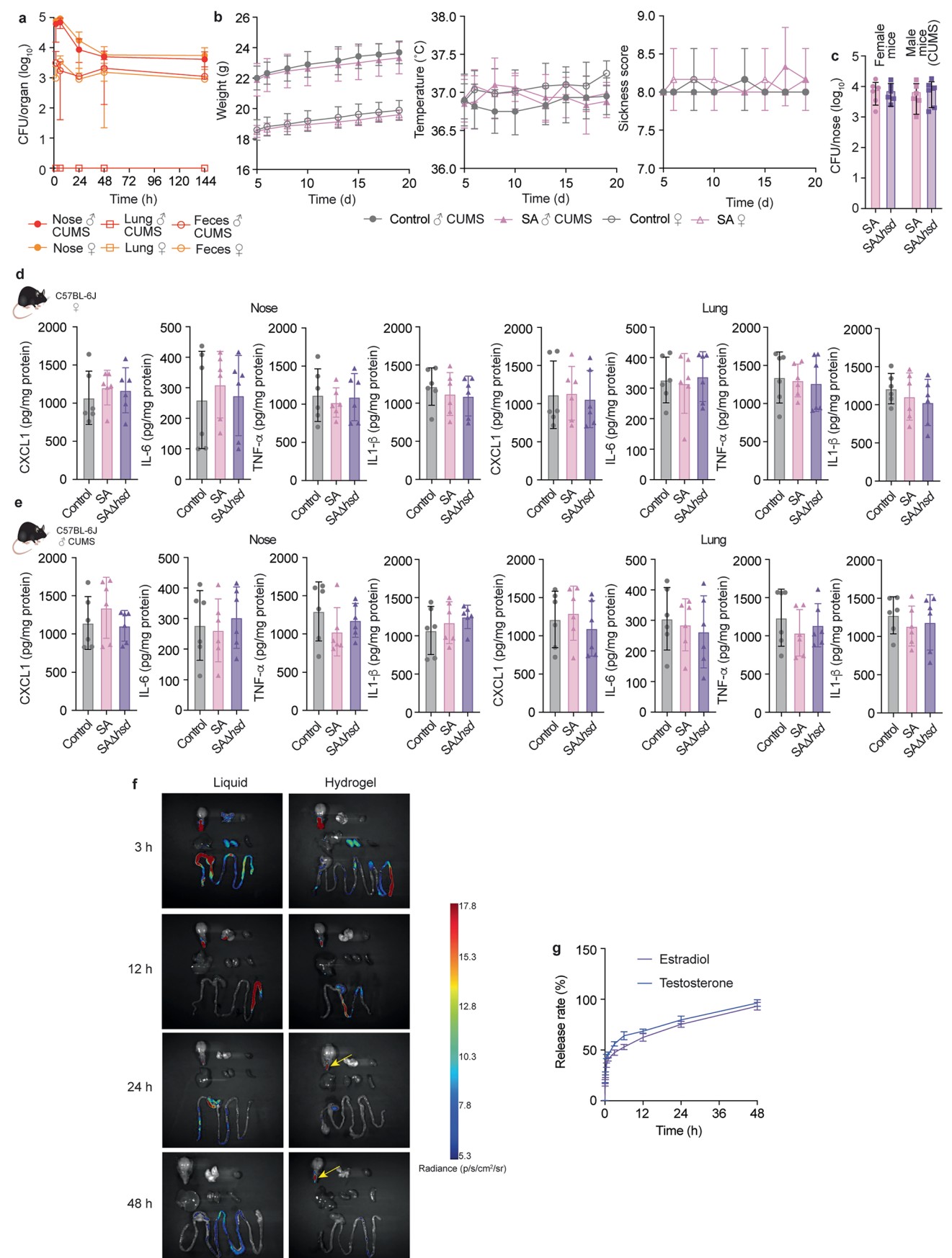

**Extended Data Fig. 10 | See next page for caption.**

**Extended Data Fig. 10 | ST88 experiments—additional information. a**, Bacterial loads in noses, lungs, and guts (faeces) after second colonization (day 13; time 0 h). $n$ = 6/group. **b**, Weights, body temperatures, and sickness scores over the time of the experiment. $n$ = 6/group. **c**, Colonizing CFU in the nose at day 20. $n$ = 6/group. **d,e** Nose and lung cytokine concentrations in *S. aureus* nasally colonized, *S. aureus* Δ*hsd* nasally colonized, and control female (**d**) and (CUMS-treated, **e**) male mice. $n$ = 6/group. **f**, Ex vivo imaging of nasal hydrogel-loaded FITC retention in the nose or excretion. Note prolonged retention in the nose (yellow arrows) in the hydrogel versus liquid control samples. **g**, In vitro cumulative release of estradiol or testosterone from the nasal hydrogel. Statistical analysis is by two-sided one-way ANOVA with Tukey's post-tests (**d,e**). There were no significant differences. $n$ = 3/group. All error bars show the mean ± standard deviation (SD). Credit: mouse illustrations in **d,e**, creazilla under a Creative Commons license CC0 1.0.

# Reporting Summary

## Statistics

For all statistical analyses, confirm that the following items are present in the figure legend, table legend, main text, or Methods section.

| n/a | Confirmed | |
|---|---|---|
| ☐ | ☒ | The exact sample size ($n$) for each experimental group/condition, given as a discrete number and unit of measurement |
| ☐ | ☒ | A statement on whether measurements were taken from distinct samples or whether the same sample was measured repeatedly |
| ☐ | ☒ | The statistical test(s) used AND whether they are one- or two-sided<br>*Only common tests should be described solely by name; describe more complex techniques in the Methods section.* |
| ☐ | ☒ | A description of all covariates tested |
| ☐ | ☒ | A description of any assumptions or corrections, such as tests of normality and adjustment for multiple comparisons |
| ☐ | ☒ | A full description of the statistical parameters including central tendency (e.g. means) or other basic estimates (e.g. regression coefficient) AND variation (e.g. standard deviation) or associated estimates of uncertainty (e.g. confidence intervals) |
| ☐ | ☒ | For null hypothesis testing, the test statistic (e.g. $F$, $t$, $r$) with confidence intervals, effect sizes, degrees of freedom and $P$ value noted<br>*Give P values as exact values whenever suitable.* |
| ☒ | ☐ | For Bayesian analysis, information on the choice of priors and Markov chain Monte Carlo settings |
| ☒ | ☐ | For hierarchical and complex designs, identification of the appropriate level for tests and full reporting of outcomes |
| ☐ | ☒ | Estimates of effect sizes (e.g. Cohen's $d$, Pearson's $r$), indicating how they were calculated |

*Our web collection on statistics for biologists contains articles on many of the points above.*

## Software and code

Policy information about availability of computer code

**Data collection**
Cultured bacterial species were identified by MALDI-TOF-MS (Bruker Daltonics, Bremen, Germany) using MALDI Biotyper Compass software v4.1 (Bruker Daltonics).
Full-length 16S rRNA gene sequencing was performed using the PacBio platform with SMRT Link software v11.0 (Pacific Biosciences).
Whole genome sequencing and RNA-seq were performed using the Illumina NovaSeq 6000 platform with Illumina Control Software v1.7 and Real-Time Analysis (RTA) v3.4.4 (Illumina).
LC-MS/MS analyses of nasal metabolomics were performed using a UHPLC system (Vanquish, Thermo Fisher Scientific) coupled to an Orbitrap Exploris 120 mass spectrometer with Xcalibur software v4.3 (Thermo Fisher Scientific).
LC-MS/MS analysis of steroid hormones was performed using the Waters ACQUITY UPLC / Xevo TQ-S mass spectrometer with MassLynx software v4.2 (Waters).
LC-MS/MS analysis of neurotransmitters was performed using the Waters ACQUITY Premier / SCIEX Triple Quad™ 6500+ mass spectrometer with SCIEX OS software v3.0 (SCIEX).
Mice behavior was recorded and analyzed using VisuTrack software v3.0 (Shanghai XinRuan Information Technology Co., Ltd.).
qRT-PCR data were obtained on an ABI 7500 thermocycler (Applied Biosystems) using 7500 Software v2.3 (Applied Biosystems).
Absorbance was detected using the BioTek Synergy2 microplate reader with Gen5 software v3.08 (BioTek Instruments).
A NanoZoomer S210 digital slide scanner was used to capture tissue paraffin sections for H&E staining with NDP.scan software v3.2 (Hamamatsu Photonics).

**Data analysis**
Full-length 16S rRNA gene sequencing data was performed using the DADA2 workflow in the QIIME2 software pipeline. The taxonomy of amplicon sequence variants (ASVs) sequences was analyzed using RDP Classifier version 2.13 against the NT_16S (v20221012) database. Whole-genome sequencing data were assembled using SPAdes v3.15.4, and MLST was identified using mlst v2.23.0 (https://github.com/tseemann/mlst).

The raw non-targeted metabolomics data were converted to the mzXML format using ProteoWizard and processed with an XCMS R package (v3.22.0), for peak detection, extraction, alignment, and integration. An in-house MS2 database (BiotreeDB) was used for metabolite annotation. Waters MassLynx V4.1 was employed for steroid hormones LC-MS/MS data acquisition and processing. SCIEX Analyst Work Station Software (Version 1.6.3) was employed for neurotransmitters LC-MS/MS data processing.

The RNA-seq data were analyzed using the HISAT2 (v2.2.1)-featureCounts (v2.0.3)-edgeR (v3.40.2) pipeline.

The Kyoto Encyclopedia of Genes and Genomes (KEGG) database was used to identify enriched pathways.

SDR superfamily members were identified using hmmer v3.0, by scanning protein sequences of the P24-2 (ST398) S. aureus genome.

Principal coordinate analyses (PCoA) and Orthogonal Partial Least Squares-Discriminant Analysis (OPLS-DA) were performed and evaluated using vegan and ropls R package (v1.30.0). qRT-PCR data and absorbance data were analyzed in Excel (Microsoft Office 2019). All other analysis methods are presented in the manuscript. Unless otherwise specified, statistical analysis was performed using Prism 10.2.0 software.

For manuscripts utilizing custom algorithms or software that are central to the research but not yet described in published literature, software must be made available to editors and reviewers. We strongly encourage code deposition in a community repository (e.g. GitHub). See the Nature Portfolio guidelines for submitting code & software for further information.

## Data

Policy information about availability of data

All manuscripts must include a data availability statement. This statement should provide the following information, where applicable:

- Accession codes, unique identifiers, or web links for publicly available datasets
- A description of any restrictions on data availability
- For clinical datasets or third party data, please ensure that the statement adheres to our policy

Raw microbiome sequencing and transcriptome data have been deposited in the NCBI's SRA database under Bioproject number PRJNA1138490. Raw metabolomics data have been deposited in the MetaboLights database under accession number MTBLS10742. The nucleotide sequences of putative HSD proteins identified in this study have been deposited in NCBI's GenBank database under accession number PQ067567 to PQ067586, and PQ106784 to PQ106786. All other data are presented in this manuscript. Source data files (see Supporting Material) contain results for all figures with quantitative data.

## Research involving human participants, their data, or biological material

Policy information about studies with human participants or human data. See also policy information about sex, gender (identity/presentation), and sexual orientation and race, ethnicity and racism.

| | |
|---|---|
| Reporting on sex and gender | Both male (n=58, healthy cohort; n=41, depressed cohort) and female (n=60, healthy cohort; n=59, depressed cohort) participants aged between 18 and 44 year were included. There was no significant difference in sex between health and depression groups. |
| Reporting on race, ethnicity, or other socially relevant groupings | All participants were Han Chinese. |
| Population characteristics | In average, the healthy participants were 31.1 ± 6.60 years old, and the depression participants 29.8 ± 6.78 years old. There was no significant difference in age between health and depression groups. |
| Recruitment | Depressive patients and healthy volunteers were enrolled between November 2022 to November 2024. Healthy volunteers were recruited from the Physical Examination Center of Renji Hospital. Depression patients were recruited from the Department of Psychological Medicine of Renji Hospital. Each participant was provided with a detailed written questionnaire by the research physicians. All participants were between 18 and 44 years of age. Depression patients were diagnosed according to DSM-IV by clinical psychologists and had no other psychiatric disorders or family history of any psychiatric disorders. The severity of depression and anxiety was evaluated using Patient Health Questionnaire-9 (PHQ-9) and Generalized Anxiety Disorder-7 (GAD-7) scores, respectively. Healthy controls were physically healthy and lacked any neurological illness or related family history, with PHQ-9 and GAD-7 scores both below 4. Other exclusion criteria for both control and depression groups included nasal or oral diseases, previous use of any type of antidepressant medication within the past three months, use of antimicrobial drugs within the past four weeks, thyroid dysfunction, diabetes, hypertension, autoimmune diseases, tumors, pregnancy, or lactation. |
| Ethics oversight | The human clinical study was approved by the ethics committee of Renji Hospital, Shanghai Jiao Tong University School of Medicine, Shanghai, China (approval number KY2022-139-B). Informed consent was obtained from all human research participants. |

Note that full information on the approval of the study protocol must also be provided in the manuscript.

# Field-specific reporting

Please select the one below that is the best fit for your research. If you are not sure, read the appropriate sections before making your selection.

☒ Life sciences  ☐ Behavioural & social sciences  ☐ Ecological, evolutionary & environmental sciences

For a reference copy of the document with all sections, see nature.com/documents/nr-reporting-summary-flat.pdf

# Life sciences study design

All studies must disclose on these points even when the disclosure is negative.

| | |
|---|---|
| Sample size | No statistical methods were used to pre-determine sample sizes but our sample sizes were chosen based on those reported in previous publications. For the human microbiome study, the number of analyzed individuals were consistent with previous study analyzing nasal microbiomes (reference: Nat Microbiol. 2023 Feb;8(2):218-230.). <br> For non-targeted metabolomics analysis, sample sizes were consistent to previous study analyzing nasal metabolites (reference: Sci Rep. 2022 Jun 15;12(1):10029.). <br> For animal studies, our sample sizes were chosen based on those reported data in previous publications (references: Cell Metab. 2023 Apr 4;35(4):685-694.e5. and Cell Host Microbe. 2024 Feb 14;32(2):227-243.e6. ). <br> For in vitro experiments, at least three replicates were performed, and deemed sufficient to achieve reliable results. |
| Data exclusions | In amplicon sequence analysis, to mitigate potential contamination effects, all ASVs detected in the negative controls were removed from subsequent analyses. Sequences with a quality score of less than three or expected errors greater than two were filtered out, and only ASV sequences present in at least two samples were retained to eliminate spurious features. <br> In non-targeted metabolomics analysis, the cutoff for peak annotation was set at 0.3 and only metabolites identified in MS2 level were used for analysis. <br> In heterologous expression of putative SDR protein in E. coli, putative SDR proteins were identified using HMMER v3.0 with an E-value below 1e-5. The sequences of putative SDR superfamily proteins were compared using blastp with the Uniprot database to exclude candidates with known other functions. |
| Replication | All in vitro sex hormones degrading experiments were performed with at least three replicates deemed sufficient to achieve reliable results, and these experiments have been validated multiple times and similar results have been obtained. Other experiments were not repeated, as they were all performed with a number of replicates deemed sufficient to achieve reliable results. |
| Randomization | In culture-based analysis, twenty-four random colonies were isolated and identified by MALDI-TOF-MS in each sample. <br> For nasal microbiota transplantation experiments, donor nasal microbiota samples were randomly selected from health and depression groups. <br> For non-targeted metabolomics analysis, metabolites were determined in the noses of n=40 randomly selected individuals from health and depression groups, respectively. <br> Sequence type distribution was analyzed using randomly selected S. aureus isolates (one per colonized individual; n=31, healthy cohort; n=45, depressed cohort). <br> Conversion of estradiol to estrone and testosterone to androstenedione were determined by culture filtrates from randomly selected different nasal bacteria (n=3/group) and randomly selected S. aureus isolates of main isolate STs (n=3/group). <br> Mice used in the experiments were litter mates, sex and age-matched, and randomized into control and experimental groups. |
| Blinding | For animal experiments, mouse tissue paraffin sections were used for H&E, IF and IHC. Slides were examined independently by a histopathologist who was blinded to the treatment. Blinding was not performed in other experiments. |

# Reporting for specific materials, systems and methods

We require information from authors about some types of materials, experimental systems and methods used in many studies. Here, indicate whether each material, system or method listed is relevant to your study. If you are not sure if a list item applies to your research, read the appropriate section before selecting a response.

## Materials & experimental systems

| n/a | Involved in the study |
|---|---|
| ☐ | ☒ Antibodies |
| ☒ | ☐ Eukaryotic cell lines |
| ☒ | ☐ Palaeontology and archaeology |
| ☐ | ☒ Animals and other organisms |
| ☒ | ☐ Clinical data |
| ☒ | ☐ Dual use research of concern |
| ☒ | ☐ Plants |

## Methods

| n/a | Involved in the study |
|---|---|
| ☒ | ☐ ChIP-seq |
| ☒ | ☐ Flow cytometry |
| ☒ | ☐ MRI-based neuroimaging |

## Antibodies

| | |
|---|---|
| Antibodies used | Primary antibodies: <br> Anti-IBA1 antibody (Guinea pig polyclonal antibody, Oasis biofarm, Catalog: OB-PGP049) was used to determine IBA1 at a dilution of 1:100. <br> Anti-Olfactory Marker Protein antibody (Rabbit polyclonal antibody, Bioss, Catalog: bs-19568R) was used to determine Olfactory Marker Protein at a dilution of 1:100. <br> Anti-Ly6G antibody (Rabbit monoclonal antibody, Abcam, Catalog: ab238132, Clone: EPR22909-135) was used to stain Neutrophils at a dilution of 1:500. <br> Secondary antibodies: <br> Goat anti rabbit IgG, HRP linked Antibody, Abcam, Catalog: ab205718, Antibody Dilution: 1:2000. |

5000

Alexa Fluor 594 goat-anti-guinea pig IgG, Oasis biofarm, Catalog: G-GP594, Antibody Dilution: 1:200.
Alexa Fluor 488 donkey anti-Rabbit IgG, ThermoFisher, Catalog: A21206, Antibody Dilution: 1:400.

Validation

Primary antibodies:
Anti-IBA1 antibody (Guinea pig polyclonal antibody, Catalog: OB-PGP049, RRID: AB_2934253)
https://www.oasisbiofarm.net/#/productdetails?proId=bd7ab575ebfa41009108908905941dd0&exp2=%E6%8A%97%E4%BD%93
Anti-Olfactory Marker Protein antibody
http://www.bioss.com.cn/prolook_03.asp?id=AF08169606023114&pro37=1
Anti-Ly6G antibody (Rabbit monoclonal antibody, Abcam, Catalog: ab238132, Clone: EPR22909-135)
https://www.abcam.cn/products/primary-antibodies/ly6g-antibody-epr22909-135-ab238132.html
Secondary antibodies:
Goat anti-rabbit IgG, HRP-linked Antibody, Abcam, Catalog: ab205718, Antibody Dilution: 1:2000.
https://www.abcam.cn/products/secondary-antibodies/goat-rabbit-igg-hl-hrp-ab205718.html
Alexa Fluor 594 goat-anti-guinea pig IgG, Oasis biofarm, Catalog: G-GP594.
https://www.oasisbiofarm.net/#/productdetails?proId=f823c846c54442509e2f9ea914a55b29&exp2=%E4%BA%8C%E6%8A%97
Alexa Fluor 488 donkey anti-Rabbit IgG, ThermoFisher, Catalog: A21206.
https://www.thermofisher.cn/cn/zh/antibody/product/Donkey-anti-Rabbit-IgG-H-L-Highly-Cross-Adsorbed-Secondary-Antibody-Polyclonal/A-21206

# Animals and other research organisms

Policy information about studies involving animals; ARRIVE guidelines recommended for reporting animal research, and Sex and Gender in Research

Laboratory animals

C57BL/6J specific pathogen-free (SPF) mice were purchased from GemPharmatech and bred in-house under SPF conditions. Mice were provided with food and water ad libitum and housed at consistent ambient temperature (22 ± 1°C) and humidity (50%±5%) with a 12-h light–dark cycle.

Wild animals

Study did not involve wild animals.

Reporting on sex

Both male and female C57BL/6J mice aged 6 weeks were used for experiments. Sex was considered in the study design because we found that sex hormones were reduced in the nasal cavity of depression people of both sexes and were related to increased abundance of S. aureus. Both male and female C57BL/6J mice received consistent nasal microbiota transplantation or nasal bacteria colonization operations. Both male and female C57BL/6J mice underwent consistent behavior tests and were used for RNA-seq, precursors of neurosteroids and neurotransmitter detection. Some male mice were additionally exposed to chronic unpredictable mild stress (CUMS). Testosterone levels were determined in the nose and brain tissues of male mice, while estradiol levels were determined in the nose and brain tissues of female mice.

Field-collected samples

Study did not involve field-collected samples.

Ethics oversight

All animal procedures followed the ethical guidelines outlined in the Guide for the Care and Use of Laboratory Animals proposed by the Institute for Laboratory Animal Research of the National Academy of Sciences, and all protocols were approved by the Animal Welfare Committee of Renji Hospital, Shanghai Jiao Tong University School of Medicine, Shanghai, China (approval number RJ2023-025B).

Note that full information on the approval of the study protocol must also be provided in the manuscript.

# Plants

Seed stocks

Not Applicable

Novel plant genotypes

Not Applicable

Authentication

Not Applicable

