## [Peer Review File · Nature Microbiology]

Nasal *Staphylococcus aureus* carriage promotes depressive behaviour in mice via sex hormone degradation

Corresponding Author: Dr Michael Otto

Version 0:

Reviewer comments:

Reviewer #1

(Remarks to the Author)

Comments to Authors:

The authors of this manuscript report nasal *Staphylococcus aureus* influence on mood and brain levels of dopamine and serotonin using SPF mice. They found more depression cases in *S. aureus* carriers in a series of 55 de novo depression patients, including 40 major depressive disorder and 102 healthy controls. They found metabolomic profile changes and sex hormone alterations in the nose of 40 randomly selected individuals from both groups and in a mouse nasal *S. aureus* colonization model. The authors identified a new isozyme encoded by a *S. aureus* gene they named *hsd12*, which can degrade the sex hormones testosterone and estradiol. Nasal colonization of mice with the *hsd12* deletion *S. aureus* strain did change neither mood nor estradiol, testosterone, dopamine and serotonin levels in the brain. Thus nasal *S. aureus* can lead to depression via sex hormone degradation in mice.

This study is well timed and relevant, and the experiments are sound whereas the small sample size. I have included comments that may be useful for improving the manuscript.

Specific concerns:

1) The authors claim that *S. aureus* induced neither inflammation nor histological alteration of nose tissues in nasally colonized mice. They should mention that nasal *S. aureus* can lead to inflammation of the olfactory epithelium (OE) with dramatic cell death of sensory neurons and neurological consequences such as anosmia in rats: Ge et al., Cell Death of Olfactory Receptor Neurons in a Rat with Nasosinusitis Infected artificially with *Staphylococcus*. *Chem. Senses* 27: 521–527, 2002. As *S. aureus* strongly damage OE at least in nasally instilled rats, the authors should analyze the OE of the mouse nasal colonization model by quantifying the OE thickness and the OE number of both IBA1+ cells (olfactory macrophages) and OMP+ cells (olfactory neurons, OMP=olfactory marker protein).

As nasal human microbiota/*Staphylococcus* /LPS might induce sickness in transplanted mice (beyond depression), they should analyze the weight curves of mice before and after nasal human microbiome instillation as well as nasal *S. aureus*/*S. epidermidis* instillation.

2) The authors claim that *S. aureus* is the predominant bacterium associated with depression. The authors should determine *S. aureus* abundance by quantitative PCR of species-specific genes for *S. aureus* to confirm the 16S DNA sequencing data.

Other comments:

- 1) The title should be completed by indicating that the study was performed in mice.
- 2) Fig 1a-b legend, page 20, please indicate that the microbiota diversities were analyzed at the species level.
- 3) Fig 1d legend, page 20, please indicate the sex of the mice. if mice of both sexes were used, please add in Extended Fig1 the behavioral data (OFT FST, TST) per sex.
- 4) Fig 1d legend, page 20, please indicate the number of donors and the number of mice per donor. How many mice per cage? One cage per donor?
- 5) Fig3c legend, please indicate when the protocol of chronic unpredictable mild stress was performed: during or between the antibiotics treatment and the series of nasal *Staphylococcus* instillation or after the bacterium instillation?
- 6) Abstract, line 36 page 2, please write “neuropsychiatric disease” instead of “neurologic disease”.
- 7) Main part, line 41 page 2, please write “neuropsychiatric disorder” instead of “neurologic disorder”.
- 8) Method part, Human participants, please indicate the date when the nasal swab samples were collected, and how they were

collected: Under endoscopic guidance and local anesthesia?

9) Method part, animal experiments, please indicate whether the nasal antibiotics and microbiota/Staphylococcus instillations were performed under anesthesia. Please also indicate the method of euthanasia.

Reviewer #2

(Remarks to the Author)

In this intriguing paper, the authors report that nasal colonization with *S. aureus* can induce a depression-like behavior and describe the depletion of sex hormones, esp. estrogen, by an *S. aureus*-derived enzyme, as the underlying mechanism. This study for the first time reports a direct link between the nasal microbiome and a major neurological disease. It derives its strength from the combination of both human clinical data and mouse models.

The results presented are of immediate interest not only to the infection research and neurology community, but also relevant to people from other disciplines. Employed methods are largely suitable, data are presented in appealing and easy to read figures and presentation and mostly interpreted with care. My major points of criticism concern the interpretation of correlation plots, and some flaws of the employed animal models, as detailed below.

Interpretation of correlation plots: The authors frequently used correlation plots to describe an association of two variables (2c, 4f,g, 5b, Fig ED 1, Fig ED 2d, ED 5b). They showed the R^2 (coefficients of determination) and the p-value. The authors repeatedly report that there is a "strong" or "significant" relationship between the tested parameters (line 93-95, lines 106/107, lines 154/155, 166/167,...). However, throughout the paper the authors refer to the significance (p-value) instead of the strength of the relationship. A statistically significant correlation does not necessarily mean that the strength of the correlation is strong. The p-value only shows the probability that this strength may occur by chance. Thus, it is important to avoid misunderstandings and data overinterpretation when reporting correlations and naming their strength. The authors can orient on the criteria provided by Akoglu et al., <https://doi.org/10.1016/j.tjem.2018.08.001> for judging the strength of the relationship (e.g. poor, fair, moderate, very strong,...), i.e. $R^2 < 0.5$ moderately strong correlation, $R^2 < 0.3$ fair correlation, $R^2 < 0.05$ poor correlation. Thus, the authors need to modify the description of correlation data in Fig. 2c, 4f,g, 5b,c, Fig ED 1, Fig ED 2d, ED 5b, accordingly and carefully rethink data interpretation in the discussion.

Antibiotic pre-treatment in transplantation and *S. aureus* colonization models: In both models, the authors pretreated the mice for 3 days with an antibiotic cocktail to clear the murine nose from endogenous microbiota. Please report how well this procedure reduced/eliminated the nasal microbiome. Moreover, SPF mice can have *S. aureus* in their endogenous microbiome (nose and gut; Schulz et al., *Front Cell Infect Microbiol.*, 2017). Such an unnoticed colonization could strongly impact on the reported results. Did the mouse vendor GemPharmatech provide information on the *S. aureus* status of the delivered animals? Please add this info to the methods section. Did the authors check whether the antibiotic treatment completely eliminated any *S. aureus* from the nares and gut?

There is also a strong gut microbiome-brain axis, which is also relevant in depression (Li et al, *Cell Host Microbe*, 2022). Did the authors clarify whether this intranasal treatment had an impact on the gut microbiome?

S. aureus "asymptomatic" nasal colonization model:

The authors used a rather uncommon and, in my eyes, artificial mouse model to study the impact of nasal *S. aureus* colonization on depression. Usually, mice are pretreated with antibiotics and afterwards inoculated with a single dose of *S. aureus*. When using mouse-adapted *S. aureus* strains and *S. aureus*-free SOPF mice, the antibiotic pre-treatment can even be omitted (Schulz et al., PMID: 28512627, Holtfreter et al., PMID: 24023720). Here, the authors repeatedly applied 10^7 CFU/nose once every two days for 7 days total treatments, which is unphysiological, causes a lot of stress for the mice and might also have a very different effect on the immune response (innate/adaptive memory).

Moreover, they applied a rather large volume (10 μ l per nostril instead of 5 μ l), risking that some bacteria reach the lungs. Indeed, a total volume of 30 μ l i.n. is often used to induce pneumonia. I wonder whether some of the observed phenotypes (OFT, FST, TST, ...) are due to sickness-related behavior rather than depression. Thus, it is essential to unequivocally show that the *S. aureus*-colonized mice were indeed symptom-free. Can the authors provide data on bacterial loads in the lung, ideally within 3-6 h after the last application as *S. aureus* is quickly cleared from the lung in the acute pneumonia model, as well as at the usual read-out time point? Similarly, it would be important to measure key cytokines (IL-6, TNF- α , IL-1 β) within the same time span. Please also provide the disease severity score (e.g. fur, breathing, behavior, weight, etc.) for the animals (again, disease symptoms in case of acute pneumonia are expected ca. 3-6h post installation).

Finally, intranasal *S. aureus* quickly and frequently translocates to the gut (Holtfreter et al., PMID: 24023720) where it could modulate the gut microbiome, and hence the gut-brain axis. Did the authors check for *S. aureus* P24-2 in the gut?

Due to the above-mentioned limitations, I currently can't agree with the statement in the discussion that the chosen model "well reflected" the human situation (lines 267ff). Indeed, I would suggest to recapitulate the key findings in a physiologically more relevant mouse colonization model. Mouse-adapted strains (Schulz et al., PMID: 28512627, Holtfreter et al., PMID: 24023720) for instance, enable persistent colonization (1) with a single dose and (2) without antibiotic pre-treatment, avoiding stress (which impacts on sex hormones via the HPA axis), and antibiotic-induced bystander effects on the gut microbiome.

Please also discuss limitations of this model in the manuscript.

Human cohort: The authors used a comparably small cohort (102 healthy, 55 depression) for studying the effect of the nasal microbiome on depression. Especially the overall weak/fair correlations between PHQ9, SA abundance, and nasal sex hormones (Fig. 2c, Fig. 4f, g, Fig. 5b) might have been clearer in a larger cohort study. Moreover, there is no human replication cohort, though data were re-evaluated in the mouse model whenever possible.

Microbiome transplant model (Line 87ff and 654ff): Please describe the nasal microbiota transplantation in more detail in the methods section. Were the donor samples pooled or was each mouse inoculated with the microbiome of a different donor? Did you expand the human microbiome by cultivation prior to inoculation? What was the recovered CFU/nose? Why did you apply the inoculum repeatedly for 7 days? What was the nasal bacterial load after inoculation and how quickly did it decline? Where the mice anaesthetized for the application of antibiotics and microbiota?

Definition of *S. aureus* carriage in humans:

S. aureus carriage was determined by two approaches: 16SrRNA data and culture-based (identification of 24 randomly picked colonies), with very different results (e.g. carriage rate in healthy ca. 30% by culture and 99% by 16S). The used culture-based approach likely underestimates the carriage rate, especially in women who have lower *S. aureus* loads in the nose than men. How well does the colonization density in culture approach correlate with the 16SrRNA-based *S. aureus* abundancies?

According to Liu et al. Sci. Adv. 2015 one could expect that Culture-based detection is strongly linked to *S. aureus* absolute abundance (determined by 16SrRNA-based quantification) but less sensitive.

Do the used 16SrRNA primers reliably differentiate between the different Staph species? In other words, is the annotation as "Staphylococcus aureus" reliable?

Hsd12 mutant: The authors did not include a complemented strain in their study. Can they show by WGS that there are no unwanted additional mutations in the mutant strain? Was the mutant attenuated in the nasal colonization model? Please show the nasal bacterial load of WT and Hsd12 mutant in the colonization model.

Study limitations: Please add a chapter on study limitations (small human cohort; mouse model, culture-based *S. aureus* detection,...).

Discussion: Overall, these are interesting and relevant findings, but what is the advantage for *S. aureus*? How does it benefit from shifting the sex hormone levels? Would it benefit from a depression-like status of its host?

Correlation vs. causation: In epidemiological studies, it is important not to confuse correlation with causation. Line 45: Do the authors mean "causes" or "associations/factors"?; Line 47: is the relationship between cytokines/neutrophic factors and neurotransmitters causal or associative?

Statistics: several figures (Fig 1d, 2b, 3b and 3d, 4d,e,...) show mean with SD, but data are compared with the U test. If the data are not normally distributed, it would be better to show median + IQR. Please state in all figure legend which statistical parameters are depicted (mean+SD or median+IQR).

Minor comments:

Fig. 4d, e: would it make sense to analyze females and males separately?

Fig. 5d: The authors report reduced sex hormone levels in nose and brain in SA vs. SE and Control groups. Did the authors check systemic (serum) sex hormone levels, too?

ED Fig. 1: Please add info on group sizes in the figure legend.

ED Fig 3: please specify the unit on the y axis. is it CFU/organ?

ED Fig. 5a: it looks like only a fraction of the study subjects provided serum samples. Please mention total numbers in the methods section (line 561).

ED Fig 7: OD values from bacterial growth data are usually plotted on log10 scale.

ED Table 1: Please provide ethnicity data for the cohorts.

Source data: please correct the total number of samples in the source data file for Fig. 2b.

Article Title: I suggest to rephrase into "Nasal *S. aureus* carriage promotes depression..."

Line 57/58: phrase "via alteration of microbial-derived neurotransmitters and sex hormone levels" is unclear. Please specify.

Line 122: "we colonized the noses of mice" is misleading as the authors are using an unusual colonization model "we colonized the noses of mice by repeated inoculation with xxx CFU".

Line 143/144: "metabolomics sequencing determination" is not the correct term

Line 148: Authors should check their statement "significant difference" (Fig. 4A). What do p-value and R² mean in this type of analysis? To me, the groups are largely overlapping and not different.

Line 181 (Fig. 5f, g): The current graph doesn't show data for CYP17A1 and HSD genes. You should add them. Also increase the size of the highlighted data points.

Line 182: Can you replace "affect" by "increase/inhibit/..." to be more specific?

Line 460: Please add info on the used method, e.g. 16SrRNA gene sequencing.

Line 471 (Fig 3 title): rather than "nasal *S. aureus*" you could write "nasal *S. aureus* carriage" in the figure legend title.

Line 495 (Fig. 5 title): "Nasal *S. aureus* [...] decreases dopamine and serotonin levels in the brain". This implies a causal, direct link between nasal colonization and dopamine levels, which is not provided in this figure yet. Please rephrase.

Line 550-553: please specify if the listed exclusion criteria applied for both cohorts or just the control group.

Line 600ff: specify from which samples you performed WGS. Only the *S. aureus* isolates?

Line 608: I understand that MLST typing was performed based on WGS data rather than DNA from individual *S. aureus* colonies? Please specify.

Line 654 and 664: Were the antibiotics and *S. aureus* inoculated under narcosis?

Line 680ff: The TST method section is hard to understand. I assume you wrap the tail with tape, with the tip sticking out about 1 cm, and then hold the mouse by the tail so that it hangs 15 cm above the ground? Please modify.

Line 720: specify bacterial culture (log or stat phase, culture medium)

Line 931: you could add the method here (e.g. "as determined by 16SrRNA gene sequencing")

Line 940: please specify the day of euthanasia.
Line 980: specify the culture medium.

Typos:

Line 32: identify a sex hormone-degrading enzyme in nasal commensal *Staphylococcus aureus*, ...
Line 105: "barely failed to reach" -> you mean "barely reached"?
Line 110: *S. aureus* colonizes the nose (...) only in about a quarter...
Figure 5e, 6d, 6j: Brian -> Brain
Line 688: water temperature
Line 722 and 724: remove "-" before μ
Line 735: write full name for "IS"
Line 505: transcriptomic -> transcriptome
Line 506: "up" -> increased
Line 562: "subpackaged" -> aliquoted
Line 602: with a Bacterial DNA Kit
Line 630: "metabolites level" -> metabolite levels

Reviewer #3

(Remarks to the Author)

Xiang, Wang, Ni and colleagues present an interesting manuscript. They investigate a possibly important correlation of the nasal microbiota composition and a (sex-specific) link to depressive disorder - this work might therefore be of high clinical relevance.

The work is outlined in a step-wise manner

1) Comparison of nasal microbiota composition of persons with depression and healthy controls. Depression was "lege artis" diagnosed according to DSM-IV manual by health care professionals and healthy controls were recruited from a physical examination center. PHQ-9 was used as a grading score (dep PHQ-9>15, controls <4). People with "nasal" disease were excluded as were persons under antidepressant treatment or antibiotic use up to 4 weeks before enrollment. Microbiota analysis is done by FL-16S-rRNA gene amplicon seq and culture (24 random colonies per swab culture). A DADA2 pipeline was used for taxonomic assignment.

Key findings were (by comparing 55 depression and 102 healthy persons) a higher alpha-diversity in healthy controls.

2) Transfer of nasal microbiota into mice (SPF BL/6J) and assessment of behavior using OFT, TST and FST. Randomly allocating microbiota from disease or control groups resulted in significant correlations between OFT, TST and FST and depression (as measured by PHQ-9).

3) More specifically there was a link between *S. aureus* colonization and depression. This association was not strain specific.

4) Colonization of female mice with *S. aureus* but not *S. epidermidis* led to increased "anxiety and depression" a.k.a scores above - this with similar colonization density and host-tissue response as assessed by cytokine measurement and histology.

5) Untargeted metabolomics of a subset of nasal swabs revealed significant differences in nasal metabolomes of depressed vs. healthy persons. Specifically nasal testosterone and estradiol levels were decreased in depressed as compared to healthy controls while serum levels were unaffected.

6) The effect was also measured with other bacteria, as well as different *S. aureus* clones and eventually in the in vivo model (*S. aureus* vs. *S. epi*). *S. aureus* colonization led to reduced levels of nasal estradiol and testosterone in the model; interestingly, this effect was also seen in the brain without upregulation of corresponding genes in transcriptomic analysis (midbrain) and as determined by precursor measurements.

7) Furthermore, in mouse brains, transcription of genes involved in Dopamine and Serotonin synthesis were downregulated in *S. aureus* carrying mice compared to controls. This was associated with reduced levels of the corresponding transmitters.

8) As end conversion of steroid sex hormones is mediated by hydroxysteroid dehydrogenases, the authors screened *S. aureus* genomes for the presence of this gene (or homologues) given that there was a negative correlation between hormone levels and *S. aureus* abundance. Hsd12 was identified and a deletion mutant created, that in female BL/6J mice did not induce a depression phenotype.

Overall, this is a very interesting manuscript investigating the role of nasal microbiota composition (i.e., *S. aureus* colonization) on BL/6J sex-specific behavior as a proxy for depression. The goal of the authors to strive to provide a mechanistic link instead of just reporting correlation is appreciated.

However, and unfortunately, the work in its current form is unable to provide sufficient evidence for the major claim (i.e., that nasal *S. aureus* promotes depression via sex hormone degradation).

Especially the human data analysis is insufficient and prone to bias.

Major:

- The title is not justified by the results; you only show associations in humans and some changes in BL/6J mice.
- The authors rightly state that depression is a "complex disorder whose pathogenesis is multifactorial and includes an array of underlying genetic, environmental, and endocrine causes." However, they then derive their main hypothesis by performing a Chi-Square test between a convenience sample of depressive patients and healthy controls. It is totally unclear how this

population was selected and what other drivers in this complex pathogenesis would and should be considered and these co-variables or confounders were not considered for this analysis. This seems crucial to prevent a biased analysis as for example a reactive depressive disorder might be something that would not be expected to be driven by *S. aureus* colonization. This obviously also applies for nasal hormone levels detected in healthy vs. depression.

- Also, curve fits for some of the regression are really questionable (e.g., 2c, 4f, 4g, e2d, e5b) doubting a linear relationship and again omitting any relevant confounders (i.e., 2c).
- It is unclear whether, albeit statistically significant, the differences in sex hormone levels are biologically relevant. This would have to be backed up by literature or experimentally. This especially in the light that a same trend in the same range is seen in blood levels (e5).
- The authors do not show, that *S. aureus* decolonization leads to a reversal of the "depressive phenotype". Given the claim that *S. aureus* decolonization might be a measure to control prevalence of depression (see abstract and discussion), this would be an absolute must.
- It would be assumed that the level of *S. aureus* colonization density would then correlate with hormone changes and this depression. This is lacking here. Relative abundance is not a good a measure for this.
- 17 β -HSDs catalyze reactions in steroidogenesis and steroid metabolism such as the interconversion of DHEA and androstenediol, estradiol and estrone as well as testosterone and androstenedione. There are different subtypes with slightly different major functions. As there were other bacteria that had quite a high level of estradiol metabolism I wonder whether for example *C. propinquum* would have similar effects.

Minor:

- please show a "table 1" with basic demographic information on the study participants
- Figure 1: how is *S. aureus* identification done? Is FL-16S really good enough to discriminate *S. aureus* from other Staphylococci?
- Figure 2b: relative abundance might be suboptimal as the presence of *S. aureus* itself might be associated with a change in overall bacterial load. Was an overall quantification done? Was there a correlation between random colony picking?
- please include information on the controls used in amplicon seq workflows (negative and pos controls, mock communities?)

Reviewer #4

(Remarks to the Author)

In the manuscript entitled, "Nasal Staphylococcus aureus promotes depression via sex hormone degradation", the authors do a very nice job describing their work. This is an exciting study with novel results that break open an understudied avenue of research in microbial-host interactions. With a few additional experiments and text edits, this work will be an advance in the field.

How many donors were used to generate mice with healthy vs depressed nasal microbiomes? The methods states that the donors were randomly selected. Does this mean that each datapoint in the behavior experiment panels represents a different mouse each colonized by a distinct donor? Then at this point were the mice single-housed? If so, please clarify details in the text and methods.

-If this is not the case, and only a single or small number of donors were used to colonize multiple mice, this will need to be clearly explained and defended, due to its problematic nature experimentally and statistically. Additional donors will need to be used to colonize mice and confirm a broad phenotype across recipient mice to make this data more meaningful.

If the donors were selected randomly, please denote in the data which of the donors in the graphs were ones that had SA in the nasal microbiome? Please answer whether the transfer of a healthy microbiome that has staph aureus sufficient to cause anxiety- and depressive-like phenotypes in mice.

Please add total locomotion/exploratory behavior, eg distance travelled for all OFT test behavior panels (in extended data figures) to serve as a control for health and movement of the animals spending less time in the center of the arena.

Does intranasal administration of the pure sex hormones decrease OFT, FST and TST phenotypes? Please include this experiment, which will be informative as to the mechanism and location that this relationship along the nose-brain axis is crucial. For instance, does the staph enzyme degrade the sex hormone locally in the nose? Is there indication that the HSD12 enzyme is secreted?

Line 144 The term "metabolomics sequencing determination of metabolites" is an unusual way to describe metabolomics. Consider rephrasing.

How many metabolite peaks were detected? How many were identified? It is mentioned that "analyses only included pathways related to steroid biosynthesis and metabolism", but this is not very meaningful if only a small handful of metabolites were identified and analyzed.

Consider adding in the text that the correlation in males in panel 4g looks to be driven solely by about 4 healthy individuals.

Decision Letter:

9th October 2024

Dear Dr Otto,

Thank you for your patience while your manuscript "Nasal Staphylococcus aureus promotes depression via sex hormone degradation" was under peer-review at Nature Microbiology. It has now been seen by four referees, whose expertise and comments you will find at the end of this email. Although they find your work of some potential interest, they have raised a number of concerns that will need to be addressed before we can consider publication of the work in Nature Microbiology.

In particular, you will see that all referees have expressed concerns regarding exaggeration of claims, unphysiological conditions used for colonisation of mice, biological relevance of the changes in hormone levels, and the lack of details of the human cohort or the transplantation of the microbiota. Thus, we'd highly recommend that you repeat critical experiments with a mouse-adapted *S. aureus* at a physiologically relevant dose, using appropriate methods to assess colonisation and complemented hsd12 mutant, to support the *S. aureus*-depressive phenotype link. An experiment with purified hormones and their degradation in the nasal cavity by *S. aureus* will also serve as a confirmation and address some doubts raised by the referees. We would also suggest verification using a larger human cohort to build more confidence in the assertion. However, for us to consider a revised version of this manuscript, all the concerns, not just the suggested ones, will need to be addressed. We'd also require a point-by-point response to all the comments made by the referees.

We acknowledge that the revision would entail a lot of additional work. If you feel it would be difficult to address all concerns in a reasonable timeframe, we'd be happy to consult with our editorial colleagues at Nature Communications to offer an easier path forward with fewer requirements.

Please include a data availability statement as a separate section after Methods but before references, under the heading "Data Availability". This section should inform readers about the availability of the data used to support the conclusions of your study. This information includes accession codes to public repositories (data banks for protein, DNA or RNA sequences, microarray, proteomics data etc...), references to source data published alongside the paper, unique identifiers such as URLs to data repository entries, or data set DOIs, and any other statement about data availability. At a minimum, you should include the following statement: "The data that support the findings of this study are available from the corresponding author upon request", mentioning any restrictions on availability. If DOIs are provided, we also strongly encourage including these in the Reference list (authors, title, publisher (repository name), identifier, year). For more guidance on how to write this section please see: <http://www.nature.com/authors/policies/data/data-availability-statements-data-citations.pdf>

* If you have not done so already we suggest that you begin to revise your manuscript so that it conforms to our Article format instructions at <http://www.nature.com/nmicrobiol/info/final-submission>. Refer also to any guidelines provided in this letter.

When submitting the revised version of your manuscript, please pay close attention to our [href="https://www.nature.com/nature-portfolio/editorial-policies/image-integrity">Digital Image Integrity Guidelines](https://www.nature.com/nature-portfolio/editorial-policies/image-integrity) and to the following points below:

- that unprocessed scans are clearly labelled and match the gels and western blots presented in figures.
- that control panels for gels and western blots are appropriately described as loading or sample processing controls
- all images in the paper are checked for duplication of panels and for splicing of gel lanes.

Link Redacted

Note: This url links to your confidential homepage and associated information about manuscripts you may have submitted or be reviewing for us. If you wish to forward this e-mail to co-authors, please delete this link to your homepage first.

Nature Microbiology is committed to improving transparency in authorship. As part of our efforts in this direction, we are now requesting that all authors identified as 'corresponding author' on published papers create and link their Open Researcher and Contributor Identifier (ORCID) with their account on the Manuscript Tracking System (MTS), prior to acceptance. This applies to primary research papers only. ORCID helps the scientific community achieve unambiguous attribution of all scholarly contributions. You can create and link your ORCID from the home page of the MTS by clicking on 'Modify my Springer Nature account'. For more information please visit www.springernature.com/orcid.

If you wish to submit a suitably revised manuscript we would hope to receive it within 6 months. If you cannot send it within this time, please let us know. We will be happy to consider your revision, even if a similar study has been accepted for publication at Nature Microbiology or published elsewhere (up to a maximum of 6 months).

Yours sincerely,

Reviewer Expertise:

Referee #1: Nasal-Brain axis
Referee #2: Staphylococcus aureus
Referee #3: Nasal microbiome
Referee #4: Gut-brain axis, metabolism

Reviewer Comments:

Reviewer #1 (Remarks to the Author):

Comments to Authors:

The authors of this manuscript report nasal *Staphylococcus aureus* influence on mood and brain levels of dopamine and serotonin using SPF mice. They found more depression cases in *S. aureus* carriers in a series of 55 de novo depression patients, including 40 major depressive disorder and 102 healthy controls. They found metabolomic profile changes and sex hormone alterations in the nose of 40 randomly selected individuals from both groups and in a mouse nasal *S. aureus* colonization model. The authors identified a new isozyme encoded by a *S. aureus* gene they named *hsd12*, which can degrade the sex hormones testosterone and estradiol. Nasal colonization of mice with the *hsd12* deletion *S. aureus* strain did change neither mood nor estradiol, testosterone, dopamine and serotonin levels in the brain. Thus nasal *S. aureus* can lead to depression via sex hormone degradation in mice.

This study is well timed and relevant, and the experiments are sound whereas the small sample size. I have included comments that may be useful for improving the manuscript.

Specific concerns:

1) The authors claim that *S. aureus* induced neither inflammation nor histological alteration of nose tissues in nasally colonized mice. They should mention that nasal *S. aureus* can lead to inflammation of the olfactory epithelium (OE) with dramatic cell death of sensory neurons and neurological consequences such as anosmia in rats: Ge et al., Cell Death of Olfactory Receptor Neurons in a Rat with Nasosinusitis Infected artificially with *Staphylococcus*. *Chem. Senses* 27: 521–527, 2002. As *S. aureus* strongly damage OE at least in nasally instilled rats, the authors should analyze the OE of the mouse nasal colonization model by quantifying the OE thickness and the OE number of both IBA1+ cells (olfactory macrophages) and OMP+ cells (olfactory neurons, OMP=olfactory marker protein).

As nasal human microbiota /*Staphylococcus* /LPS might induce sickness in transplanted mice (beyond depression), they should analyze the weight curves of mice before and after nasal human microbiome instillation as well as nasal *S. aureus*/*S. epidermidis* instillation.

2) The authors claim that *S. aureus* is the predominant bacterium associated with depression. The authors should determine *S. aureus* abundance by quantitative PCR of species-specific genes for *S. aureus* to confirm the 16S DNA sequencing data.

Other comments:

- 1) The title should be completed by indicating that the study was performed in mice.
- 2) Fig 1a-b legend, page 20, please indicate that the microbiota diversities were analyzed at the species level.
- 3) Fig 1d legend, page 20, please indicate the sex of the mice. if mice of both sexes were used, please add in Extended Fig1 the

behavioral data (OFT FST, TST) per sex.

4) Fig 1d legend, page 20, please indicate the number of donors and the number of mice per donor. How many mice per cage? One cage per donor?

5) Fig3c legend, please indicate when the protocol of chronic unpredictable mild stress was performed: during or between the antibiotics treatment and the series of nasal Staphylococcus instillation or after the bacterium instillation?

6) Abstract, line 36 page 2, please write "neuropsychiatric disease" instead of "neurologic disease".

7) Main part, line 41 page 2, please write "neuropsychiatric disorder" instead of "neurologic disorder".

8) Method part, Human participants, please indicate the date when the nasal swab samples were collected, and how they were collected: Under endoscopic guidance and local anesthesia?

9) Method part, animal experiments, please indicate whether the nasal antibiotics and microbiota/Staphylococcus instillations were performed under anesthesia. Please also indicate the method of euthanasia.

Reviewer #2 (Remarks to the Author):

In this intriguing paper, the authors report that nasal colonization with *S. aureus* can induce a depression-like behavior and describe the depletion of sex hormones, esp. estrogen, by an *S. aureus*-derived enzyme, as the underlying mechanism. This study for the first time reports a direct link between the nasal microbiome and a major neurological disease. It derives its strength from the combination of both human clinical data and mouse models.

The results presented are of immediate interest not only to the infection research and neurology community, but also relevant to people from other disciplines. Employed methods are largely suitable, data are presented in appealing and easy to read figures and presentation and mostly interpreted with care. My major points of criticism concern the interpretation of correlation plots, and some flaws of the employed animal models, as detailed below.

Interpretation of correlation plots: The authors frequently used correlation plots to describe an association of two variables (2c, 4f,g, 5b, Fig ED 1, Fig ED 2d, ED 5b). They showed the R^2 (coefficients of determination) and the p-value. The authors repeatedly report that there is a "strong" or "significant" relationship between the tested parameters (line 93-95, lines 106/107, lines 154/155, 166/167,...). However, throughout the paper the authors refer to the significance (p-value) instead of the strength of the relationship. A statistically significant correlation does not necessarily mean that the strength of the correlation is strong. The p-value only shows the probability that this strength may occur by chance. Thus, it is important to avoid misunderstandings and data overinterpretation when reporting correlations and naming their strength. The authors can orient on the criteria provided by Akoglu et al., <https://doi.org/10.1016/j.tjem.2018.08.001> for judging the strength of the relationship (e.g. poor, fair, moderate, very strong,...), i.e. $R^2 < 0.5$ moderately strong correlation, $R^2 < 0.3$ fair correlation, $R^2 < 0.05$ poor correlation. Thus, the authors need to modify the description of correlation data in Fig. 2c, 4f,g, 5b,c, Fig ED 1, Fig ED 2d, ED 5b, accordingly and carefully rethink data interpretation in the discussion.

Antibiotic pre-treatment in transplantation and *S. aureus* colonization models: In both models, the authors pretreated the mice for 3 days with an antibiotic cocktail to clear the murine nose from endogenous microbiota. Please report how well this procedure reduced/eliminated the nasal microbiome. Moreover, SPF mice can have *S. aureus* in their endogenous microbiome (nose and gut; Schulz et al., *Front Cell Infect Microbiol.*, 2017). Such an unnoticed colonization could strongly impact on the reported results. Did the mouse vendor GemPharmatech provide information on the *S. aureus* status of the delivered animals? Please add this info to the methods section. Did the authors check whether the antibiotic treatment completely eliminated any *S. aureus* from the nares and gut?

There is also a strong gut microbiome-brain axis, which is also relevant in depression (Li et al, *Cell Host Microbe*, 2022). Did the authors clarify whether this intranasal treatment had an impact on the gut microbiome?

S. aureus "asymptomatic" nasal colonization model:

The authors used a rather uncommon and, in my eyes, artificial mouse model to study the impact of nasal *S. aureus* colonization on depression. Usually, mice are pretreated with antibiotics and afterwards inoculated with a single dose of *S. aureus*. When using mouse-adapted *S. aureus* strains and *S. aureus*-free SOPF mice, the antibiotic pre-treatment can even be omitted (Schulz et al., PMID: 28512627, Holtfreter et al., PMID: 24023720). Here, the authors repeatedly applied 10^7 CFU/nose once every two days for 7 days total treatments, which is unphysiological, causes a lot of stress for the mice and might also have a very different effect on the immune response (innate/adaptive memory).

Moreover, they applied a rather large volume (10 μ l per nostril instead of 5 μ l), risking that some bacteria reach the lungs. Indeed, a total volume of 30 μ l i.n. is often used to induce pneumonia. I wonder whether some of the observed phenotypes (OFT, FST, TST, ...) are due to sickness-related behavior rather than depression. Thus, it is essential to unequivocally show that the *S. aureus*-colonized mice were indeed symptom-free. Can the authors provide data on bacterial loads in the lung, ideally within 3-6 h after the last application as *S. aureus* is quickly cleared from the lung in the acute pneumonia model, as well as at the usual read-out time point? Similarly, it would be important to measure key cytokines (IL-6, TNF- α , IL-1 β) within the same time span. Please also provide the disease severity score (e.g. fur, breathing, behavior, weight, etc.) for the animals (again, disease symptoms in case of acute pneumonia are expected ca. 3-6h post installation).

Finally, intranasal *S. aureus* quickly and frequently translocates to the gut (Holtfreter et al., PMID: 24023720) where it could modulate the gut microbiome, and hence the gut-brain axis. Did the authors check for *S. aureus* P24-2 in the gut?

Due to the above-mentioned limitations, I currently can't agree with the statement in the discussion that the chosen model "well reflected" the human situation (lines 267ff). Indeed, I would suggest to recapitulate the key findings in a physiologically more relevant mouse colonization model. Mouse-adapted strains (Schulz et al., PMID: 28512627, Holtfreter et al., PMID: 24023720)

for instance, enable persistent colonization (1) with a single dose and (2) without antibiotic pre-treatment, avoiding stress (which impacts on sex hormones via the HPA axis), and antibiotic-induced bystander effects on the gut microbiome. Please also discuss limitations of this model in the manuscript.

Human cohort: The authors used a comparably small cohort (102 healthy, 55 depression) for studying the effect of the nasal microbiome on depression. Especially the overall weak/fair correlations between PHQ9, SA abundance, and nasal sex hormones (Fig. 2c, Fig. 4f, g, Fig. 5b) might have been clearer in a larger cohort study. Moreover, there is no human replication cohort, though data were re-evaluated in the mouse model whenever possible.

Microbiome transplant model (Line 87ff and 654ff): Please describe the nasal microbiota transplantation in more detail in the methods section. Were the donor samples pooled or was each mouse inoculated with the microbiome of a different donor? Did you expand the human microbiome by cultivation prior to inoculation? What was the recovered CFU/nose? Why did you apply the inoculum repeatedly for 7 days? What was the nasal bacterial load after inoculation and how quickly did it decline? Where the mice anaesthetized for the application of antibiotics and microbiota?

Definition of *S. aureus* carriage in humans:

S. aureus carriage was determined by two approaches: 16SrRNA data and culture-based (identification of 24 randomly picked colonies), with very different results (e.g. carriage rate in healthy ca. 30% by culture and 99% by 16S). The used culture-based approach likely underestimates the carriage rate, especially in women who have lower *S. aureus* loads in the nose than men. How well does the colonization density in culture approach correlate with the 16SrRNA-based *S. aureus* abundances? According to Liu et al. Sci. Adv. 2015 one could expect that Culture-based detection is strongly linked to *S. aureus* absolute abundance (determined by 16SrRNA-based quantification) but less sensitive. Do the used 16SrRNA primers reliably differentiate between the different Staph species? In other words, is the annotation as “*Staphylococcus aureus*” reliable?

Hsd12 mutant: The authors did not include a complemented strain in their study. Can they show by WGS that there are no unwanted additional mutations in the mutant strain? Was the mutant attenuated in the nasal colonization model? Please show the nasal bacterial load of WT and Hsd12 mutant in the colonization model.

Study limitations: Please add a chapter on study limitations (small human cohort; mouse model, culture-based *S. aureus* detection,...).

Discussion: Overall, these are interesting and relevant findings, but what is the advantage for *S. aureus*? How does it benefit from shifting the sex hormone levels? Would it benefit from a depression-like status of its host?

Correlation vs. causation: In epidemiological studies, it is important not to confuse correlation with causation. Line 45: Do the authors mean “causes” or “associations/factors”?; Line 47: is the relationship between cytokines/neutrophic factors and neurotransmitters causal or associative?

Statistics: several figures (Fig 1d, 2b, 3b and 3d, 4d,e,...) show mean with SD, but data are compared with the U test. If the data are not normally distributed, it would be better to show median + IQR. Please state in all figure legend which statistical parameters are depicted (mean+SD or median+IQR).

Minor comments:

Fig. 4d, e: would it make sense to analyze females and males separately?

Fig. 5d: The authors report reduced sex hormone levels in nose and brain in SA vs. SE and Control groups. Did the authors check systemic (serum) sex hormone levels, too?

ED Fig. 1: Please add info on group sizes in the figure legend.

ED Fig 3: please specify the unit on the y axis. is it CFU/organ?

ED Fig. 5a: it looks like only a fraction of the study subjects provided serum samples. Please mention total numbers in the methods section (line 561).

ED Fig 7: OD values from bacterial growth data are usually plotted on log10 scale.

ED Table 1: Please provide ethnicity data for the cohorts.

Source data: please correct the total number of samples in the source data file for Fig. 2b.

Article Title: I suggest to rephrase into “Nasal *S. aureus* carriage promotes depression...”

Line 57/58: phrase “via alteration of microbial-derived neurotransmitters and sex hormone levels” is unclear. Please specify.

Line 122: “we colonized the noses of mice” is misleading as the authors are using an unusual colonization model “we colonized the noses of mice by repeated inoculation with xxx CFU”.

Line 143/144: “metabolomics sequencing determination” is not the correct term

Line 148: Authors should check their statement “significant difference” (Fig. 4A). What do p-value and R² mean in this type of analysis? To me, the groups are largely overlapping and not different.

Line 181 (Fig. 5f, g): The current graph doesn't show data for CYP17A1 and HSD genes. You should add them. Also increase the size of the highlighted data points.

Line 182: Can you replace “affect” by “increase/inhibit/...” to be more specific?

Line 460: Please add info on the used method, e.g. 16SrRNA gene sequencing.

Line 471 (Fig 3 title): rather than “nasal *S. aureus*” you could write “nasal *S. aureus* carriage” in the figure legend title.

Line 495 (Fig. 5 title): “Nasal *S. aureus* [...] decreases dopamine and serotonin levels in the brain”. This implies a causal, direct link between nasal colonization and dopamine levels, which is not provided in this figure yet. Please rephrase.

Line 550-553: please specify if the listed exclusion criteria applied for both cohorts or just the control group.

Line 600ff: specify from which samples you performed WGS. Only the *S. aureus* isolates?
Line 608: I understand that MLST typing was performed based on WGS data rather than DNA from individual *S. aureus* colonies? Please specify.
Line 654 and 664: Were the antibiotics and *S. aureus* inoculated under narcosis?
Line 680ff: The TST method section is hard to understand. I assume you wrap the tail with tape, with the tip sticking out about 1 cm, and then hold the mouse by the tail so that it hangs 15 cm above the ground? Please modify.
Line 720: specify bacterial culture (log or stat phase, culture medium)
Line 931: you could add the method here (e.g. "as determined by 16SrRNA gene sequencing")
Line 940: please specify the day of euthanasia.
Line 980: specify the culture medium.

Typos:

Line 32: identify a sex hormone-degrading enzyme in nasal commensal *Staphylococcus aureus*, ...
Line 105: "barely failed to reach" -> you mean "barely reached"?
Line 110: *S. aureus* colonizes the nose (...) only in about a quarter...
Figure 5e, 6d, 6j: Brian -> Brain
Line 688: water temperature
Line 722 and 724: remove "- before μ
Line 735: write full name for "IS"
Line 505: transcriptomic -> transcriptome
Line 506: "up" -> increased
Line 562: "subpackaged" -> aliquoted
Line 602: with a Bacterial DNA Kit
Line 630: "metabolites level" -> metabolite levels

Reviewer #3 (Remarks to the Author):

Xiang, Wang, Ni and colleagues present an interesting manuscript. They investigate a possibly important correlation of the nasal microbiota composition and a (sex-specific) link to depressive disorder - this work might therefore be of high clinical relevance.

The work is outlined in a step-wise manner

1) Comparison of nasal microbiota composition of persons with depression and healthy controls. Depression was "lege artis" diagnosed according to DSM-IV manual by health care professionals and healthy controls were recruited from a physical examination center. PHQ-9 was used as a grading score (dep PHQ-9>15, controls <4). People with "nasal" disease were excluded as were persons under antidepressant treatment or antibiotic use up to 4 weeks before enrollment. Microbiota analysis is done by FL-16S-rRNA gene amplicon seq and culture (24 random colonies per swab culture). A DADA2 pipeline was used for taxonomic assignment.

Key findings were (by comparing 55 depression and 102 healthy persons) a higher alpha-diversity in healthy controls.

2) Transfer of nasal microbiota into mice (SPF BL/6J) and assessment of behavior using OFT, TST and FST. Randomly allocating microbiota from disease or control groups resulted in significant correlations between OFT, TST and FST and depression (as measured by PHQ-9).

3) More specifically there was a link between *S. aureus* colonization and depression. This association was not strain specific.

4) Colonization of female mice with *S. aureus* but not *S. epidermidis* led to increased "anxiety and depression" a.k.a scores above - this with similar colonization density and host-tissue response as assessed by cytokine measurement and histology.

5) Untargeted metabolomics of a subset of nasal swabs revealed significant differences in nasal metabolomes of depressed vs. healthy persons. Specifically nasal testosterone and estradiol levels were decreased in depressed as compared to healthy controls while serum levels were unaffected.

6) The effect was also measured with other bacteria, as well as different *S. aureus* clones and eventually in the in vivo model (*S. aureus* vs. *S. epi*). *S. aureus* colonization led to reduced levels of nasal estradiol and testosterone in the model; interestingly, this effect was also seen in the brain without upregulation of corresponding genes in transcriptomic analysis (midbrain) and as determined by precursor measurements.

7) Furthermore, in mouse brains, transcription of genes involved in Dopamine and Serotonin synthesis were downregulated in *S. aureus* carrying mice compared to controls. This was associated with reduced levels of the corresponding transmitters.

8) As end conversion of steroid sex hormones is mediated by hydroxysteroid dehydrogenases, the authors screened *S. aureus* genomes for the presence of this gene (or homologues) given that there was a negative correlation between hormone levels and *S. aureus* abundance. Hsd12 was identified and a deletion mutant created, that in female BL/6J mice did not induce a depression phenotype.

Overall, this is a very interesting manuscript investigating the role of nasal microbiota composition (i.e., *S. aureus* colonization) on BL/6J sex-specific behavior as a proxy for depression. The goal of the authors to strive to provide a mechanistic link instead of just reporting correlation is appreciated.

However, and unfortunately, the work in its current form is unable to provide sufficient evidence for the major claim (i.e., that nasal *S. aureus* promotes depression via sex hormone degradation).

Especially the human data analysis is insufficient and prone to bias.

Major:

- The title is not justified by the results; you only show associations in humans and some changes in BL/6J mice.
- The authors rightly state that depression is a "complex disorder whose pathogenesis is multifactorial and includes an array of underlying genetic, environmental, and endocrine causes." However, they then derive their main hypothesis by performing a Chi-Square test between a convenience sample of depressive patients and healthy controls. It is totally unclear how this population was selected and what other drivers in this complex pathogenesis would and should be considered and these co-variables or confounders were not considered for this analysis. This seems crucial to prevent a biased analysis as for example a reactive depressive disorder might be something that would not be expected to be driven by *S. aureus* colonization. This obviously also applies for nasal hormone levels detected in healthy vs. depression.
- Also, curve fits for some of the regression are really questionable (e.g., 2c, 4f, 4g, e2d, e5b) doubting a linear relationship and again omitting any relevant confounders (i.e., 2c).
- It is unclear whether, albeit statistically significant, the differences in sex hormone levels are biologically relevant. This would have to be backed up by literature or experimentally. This especially in the light that a same trend in the same range is seen in blood levels (e5).
- The authors do not show, that *S. aureus* decolonization leads to a reversal of the "depressive phenotype". Given the claim that *S. aureus* decolonization might be a measure to control prevalence of depression (see abstract and discussion), this would be an absolute must.
- It would be assumed that the level of *S. aureus* colonization density would then correlate with hormone changes and this depression. This is lacking here. Relative abundance is not a good a measure for this.
- 17 β -HSDs catalyze reactions in steroidogenesis and steroid metabolism such as the interconversion of DHEA and androstenediol, estradiol and estrone as well as testosterone and androstenedione. There are different subtypes with slightly different major functions. As there were other bacteria that had quite a high level of estradiol metabolism I wonder whether for example *C. propinquum* would have similar effects.

Minor:

- please show a "table 1" with basic demographic information on the study participants
- Figure 1: how is *S. aureus* identification done? Is FL-16S really good enough to discriminate *S. aureus* from other Staphylococci?
- Figure 2b: relative abundance might be suboptimal as the presence of *S. aureus* itself might be associated with a change in overall bacterial load. Was an overall quantification done? Was there a correlation between random colony picking?
- please include information on the controls used in amplicon seq workflows (negative and pos controls, mock communities?)

Reviewer #4 (Remarks to the Author):

In the manuscript entitled, "Nasal *Staphylococcus aureus* promotes depression via sex hormone degradation", the authors do a very nice job describing their work. This is an exciting study with novel results that break open an understudied avenue of research in microbial-host interactions. With a few additional experiments and text edits, this work will be an advance in the field.

How many donors were used to generate mice with healthy vs depressed nasal microbiomes? The methods states that the donors were randomly selected. Does this mean that each datapoint in the behavior experiment panels represents a different mouse each colonized by a distinct donor? Then at this point were the mice single-housed? If so, please clarify details in the text and methods.

-If this is not the case, and only a single or small number of donors were used to colonize multiple mice, this will need to be clearly explained and defended, due to its problematic nature experimentally and statistically. Additional donors will need to be used to colonize mice and confirm a broad phenotype across recipient mice to make this data more meaningful.

If the donors were selected randomly, please denote in the data which of the donors in the graphs were ones that had SA in the nasal microbiome? Please answer whether the transfer of a healthy microbiome that has staph aureus sufficient to cause anxiety- and depressive-like phenotypes in mice.

Please add total locomotion/exploratory behavior, eg distance travelled for all OFT test behavior panels (in extended data figures) to serve as a control for health and movement of the animals spending less time in the center of the arena.

Does intranasal administration of the pure sex hormones decrease OFT, FST and TST phenotypes? Please include this experiment, which will be informative as to the mechanism and location that this relationship along the nose-brain axis is crucial. For instance, does the staph enzyme degrade the sex hormone locally in the nose? Is there indication that the HSD12 enzyme is secreted?

Line 144 The term "metabolomics sequencing determination of metabolites" is an unusual way to describe metabolomics. Consider rephrasing.

How many metabolite peaks were detected? How many were identified? It is mentioned that "analyses only included pathways related to steroid biosynthesis and metabolism", but this is not very meaningful if only a small handful of metabolites were identified and analyzed.

Consider adding in the text that the correlation in males in panel 4g looks to be driven solely by about 4 healthy individuals.

Version 1:

Reviewer comments:

Reviewer #1

(Remarks to the Author)

Thank you for the revised manuscript. All our questions were well addressed.

Reviewer #2

(Remarks to the Author)

The authors comprehensively addressed all questions raised by the reviewer. The additional animal experiments have resolved my worries that the observed depressive behavior is due to unnoticed *S. aureus* infections (sickness behavior).

Statistics: The authors have now included an accurate description of the error bars (SD) in all figure legends. Correlation data are now interpreted with more caution.

If available, please provide the nasal bacterial load for WT vs. Hsd12 mutant in the colonization model (to exclude that the mutant is per se attenuated).

Authors might want to swap Fig 3 and 4 so that their order matches their new order of reference in the results section.

Adjust citation format in lines 903-918.

Reviewer #3

(Remarks to the Author)

The authors have revised the manuscript and extended sample sizes and provided reanalyses and additional experimental data. As initially stated this is a highly interesting manuscript with very solid experimental approaches and high novelty but still has some critical limitations. As also outlined the main problems are discussion of limitations with the human data-set and -analyses as well reversal of phenotype by removal of *S. aureus*.

The title now accurately reflects what was done and shown in the manuscript which is highly appreciated. Nevertheless, some parts still slightly overselling presented findings and for the sake of consistency please also make association and causation (your mouse experiments) clear in the abstract. You don't establish proof that *S. aureus* in humans causes depression.

It is also acknowledged that the authors have expanded their cohort - however, limitations remain and this has to at least be more clearly discussed in the limitations section. Your cohort is not only "rather small" (as stated) but basically very healthy young (except the depression) people from one ethnicity that surprisingly neither anyone smokes or drinks alcohol or has any history of depression in the family. This indicates a highly specific cohort and limits generalizability. Additionally, confounders for depressive disorders are manifold and include:

Demographic & socioeconomic factors such as age, gender, race/ethnicity, income, education, employment status, marital status
clinical & psychiatric comorbidities such as chronic physical illnesses (e.g., diabetes, CVD), pain conditions and other mental disorders (anxiety disorders, PTSD, substance-use disorders)

Lifestyle & behavioral factors such as physical activity levels, dietary patterns, sleep habits, smoking, alcohol use, etc

Environmental & social factors such as childhood adversity or trauma, social isolation vs. support, loneliness

Genetic & biological predispositions such as genetic polymorphisms (e.g., BDNF, serotonin-transporter variants) or cortisol levels

Methodological issues especially selection bias and confounding by indication in observational studies

While you now indicate some of those in the baseline table, key factors such as education and other (see above) are lacking.

And still, your key analysis is based on a Fisher/Chi-Square between depressive and non-depressive patients and a correlation which (questionably) correlates abundance and PHQ 9 (Fig. 1). The absolute minimum is to extend a critical discussion of missing adjustment for unmeasured confounders that further limits the conclusions of human data. Normally you'd want to present some uni- vs. multivariate regression models comparing crude vs. adjusted ORs. Alternatively, as suggested by other reviewers, a validation cohort would be an option. Please expand the limitations.

You cite Kuehner (<https://pubmed.ncbi.nlm.nih.gov/27856392/>) for the claim that the impact of sex hormones on depression is reflected by the fact that depression is more common in women than men. However, this review discusses gender-related depression phenotypes in relation to different exposures where sex hormone alterations is one of them. Please consider rephrasing this sentence to accurately reflect the role of hormones in the pathophysiology (i.e., postpubertal phenotypes (e.g, early-onset has no sex difference)).

I was uncertain to find what you added in the multivariate analyses with MaAsLin2 except for age, sex and batch. Can you indicate what was used in which analysis in the legends?

Also, you might want to include some of the work that has looked at *S. aureus* and depression: While MRSA isolation in patients has been linked to depressive symptoms (likely due to stigma related factors); the correlation between carriage and depression itself is less strong (<https://pubmed.ncbi.nlm.nih.gov/11740872/>). In mice this work might be of interest: <https://pubs.acs.org/doi/10.1021/acs.est.4c09497> (Airborne Staphylococcus aureus Exposure Induces Depression-like Behaviors in Mice via Abnormal Neural Oscillation and Mitochondrial Dysfunction).

Experimentally, given your claims based on the well-done set of experiments, there is still a key experiment missing: does removal of *S. aureus* from mice nasal passage reverse the depressive phenotype. Although you show no effect with ko colonization, a reversal of depressive phenotype by removal of *S. aureus* would be a key experiment for the claims of this work. I've requested this experiment earlier (not in humans) - this would add substantial strength for your mechanistic claims. You can use narrow spectrum antibiotics for this such as vancomycin or mupirocin.

Also, I still am not satisfied by some of the correlation analyses (e.g., 1f, 3g) - how is this a "relevant" correlation/relationship between load and score? This should be discussed - see above.

Reviewer #4

(Remarks to the Author)

I appreciate the revised experiments and improvements to the manuscript. The additional experiments support the findings and clarify necessary components. My concerns are resolved.

Decision Letter:

19th May 2025

Dear Dr Otto,

Thank you for your patience while your manuscript "Nasal Staphylococcus aureus carriage is associated with human depression and promotes depressive behavior in mice via sex hormone degradation" was under peer-review at Nature Microbiology. It has now been seen by 4 referees, whose expertise and comments you will find at the of this email. You will see from their comments below that while they find your work of interest, some important points are raised. We are very interested in the possibility of publishing your study in Nature Microbiology, but would like to consider your response to these concerns in the form of a revised manuscript before we make a final decision on publication.

In particular, you will see that R3 has some remaining concerns regarding the relevance to the humans. As suggested by the referee, we recommend that you tone down the claims pertaining to human relevance and discuss the necessary caveats of this work. We do not require a new in vivo experiment but would require you to address the all the concerns highlighted by R3 to be addressed before we can proceed.

If you have not done so already please begin to revise your manuscript so that it conforms to our Article format instructions at <http://www.nature.com/nmicrobiol/info/final-submission/>

The usual length limit for a Nature Microbiology Article is six display items (figures or tables) and 3,000 words. We have some flexibility, and can allow a revised manuscript at 3,500 words, but please consider this a firm upper limit. There is a trade-off of ~250 words per display item, so if you need more space, you could move a Figure or Table to Supplementary Information.

Some reduction could be achieved by focusing any introductory material and moving it to the start of your opening 'bold' paragraph, whose function is to outline the background to your work, describe in a sentence your new observations, and explain your main conclusions. The discussion should also be limited. Methods should be described in a separate section following the discussion, we do not place a word limit on Methods.

Nature Microbiology titles should give a sense of the main new findings of a manuscript, and should not contain punctuation. Please keep in mind that we strongly discourage active verbs in titles, and that they should ideally fit within 90 characters each (including spaces).

Please include a data availability statement as a separate section after Methods but before references, under the heading "Data

Availability". This section should inform readers about the availability of the data used to support the conclusions of your study. This information includes accession codes to public repositories (data banks for protein, DNA or RNA sequences, microarray, proteomics data etc...), references to source data published alongside the paper, unique identifiers such as URLs to data repository entries, or data set DOIs, and any other statement about data availability. At a minimum, you should include the following statement: "The data that support the findings of this study are available from the corresponding author upon request", mentioning any restrictions on availability. If DOIs are provided, we also strongly encourage including these in the Reference list (authors, title, publisher (repository name), identifier, year). For more guidance on how to write this section please see: <http://www.nature.com/authors/policies/data/data-availability-statements-data-citations.pdf>

To improve the accessibility of your paper to readers from other research areas, please pay particular attention to the wording of the paper's opening bold paragraph, which serves both as an introduction and as a brief, non-technical summary in about 150 words. If, however, you require one or two extra sentences to explain your work clearly, please include them even if the paragraph is over-length as a result. The opening paragraph should not contain references. Because scientists from other sub-disciplines will be interested in your results and their implications, it is important to explain essential but specialised terms concisely. We suggest you show your summary paragraph to colleagues in other fields to uncover any problematic concepts.

If your paper is accepted for publication, we will edit your display items electronically so they conform to our house style and will reproduce clearly in print. If necessary, we will re-size figures to fit single or double column width. If your figures contain several parts, the parts should form a neat rectangle when assembled. Choosing the right electronic format at this stage will speed up the processing of your paper and give the best possible results in print. We would like the figures to be supplied as vector files - EPS, PDF, AI or postscript (PS) file formats (not raster or bitmap files), preferably generated with vector-graphics software (Adobe Illustrator for example). Please try to ensure that all figures are non-flattened and fully editable. All images should be at least 300 dpi resolution (when figures are scaled to approximately the size that they are to be printed at) and in RGB colour format. Please do not submit Jpeg or flattened TIFF files. Please see also 'Guidelines for Electronic Submission of Figures' at the end of this letter for further detail.

Figure legends must provide a brief description of the figure and the symbols used, within 350 words, including definitions of any error bars employed in the figures.

When submitting the revised version of your manuscript, please pay close attention to our [href="https://www.nature.com/nature-research/editorial-policies/image-integrity">Digital Image Integrity Guidelines. and to the following points below:](https://www.nature.com/nature-research/editorial-policies/image-integrity)

EXTENDED DATA FIGURES

Please include a statement before the acknowledgements naming the author to whom correspondence and requests for materials should be addressed.

Finally, we require authors to include a statement of their individual contributions to the paper -- such as experimental work, project planning, data analysis, etc. -- immediately after the acknowledgements. The statement should be short, and refer to authors by their initials. For details please see the Authorship section of our joint Editorial policies at http://www.nature.com/authors/editorial_policies/authorship.html

* include a point-by-point response to any editorial suggestions and to our referees. Please include your response to the editorial suggestions in your cover letter, and please upload your response to the referees as a separate document.

* ensure it complies with our format requirements for Letters as set out in our guide to authors at www.nature.com/nmicrobiol/info/gta/

* state in a cover note the length of the text, methods and legends; the number of references; number and estimated final size of figures and tables

* resubmit electronically if possible using the link below to access your home page:

Link Redacted

*This url links to your confidential homepage and associated information about manuscripts you may have submitted or be reviewing for us. If you wish to forward this e-mail to co-authors, please delete this link to your homepage first.

Please ensure that all correspondence is marked with your Nature Microbiology reference number in the subject line.

Nature Microbiology is committed to improving transparency in authorship. As part of our efforts in this direction, we are now requesting that all authors identified as 'corresponding author' on published papers create and link their Open Researcher and Contributor Identifier (ORCID) with their account on the Manuscript Tracking System (MTS), prior to acceptance. This applies to primary research papers only. ORCID helps the scientific community achieve unambiguous attribution of all scholarly contributions. You can create and link your ORCID from the home page of the MTS by clicking on 'Modify my Springer Nature account'. For more information please visit www.springernature.com/orcid.

We hope to receive your revised paper within three weeks. If you cannot send it within this time, please let us know.

Yours sincerely,

Reviewer Expertise:

Referee #1:

Referee #2:

Referee #3:

Reviewers Comments:

Reviewer #1 (Remarks to the Author):

Thank you for the revised manuscript. All our questions were well addressed.

Reviewer #2 (Remarks to the Author):

The authors comprehensively addressed all questions raised by the reviewer. The additional animal experiments have resolved my worries that the observed depressive behavior is due to unnoticed *S. aureus* infections (sickness behavior).

Statistics: The authors have now included an accurate description of the error bars (SD) in all figure legends. Correlation data are now interpreted with more caution.

If available, please provide the nasal bacterial load for WT vs. Hsd12 mutant in the colonization model (to exclude that the mutant is per se attenuated).

Authors might want to swap Fig 3 and 4 so that their order matches their new order of reference in the results section.

Adjust citation format in lines 903-918.

Reviewer #3 (Remarks to the Author):

The authors have revised the manuscript and extended sample sizes and provided reanalyses and additional experimental data. As initially stated this is a highly interesting manuscript with very solid experimental approaches and high novelty but still has some critical limitations. As also outlined the main problems are discussion of limitations with the human data-set and -analyses as well reversal of phenotype by removal of *S. aureus*.

The title now accurately reflects what was done and shown in the manuscript which is highly appreciated. Nevertheless, some parts still slightly overselling presented findings and for the sake of consistency please also make association and causation (your mouse experiments) clear in the abstract. You don't establish proof that *S. aureus* in humans causes depression.

It is also acknowledged that the authors have expanded their cohort - however, limitations remain and this has to at least be more clearly discussed in the limitations section. Your cohort is not only "rather small" (as stated) but basically very healthy young (except the depression) people from one ethnicity that surprisingly neither anyone smokes or drinks alcohol or has any history of depression in the family. This indicates a highly specific cohort and limits generalizability. Additionally, confounders for depressive disorders are manifold and include:

Demographic & socioeconomic factors such as age, gender, race/ethnicity, income, education, employment status, marital status
clinical & psychiatric comorbidities such as chronic physical illnesses (e.g., diabetes, CVD), pain conditions and other mental disorders (anxiety disorders, PTSD, substance-use disorders)

Lifestyle & behavioral factors such as physical activity levels, dietary patterns, sleep habits, smoking, alcohol use, etc

Environmental & social factors such as childhood adversity or trauma, social isolation vs. support, loneliness

Genetic & biological predispositions such as genetic polymorphisms (e.g., BDNF, serotonin-transporter variants) or cortisol levels

Methodological issues especially selection bias and confounding by indication in observational studies

While you now indicate some of those in the baseline table, key factors such as education and other (see above) are lacking. And still, your key analysis is based on a Fisher/Chi-Square between depressive and non-depressive patients and a correlation which (questionably) correlates abundance and PHQ 9 (Fig. 1). The absolute minimum is to extend a critical discussion of missing adjustment for unmeasured confounders that further limits the conclusions of human data. Normally you'd want to present some uni- vs. multivariate regression models comparing crude vs. adjusted ORs. Alternatively, as suggested by other reviewers, a validation cohort would be an option. Please expand the limitations.

You cite Kuehner (<https://pubmed.ncbi.nlm.nih.gov/27856392/>) for the claim that the impact of sex hormones on depression is reflected by the fact that depression is more common in women than men. However, this review discusses gender-related depression phenotypes in relation to different exposures where sex hormone alterations is one of them. Please consider rephrasing this sentence to accurately reflect the role of hormones in the pathophysiology (i.e., postpubertal phenotypes (e.g, early-onset has no sex difference)).

I was uncertain to find what you added in the multivariate analyses with MaAsLin2 except for age, sex and batch. Can you indicate what was used in which analysis in the legends?

Also, you might want to include some of the work that has looked at *S. aureus* and depression: While MRSA isolation in patients has been linked to depressive symptoms (likely due to stigma related factors); the correlation between carriage and depression itself is less strong (<https://pubmed.ncbi.nlm.nih.gov/11740872/>). In mice this work might be of interest:

<https://pubs.acs.org/doi/10.1021/acs.est.4c09497> (Airborne Staphylococcus aureus Exposure Induces Depression-like Behaviors in Mice via Abnormal Neural Oscillation and Mitochondrial Dysfunction).

Experimentally, given your claims based on the well-done set of experiments, there is still a key experiment missing: does removal of *S. aureus* from mice nasal passage reverse the depressive phenotype. Although you show no effect with ko colonization, a reversal of depressive phenotype by removal of *S. aureus* would be a key experiment for the claims of this work. I've requested this experiment earlier (not in humans) - this would add substantial strength for your mechanistic claims. You can use narrow spectrum antibiotics for this such as vancomycin or mupirocin.

Also, I still am not satisfied by some of the correlation analyses (e.g., 1f, 3g) - how is this a "relevant" correlation/relationship between load and score? This should be discussed - see above.

Reviewer #4 (Remarks to the Author):

I appreciate the revised experiments and improvements to the manuscript. The additional experiments support the findings and clarify necessary components. My concerns are resolved.

Version 2:

Decision Letter:

Our ref: NMICROBIOL-24082526B

17th June 2025

Dear Dr. Otto,

Thank you for submitting your revised manuscript "Nasal Staphylococcus aureus carriage is associated with human depression and promotes depressive behavior in mice via sex hormone degradation" (NMICROBIOL-24082526B). It has now been seen by the original referees and their comments are below. The reviewers find that the paper has improved in revision, and therefore we'll be happy to publish it in principle in Nature Microbiology, pending minor revisions to satisfy the referees' final requests and to comply with our editorial and formatting guidelines.

Thank you again for your interest in Nature Microbiology Please do not hesitate to contact me if you have any questions.

Best wishes,

Version 3:

Decision Letter:

14th August 2025

Dear Dr Otto,

I am pleased to accept your Article "Nasal Staphylococcus aureus carriage promotes depressive behaviour in mice via sex hormone degradation" for publication in Nature Microbiology. Thank you for having chosen to submit your work to us and many congratulations.

Authors may need to take specific actions to achieve compliance with funder and institutional open access mandates. If your research is supported by a funder that requires immediate open access (e.g. according to [a href="https://www.springernature.com/gp/open-science/plan-s-compliance">Plan S principles](https://www.springernature.com/gp/open-science/plan-s-compliance) or the [a href="https://www.springernature.com/gp/open-science/us-federal-agency-compliance">NIH public access policy](https://www.springernature.com/gp/open-science/us-federal-agency-compliance)) then you should select the gold OA route, and we will direct you to the compliant route where possible. Because authors warrant under our subscription licensing terms that they haven't committed to licensing any version of their article under a licence inconsistent with the terms of our agreement – including the applicable embargo period – publication under the subscription model isn't suitable for authors whose funders require no embargo.

An online order form for reprints of your paper is available at [a href="https://www.nature.com/reprints/author-reprints.html">https://www.nature.com/reprints/author-reprints.html](https://www.nature.com/reprints/author-reprints.html). All co-authors, authors' institutions and authors' funding agencies can order reprints using the form appropriate to their geographical region.

We welcome the submission of potential cover material (including a short caption of around 40 words) related to your manuscript; suggestions should be sent to Nature Microbiology as electronic files (the image should be 300 dpi at 210 x 297 mm in either TIFF or JPEG format). Please note that such pictures should be selected more for their aesthetic appeal than for their scientific content, and that colour images work better than black and white or grayscale images. Please do not try to design a cover with the Nature Microbiology logo etc., and please do not submit composites of images related to your work. I am sure you

will understand that we cannot make any promise as to whether any of your suggestions might be selected for the cover of the journal.

With kind regards,

P.S. Click on the following link if you would like to recommend Nature Microbiology to your librarian
<http://www.nature.com/subscriptions/recommend.html#forms>

** Visit the Springer Nature Editorial and Publishing website at http://editorial-jobs.springernature.com?utm_source=ejP_NMicro_email&utm_medium=ejP_NMicro_email&utm_campaign=ejp_NMicro for more information about our career opportunities. If you have any questions please click [here](mailto:editorial.publishing.jobs@springernature.com).

Open Access This Peer Review File is licensed under a Creative Commons Attribution 4.0 International License, which permits use, sharing, adaptation, distribution and reproduction in any medium or format, as long as you give appropriate credit to the original author(s) and the source, provide a link to the Creative Commons license, and indicate if changes were made. In cases where reviewers are anonymous, credit should be given to 'Anonymous Referee' and the source.

Response to reviewers

Reviewer Expertise:

Referee #1: Nasal-Brain axis

Referee #2: Staphylococcus aureus

Referee #3: Nasal microbiome

Referee #4: Gut-brain axis, metabolism

Reviewer Comments:

Reviewer #1 (Remarks to the Author):

Comments to Authors:

The authors of this manuscript report nasal *Staphylococcus aureus* influence on mood and brain levels of dopamine and serotonin using SPF mice. They found more depression cases in *S. aureus* carriers in a series of 55 de novo depression patients, including 40 major depressive disorder and 102 healthy controls. They found metabolomic profile changes and sex hormone alterations in the nose of 40 randomly selected individuals from both groups and in a mouse nasal *S. aureus* colonization model. The authors identified a new isozyme encoded by a *S. aureus* gene they named hsd12, which can degrade the sex hormones testosterone and estradiol. Nasal colonization of mice with the hsd12 deletion *S. aureus* strain did change neither mood nor estradiol, testosterone, dopamine and serotonin levels in the brain. Thus nasal *S. aureus* can lead to depression via sex hormone degradation in mice.

This study is well timed and relevant, and the experiments are sound whereas the small sample size. I have included comments that may be useful for improving the manuscript.

Reply: Thank you very much for your positive and encouraging remarks! As for the small sample size, we now increased the human study to at least 100 per cohort and reanalyzed all associated data. We would also like to stress that our study combines the human cohort analyses with a large number of mechanistic experiments and animal models that form the center of our manuscript and allowed us to base our conclusions not solely on correlative human data. We hope the reviewer appreciated this and the fact that therefore, the human part of our study cannot be at the level of a large, stand-alone human study.

Specific concerns:

1) The authors claim that *S. aureus* induced neither inflammation nor histological alteration of nose tissues in nasally colonized mice. They should mention that nasal *S. aureus* can lead to inflammation of the olfactory epithelium (OE) with dramatic cell death of sensory neurons and neurological consequences such as anosmia in rats: Ge et al., Cell Death of Olfactory Receptor Neurons in a Rat with Nasosinusitis Infected artificially with Staphylococcus. *Chem. Senses* 27: 521–527, 2002. As *S. aureus* strongly damage OE at least in nasally instilled rats, the authors should analyze the OE of the mouse nasal colonization model by quantifying the OE thickness and the OE number of both IBA1+ cells (olfactory macrophages) and OMP+ cells (olfactory neurons, OMP=olfactory marker protein).

As nasal human microbiota /Staphylococcus /LPS might induce sickness in transplanted mice (beyond depression), they should analyze the weight curves of mice before and after nasal human microbiome instillation as well as nasal *S. aureus*/*S. epidermis* instillation.

Reply: We took this criticism very serious and now respond with a large set of experiments as controls for our animal models. These experiments include the requested weight curves and histological analyses, and even much more to ascertain that our models are indeed free of inflammation/disease phenotypes. These new data sets are shown in Extended Data Fig. 2, 4, 5, 11, and 12.

2) The authors claim that *S. aureus* is the predominant bacterium associated with depression. The authors should determine *S. aureus* abundance by quantitative PCR of species-specific genes for *S. aureus* to confirm the 16S DNA sequencing data.

Reply: We appreciate the reviewer's criticism and entirely agree about the more conclusive power of absolute abundance measurements. We have now used qPCR to determine absolute abundance of *S. aureus* in the human part of our study and re-analyzed all associated correlative analyses on that basis.

Other comments:

1) The title should be completed by indicating that the study was performed in mice.

Reply: We previously wanted to keep the title as short as possible, but we do understand the reviewer's concern and thus changed the to: "Nasal *Staphylococcus aureus* carriage is associated with human depression and promotes depressive behavior in mice via sex hormone degradation." We also responded to the request of another reviewer to include "carriage". We hope this change finds the reviewer's approval.

2) Fig 1a-b legend, page 20, please indicate that the microbiota diversities were analyzed at the species level.

Reply: This was corrected as requested.

3) Fig 1d legend, page 20, please indicate the sex of the mice. if mice of both sexes were used, please add in Extended Fig1 the behavioral data (OFT FST, TST) per sex.

Reply: This was corrected as requested. Due to some re-organization of the manuscript, these sex-specific data are now in Extended Data Fig. 3a.

4) Fig 1d legend, page 20, please indicate the number of donors and the number of mice per donor. How many mice per cage? One cage per donor?

Reply: We apologize for not having included this information. We now added the relevant information to the figure legend, or, for more detailed description, to the methods. Male mice received nasal transplants from male and female from female donors (one donor per one mouse). There were n=8 male and n=8 female mice, and 4 mice were housed per cage.

5) Fig3c legend, please indicate when the protocol of chronic unpredictable mild stress was performed: during or between the antibiotics treatment and the series of nasal Staphylococcus instillation or after the bacterium instillation?

Reply: We included a more detailed description of when bacterial instillation was performed regarding the CUMS treatment period.

6) Abstract, line 36 page 2, please write “neuropsychiatric disease” instead of “neurologic disease”.

Reply: This was corrected as requested.

7) Main part, line 41 page 2, please write “neuropsychiatric disorder” instead of “neurologic disorder”.

Reply: This was corrected as requested.

8) Method part, Human participants, please indicate the date when the nasal swab samples were collected, and how they were collected: Under endoscopic guidance and local anesthesia?

Reply: This information was included in the updated methods part.

9) Method part, animal experiments, please indicate whether the nasal antibiotics and

microbiota/Staphylococcus instillations were performed under anesthesia. Please also indicate the method of euthanasia.

Reply: This information was included in the updated methods part.

Reviewer #2 (Remarks to the Author):

In this intriguing paper, the authors report that nasal colonization with *S. aureus* can induce a depression-like behavior and describe the depletion of sex hormones, esp. estrogen, by an *S. aureus*-derived enzyme, as the underlying mechanism. This study for the first time reports a direct link between the nasal microbiome and a major neurological disease. It derives its strength from the combination of both human clinical data and mouse models.

The results presented are of immediate interest not only to the infection research and neurology community, but also relevant to people from other disciplines. Employed methods are largely suitable, data are presented in appealing and easy to read figures and presentation and mostly interpreted with care. My major points of criticism concern the interpretation of correlation plots, and some flaws of the employed animal models, as detailed below.

Reply: We are immensely grateful to the reviewer for this positive assessment of our work and its impact. We hope that we could answer satisfactorily to the remaining criticism by the large set of new experiments that we performed and the change of interpretation along the lines of what the reviewer suggested.

Interpretation of correlation plots: The authors frequently used correlation plots to describe an association of two variables (2c, 4f,g, 5b, Fig ED 1, Fig ED 2d, ED 5b). They showed the R^2 (coefficients of determination) and the p-value. The authors repeatedly report that there is a “strong” or “significant” relationship between the tested parameters (line 93-95, lines 106/107, lines 154/155, 166/167,..). However, throughout the paper the authors refer to the significance (p-value) instead of the strength of the relationship. A statistically significant correlation does not

necessarily mean that the strength of the correlation is strong. The p-value only shows the probability that this strength may occur by chance. Thus, it is important to avoid misunderstandings and data overinterpretation when reporting correlations and naming their strength. The authors can orient on the criteria provided by Akoglu et al., <https://gcc02.safelinks.protection.outlook.com/?url=https%3A%2F%2Fdoi.org%2F10.1016%2Fj.tjem.2018.08.001&data=05%7C02%7Cmotto%40niaid.nih.gov%7C6216bb6cecd845ee1d9608dceab7affe%7C14b77578977342d58507251ca2dc2b06%7C0%7C0%7C638643322696139441%7CUnknown%7CTWFpbGZsb3d8eyJWIjoiMC4wLjAwMDAiLCJQIjoiV2luMzIiLCJBTiI6IklhaWwiLCJXVCI6Mn0%3D%7C60000%7C%7C%7C&sdata=NZVfD7CU1ko5ZKC7o3%2F0a5C7Y8RuZoY2N6U%2BQzKedw4%3D&reserved=0> for judging the strength of the relationship (e.g. poor, fair, moderate, very strong,...), i.e. $R^2 < 0.5$ moderately strong correlation, $R^2 < 0.3$ fair correlation, $R^2 < 0.05$ poor correlation. Thus, the authors need to modify the description of correlation data in Fig. 2c, 4f,g, 5b,c, Fig ED 1, Fig ED 2d, ED 5b, accordingly and carefully rethink data interpretation in the discussion.

Reply: We absolutely share the reviewer’s concern about strengths of correlations. As suggested, we now followed the classification by Chan et al. that is mentioned in the paper by Akoglu et al. for “medicine” (see Table 1 in Akoglu et al.), cited both studies, and specifically included the strength of correlation according to this classification in every case of correlational analysis in the text. Furthermore, for consistency, we everywhere show r computed by Spearman analysis (not as previously, Pearson’s R^2).

Antibiotic pre-treatment in transplantation and *S. aureus* colonization models: In both models, the authors pretreated the mice for 3 days with an antibiotic cocktail to clear the murine nose from endogenous microbiota. Please report how well this procedure reduced/eliminated the nasal microbiome.

Reply: We now show CFU after antibiotic treatment in the transplant model (Extended Data Fig. 2b shows the CFU over the course of the transplant experiment. The “dip” at 8 days of ~ 2 logs represents the antibiotic treatment effect). We also performed 16S rRNA sequencing to evaluate the effect of the antibiotic treatment. The treatment decreased the α

diversity and changed the β diversity of the nasal microbiota, while it did not change the α diversity or β diversity of the lung and gut microbiota (see Extended Data 2c-k).

Moreover, SPF mice can have *S. aureus* in their endogenous microbiome (nose and gut; Schulz et al., Front Cell Infect Microbiol., 2017). Such an unnoticed colonization could strongly impact on the reported results. Did the mouse vendor GemPharmatech provide information on the *S. aureus* status of the delivered animals? Please add this info to the methods section. Did the authors check whether the antibiotic treatment completely eliminated any *S. aureus* from the nares and gut?

Reply: The mouse vendor GemPharmatech provided certification that there was no *S. aureus* colonization in the nose, lung, gut or other body parts of the delivered animals.

There is also a strong gut microbiome-brain axis, which is also relevant in depression (Li et al, Cell Host Microbe, 2022). Did the authors clarify whether this intranasal treatment had an impact on the gut microbiome?

Reply: Yes, our new assessment included measurements of the lung and gut microbiomes. There was a significant change of the nasal, but not lung or gut microbiomes (see Extended Data Fig. 2c-k).

S. aureus “asymptomatic” nasal colonization model:

The authors used a rather uncommon and, in my eyes, artificial mouse model to study the impact of nasal *S. aureus* colonization on depression. Usually, mice are pretreated with antibiotics and afterwards inoculated with a single dose of *S. aureus*. When using mouse-adapted *S. aureus* strains and *S. aureus*-free SOPF mice, the antibiotic pre-treatment can even be omitted (Schulz et al., PMID: 28512627, Holtfreter et al., PMID: 24023720). Here, the authors repeatedly applied 10^7 CFU/nose once every two days for 7 days total treatments, which is unphysiological, causes a lot of stress for the mice and might also have a very different effect on the immune response (innate/adaptive memory).

Moreover, they applied a rather large volume (10 μ l per nostril instead of 5 μ l), risking that some

bacteria reach the lungs. Indeed, a total volume of 30 μ l i.n. is often used to induce pneumonia. I wonder whether some of the observed phenotypes (OFT, FST, TST, ...) are due to sickness-related behavior rather than depression. Thus, it is essential to unequivocally show that the *S. aureus*-colonized mice were indeed symptom-free. Can the authors provide data on bacterial loads in the lung, ideally within 3-6 h after the last application as *S. aureus* is quickly cleared from the lung in the acute pneumonia model, as well as at the usual read-out time point? Similarly, it would be important to measure key cytokines (IL-6, TNF- α , IL-1 β) within the same time span. Please also provide the disease severity score (e.g. fur, breathing, behavior, weight, etc.) for the animals (again, disease symptoms in case of acute pneumonia are expected ca. 3-6h post installation).

Finally, intranasal *S. aureus* quickly and frequently translocates to the gut (Holtfreter et al., PMID: 24023720) where it could modulate the gut microbiome, and hence the gut-brain axis. Did the authors check for *S. aureus* P24-2 in the gut?

Due to the above-mentioned limitations, I currently can't agree with the statement in the discussion that the chosen model "well reflected" the human situation (lines 267ff). Indeed, I would suggest to recapitulate the key findings in a physiologically more relevant mouse colonization model. Mouse-adapted strains (Schulz et al., PMID: 28512627, Holtfreter et al., PMID: 24023720) for instance, enable persistent colonization (1) with a single dose and (2) without antibiotic pre-treatment, avoiding stress (which impacts on sex hormones via the HPA axis), and antibiotic-induced bystander effects on the gut microbiome.

Reply: We appreciate the questions the reviewer had about our animal model. To respond, we used two approaches:

- (1) We ascertained that the colonization model we used with the human strain ST398 is free of changes that can be interpreted as infection or inflammation. This included a large set of experiments that included measurements of weight, temperature, physiological and behavioral signs according to Huet et. al (including fur aspect, activity, posture, behavior, respiration, chest sounds, eyes, body weight). See Extended Data Fig. 2 for control data on the nasal microbiome transplantation experiment and Extended Data Fig. 4 and 5 for those relating to the colonization experiment. None of those parameters changed much. Furthermore, these new**

analyses included comparison of CFU in the noses, lungs, and guts for several days past the time of the last bacterial application and RNA-seq analysis of the nasal mucosa at 3 h post the last colonization and ELISA to measure cytokine expression in lung tissue at 6 h post the last colonization. The observed gene expression changes at the 3-hour timepoint likely represent an early immune response to microbial colonization, reflecting physiological immune activation associated with commensal colonization. Three days post the last nasal colonization (before behavioral tests), both the SE and SA groups demonstrated normalization of inflammatory factors (IL-6, IL-1 β , TNF- α , CXCL1) to baseline levels comparable to the control group. Notably, the mice exhibited no overt signs of infection and despite complete microbial persistence in the nasal cavity, translocated bacteria in the intestinal tract had been fully cleared.

- (2) As suggested, we now performed key experiments using the mouse-adapted strain ST88 with only two-times application and strongly reduced CFU (see Figure 6 and Extended Data Fig. 12). All these experiments confirmed what we had observed with ST398. However, we note and discuss in the manuscript that ST88 colonization, in contrast to ST398 colonization, led to persistent colonization of the gut (compare Extended Data Fig. 5g and 12a). Thus, with ST88 a contribution of the gut-brain axis in our experiments cannot be excluded, while for ST398 it is much less likely. We therefore believe that a combination of results of both ST398 (focus on nose-brain axis) and ST88 (more “physiological” condition) strongly benefited our manuscript and its conclusions, and we are grateful to the reviewer for encouraging us to include ST88 analysis and experiments.

In addition to our CFU assessment, we also ascertained that *S. aureus* did not enter the lung 2 h post installation using small animal imaging technology. Please see the following data (we did not include them in the manuscript, as we already have the CFU data included):

Please also discuss limitations of this model in the manuscript.

Reply: We added a limitations paragraph to the discussion.

Human cohort: The authors used a comparably small cohort (102 healthy, 55 depression) for studying the effect of the nasal microbiome on depression. Especially the overall weak/fair correlations between PHQ9, SA abundance, and nasal sex hormones (Fig. 2c, Fig. 4f, g, Fig. 5b) might have been clearer in a larger cohort study. Moreover, there is no human replication cohort, though data were re-evaluated in the mouse model whenever possible.

Reply: We now increased the sample size of the human study by adding more participants particularly to the "depressed" cohort, achieving a sample size of at least 100 per cohort, and reanalyzed all associated data. We would also like to stress, and the reviewer alludes to this situation, that our study combines the human cohort analyses with a large number of mechanistic experiments and animal models that form the center of our manuscript and allowed us to base our conclusions not solely on correlative human data. We believe the reviewer understands that due to that overall setup of our study, the human part of our study cannot be at the level of a large, stand-alone human study.

Microbiome transplant model (Line 87ff and 654ff): Please describe the nasal microbiota transplantation in more detail in the methods section. Were the donor samples pooled or was each mouse inoculated with the microbiome of a different donor?

Reply: We apologize for not having included this information. We now added the relevant information to the figure legend, or, for more detailed description, to the methods. Male mice received nasal transplants from male and female from female donors (one donor per one mouse). There were n=8 male and n=8 female mice, and 4 mice were housed per cage.

Did you expand the human microbiome by cultivation prior to inoculation?

Reply: No.

What was the recovered CFU/nose?

Reply: We performed CFU assessment (by plating) for the colonization experiments in the noses, lungs, and guts (see Extended Data Fig. 5g and 12a).

In the transplant experiment, we observed a considerable (~ 2-log) reduction in CFU following antibiotic treatment (Extended Data Fig. 2b). Upon receiving human-derived nasal microbiota transplantation, mice exhibited substantial microbial recolonization, with restored bacterial loads approaching the initial inoculation concentration of donor suspensions (approximately 10^3 - 10^5 CFU). This recovery pattern indicates successful short-term engraftment of transplanted microbiota in the murine nasal cavity.

Peak bacterial load was achieved 24 hours after the final transplantation (Day 17 of the experimental timeline). Afterwards, the load gradually decreased and was accompanied by the recovery of the native microbiota in mice.

While we now performed the requested CFU analysis, we judged that analysis of microbiome changes is also relevant. To that end, we now used to 16S rRNA sequencing to compare the ecological differences between mice colonized with patient-derived vs. healthy donor-derived microbiota (see Extended Data Fig. 2l-n). In general, the extensive 16S rRNA sequencing data we now performed to evaluate the transplant model provide important insights into the dynamics of bacterial load post-inoculation: 1. Antibiotic-driven depletion: Nasal antibiotic pretreatment significantly reduced microbial diversity, indicating effective clearance of resident bacteria and creation of a "niche vacancy" for

donor microbiota. 2. Divergence between donor groups: Mice transplanted with patient-derived microbiota exhibited distinct α/β -diversity compared to healthy donor recipients, suggesting that the inoculated bacterial load successfully colonized and altered the community structure. 3. Convergence to baseline at 2 weeks: By 2 weeks post the last transplantation, both α -diversity (Shannon and Simpson index) and β -diversity (Bray-Curtis) of transplanted mice were statistically indistinguishable from untreated controls.

See Extended Data Fig. 2c-q for these analyses.

Why did you apply the inoculum repeatedly for 7 days?

Reply: We applied the inoculum repeatedly for 7 days to ensure effective colonization, stabilization, and reinforcement of the transplanted microbiota, following the depletion of their native microbiota through antibiotic treatment. This extended inoculation strategy has often been performed in similar studies for other bacteria and has been validated in gut microbiota transplantation studies to improve microbial community stability.

(We believe the reviewer here referred to the transplant experiment. As for the colonization experiment, we understand the problems the reviewer has with the model and refer to our answers to the comments above, our extensive added control and ST88 data.)

What was the nasal bacterial load after inoculation and how quickly did it decline?

Reply: Please see our reply above.

Where the mice anaesthetized for the application of antibiotics and microbiota?

Reply: Yes, we now specified this in the methods part. The mice were anaesthetized using isoflurane for the application of antibiotics and microbiota.

Definition of *S. aureus* carriage in humans:

S. aureus carriage was determined by two approaches: 16SrRNA data and culture-based (identification of 24 randomly picked colonies), with very different results (e.g. carriage rate in

healthy ca. 30% by culture and 99% by 16S). The used culture-based approach likely underestimates the carriage rate, especially in women who have lower *S. aureus* loads in the nose than men. How well does the colonization density in culture approach correlate with the 16SrRNA-based *S. aureus* abundancies? According to Liu et al. Sci. Adv. 2015 one could expect that Culture-based detection is strongly linked to *S. aureus* absolute abundance (determined by 16SrRNA-based quantification) but less sensitive.

Do the used 16SrRNA primers reliably differentiate between the different Staph species? In other words, is the annotation as “*Staphylococcus aureus*” reliable?

Reply: We understand the reviewer’s concerns about the difference of measurement using culture-based versus sequencing approaches, which is why we included both approaches in our study. We analyzed the correlation between these two methods; it is good but not great (according to Akoglu et al. /Chan et al, it would be considered “moderate”):

Notably, we recalculated all correlative analyses based on our new, qPCR-based assessment of absolute abundance.

As for the primers, these amplify the entire 16S rRNA, a method reportedly appropriate to distinguish staphylococcal species. We discuss this now and included references.

Hsd12 mutant: The authors did not include a complemented strain in their study. Can they show by WGS that there are no unwanted additional mutations in the mutant strain? Was the mutant attenuated in the nasal colonization model? Please show the nasal bacterial load of WT and Hsd12 mutant in the colonization model.

Reply: We performed WGS and transcriptomic analysis to ascertain fidelity of the mutation and absence of secondary site mutations that affect phenotypes. See Extended Data Fig. 10. WGS showed only two SNPs, a relatively low number considering how fast *S. aureus* can acquire mutations in the laboratory. One was intergenic, and one was in a non-conserved amino acid position of a protein (dihydropicolinate synthase). We believe it is fair to assume these do not affect the measured phenotypes and most likely, they don't have any phenotype at all.

Study limitations: Please add a chapter on study limitations (small human cohort; mouse model, culture-based *S. aureus* detection,...).

Reply: We added a paragraph on study limitations to the discussion.

Discussion: Overall, these are interesting and relevant findings, but what is the advantage for *S. aureus*? How does it benefit from shifting the sex hormone levels? Would it benefit from a depression-like status of its host?

Reply: This is an interesting question that we now discuss. We are afraid there are no conclusive answers. It has been speculated that *S. aureus* may use sex hormones as food sources, and we have cited that study.

Correlation vs. causation: In epidemiological studies, it is important not to confuse correlation with causation. Line 45: Do the authors mean "causes" or "associations/factors"?; Line 47: is the relationship between cytokines/neutrophilic factors and neurotransmitters causal or associative?

Reply: We completely agree with the reviewer as for the important distinction between association and causation. This was the most important reason for not solely reporting associations found in humans but to conduct an extensive mechanistic study in mice.

(previous) line 45, we corrected to "factors".

(previous) line 47, in this instance in the introduction, we are simply discussing the literature and refrain from making any judgement about whether a causative relationship

has been conclusively established in those studies.

Statistics: several figures (Fig 1d, 2b, 3b and 3d, 4d,e,...) show mean with SD, but data are compared with the U test. If the data are not normally distributed, it would be better to show median + IQR. Please state in all figure legend which statistical parameters are depicted (mean+SD or median+IQR).

Reply: We understand the concern of the reviewer and thought intensively about the suggestion. However, showing IQR instead of SD in cases of non-normally distributed data in our experience is quite uncommon. Furthermore, we meticulously analyzed for normal distribution in our study to perform the appropriate statistical analyses (e.g., t-test vs. Mann-Whitney test), and these sometimes vary even among the analyses in one panel of a figure. We therefore judged that it would be very confusing not to be consistent and use different representations of error bars, and therefore decided to consistently show the SD.

Minor comments:

Fig. 4d, e: would it make sense to analyze females and males separately?

Yes, we now present these data for all and male (for testosterone), and all and female (for estradiol). We judged this analysis to make most sense. (See Fig 3 d,e.)

Fig. 5d: The authors report reduced sex hormone levels in nose and brain in SA vs. SE and Control groups. Did the authors check systemic (serum) sex hormone levels, too?

Reply: We now report sex hormone concentrations in the sera, which were not significantly different between groups (see Extended Data Fig. 9b).

ED Fig. 1: Please add info on group sizes in the figure legend.

Reply: This is now Extended Data Fig. 3b. We added the group numbers.

ED Fig 3: please specify the unit on the y axis. is it CFU/organ?

Reply: It is CFU/nose (log10). We corrected this.

ED Fig. 5a: it looks like only a fraction of the study subjects provided serum samples. Please mention total numbers in the methods section (line 561).

Reply: This info was now included.

ED Fig 7: OD values from bacterial growth data are usually plotted on log10 scale.

Reply: We now present the data plotted on a log10 scale, as requested. (This is now Extended Data Fig. 10).

ED Table 1: Please provide ethnicity data for the cohorts.

Reply: Because all participants were Han Chinese, we added this information to the methods part rather than the (new and updated) Extended Data Table 1.

Source data: please correct the total number of samples in the source data file for Fig. 2b.

Reply: We updated the source data file.

Article Title: I suggest to rephrase into “Nasal *S. aureus* carriage promotes depression...”

Reply: Another reviewer insisted on including in the title that the human data were associative and the mechanistic data from mice. We did this and now also include the word “carriage”.

Line 57/58: phrase “via alteration of microbial-derived neurotransmitters and sex hormone levels” is unclear. Please specify.

Reply: We corrected to “via alteration of dietary neurotransmitter precursor levels or degradation of sex hormones”.

Line 122: “we colonized the noses of mice” is misleading as the authors are using an unusual colonization model à “we colonized the noses of mice by repeated inoculation with xxx CFU”.

Reply: We appreciate that the reviewer prefers use of the mouse-adapted strain ST88, and we put extensive work in the revised manuscript in which we used that strain and repeated key experiments. However, we do not believe that our colonization model for ST398 is “unusual”. There are many studies that used repeated inoculation to achieve colonization with staphylococci and other bacteria.

We corrected the sentence the reviewer specified as requested.

Line 143/144: “metabolomics sequencing determination” is not the correct term

Reply: Thank you for catching this. We deleted “sequencing”.

Line 148: Authors should check their statement “significant difference” (Fig. 4A). What do p-value and R^2 mean in this type of analysis? To me, the groups are largely overlapping and not different.

Reply: We now consistently use Spearman analysis showing r and the interpretation as described by Akoglu et al. and Chan et al., as directly specified by another reviewer, stating degrees of correlation (poor, fair, etc.) as suggested in those papers.

Line 181 (Fig. 5f, g): The current graph doesn’t show data for CYP17A1 and HSD genes. You should add them. Also increase the size of the highlighted data points.

Reply: We now highlighted CYP17A1 and HSD genes and increased the symbol size for all highlighted genes.

Line 182: Can you replace “affect” by “increase/inhibit/...” to be more specific?

Reply: The wording was changed to be more specific, as suggested.

Line 460: Please add info on the used method, e.g. 16SrRNA gene sequencing.

Reply: This info was now included.

Line 471 (Fig 3 title): rather than “nasal S. aureus” you could write “nasal S. aureus carriage” in the figure legend title.

Reply: This was changed as requested.

Line 495 (Fig. 5 title): “Nasal S. aureus [...] decreases dopamine and serotonin levels in the brain”. This implies a causal, direct link between nasal colonization and dopamine levels, which is not provided in this figure yet. Please rephrase.

Reply: This was changed as requested and the main text heading this figure refers to was changed in the same fashion.

Line 550-553: please specify if the listed exclusion criteria applied for both cohorts or just the control group.

Reply: This was updated to include the requested information.

Line 600ff: specify from which samples you performed WGS. Only the S. aureus isolates?

Reply: This was updated to include the requested information.

Line 608: I understand that MLST typing was performed based on WGS data rather than DNA from individual S. aureus colonies? Please specify.

Reply: Correct. This was updated to include the requested information.

Line 654 and 664: Were the antibiotics and *S. aureus* inoculated under narcosis?

Reply: We added a paragraph “anesthesia” at the end of the “animal experiments” methods section that includes this information.

Line 680ff: The TST method section is hard to understand. I assume you wrap the tail with tape, with the tip sticking out about 1 cm, and then hold the mouse by the tail so that it hangs 15 cm above the ground? Please modify.

Reply: This section was updated in the methods part.

Line 720: specify bacterial culture (log or stat phase, culture medium)

Reply: This section was updated in the methods part, including the requested information.

Line 931: you could add the method here (e.g. “as determined by 16SrRNA gene sequencing”)

Reply: This was added as requested.

Line 940: please specify the day of euthanasia.

Reply: This information was added as requested.

Line 980: specify the culture medium.

Reply: This information was added as requested.

Typos:

Line 32: identify a sex hormone-degrading enzyme in nasal commensal *Staphylococcus aureus*,

...

Line 105: “barely failed to reach“ -> you mean “barely reached”?

Line 110: *S. aureus* colonizes the nose (...) only in about a quarter...

Figure 5e, 6d, 6j: Brian -> Brain

Line 688: water temperature

Line 722 and 724: remove “-“ before μ l

Line 735: write full name for “IS”

Line 505: transcriptomic -> transcriptome

Line 506: “up” -> increased

Line 562: “subpackaged” -> aliquoted

Line 602: with a Bacterial DNA Kit

Line 630: “metabolites level” -> metabolite levels

Reply: Thank you for catching these typos! They were all corrected.

Reviewer #3 (Remarks to the Author):

Xiang, Wang, Ni and colleagues present an interesting manuscript. They investigate a possibly important correlation of the nasal microbiota composition and a (sex-specific) link to depressive disorder - this work might therefore be of high clinical relevance.

The work is outlined in a step-wise manner

1) Comparison of nasal microbiota composition of persons with depression and healthy controls. Depression was "lege artis" diagnosed according to DSM-IV manual by health care professionals and healthy controls were recruited from a physical examination center. PHQ-9 was used as a grading score (dep PHQ-9>15, controls <4). People with "nasal" disease were excluded as were persons under antidepressant treatment or antibiotic use up to 4 weeks before enrollment. Microbiota analysis is done by FL-16S-rRNA gene amplicon seq and culture (24 random colonies per swab culture). A DADA2 pipeline was used for taxonomic assignment. Key findings were (by comparing 55 depression and 102 healthy persons) a higher alpha-diversity in healthy controls.

- 2) Transfer of nasal microbiota into mice (SPF BL/6J) and assessment of behavior using OFT, TST and FST. Randomly allocating microbiota from disease or control groups resulted in significant correlations between OFT, TST and FST and depression (as measured by PHQ-9).
- 3) More specifically there was a link between *S. aureus* colonization and depression. This association was not strain specific.
- 4) Colonization of female mice with *S. aureus* but not *S. epidermidis* led to increased "anxiety and depression" a.k.a scores above - this with similar colonization density and host-tissue response as assessed by cytokine measurement and histology.
- 5) Untargeted metabolomics of a subset of nasal swabs revealed significant differences in nasal metabolomes of depressed vs. healthy persons. Specifically nasal testosterone and estradiol levels were decreased in depressed as compared to healthy controls while serum levels were unaffected.
- 6) The effect was also measured with other bacteria, as well as different *S. aureus* clones and eventually in the in vivo model (*S. aureus* vs. *S. epi*). *S. aureus* colonization led to reduced levels of nasal estradiol and testosterone in the model; interestingly, this effect was also seen in the brain without upregulation of corresponding genes in transcriptomic analysis (midbrain) and as determined by precursor measurements.
- 7) Furthermore, in mouse brains, transcription of genes involved in Dopamine and Serotonin synthesis were downregulated in *S. aureus* carrying mice compared to controls. This was associated with reduced levels of the corresponding transmitters.
- 8) As end conversion of steroid sex hormones is mediated by hydroxysteroid dehydrogenases, the authors screened *S. aureus* genomes for the presence of this gene (or homologues) given that there was a negative correlation between hormone levels and *S. aureus* abundance. Hsd12 was identified and a deletion mutant created, that in female BL/6J mice did not induce a depression

phenotype.

Overall, this is a very interesting manuscript investigating the role of nasal microbiota composition (i.e., *S. aureus* colonization) on BL/6J sex-specific behavior as a proxy for depression. The goal of the authors to strive to provide a mechanistic link instead of just reporting correlation is appreciated.

However, and unfortunately, the work in its current form is unable to provide sufficient evidence for the major claim (i.e., that nasal *S. aureus* promotes depression via sex hormone degradation). Especially the human data analysis is insufficient and prone to bias.

Reply: We thank the reviewer for the positive comments and hope we could satisfactorily address remaining issues, particularly as for the human data.

Major:

- The title is not justified by the results; you only show associations in humans and some changes in BL/6J mice.

Reply: We changed the title to “Nasal *Staphylococcus aureus* carriage is associated with human depression and promotes depressive behavior in mice via sex hormone degradation.” We hope the reviewer approves this new title (it also includes requests made by other reviewers).

- The authors rightly state that depression is a "complex disorder whose pathogenesis is multifactorial and includes an array of underlying genetic, environmental, and endocrine causes." However, they then derive their main hypothesis by performing a Chi-Square test between a convenience sample of depressive patients and healthy controls. It is totally unclear how this population was selected and what other drivers in this complex pathogenesis would and should be considered and these co-variates or confounders were not considered for this analysis. This seems crucial to prevent a biased analysis as for example a reactive depressive disorder might be

something that would not be expected to be driven by *S. aureus* colonization. This obviously also applies for nasal hormone levels detected in healthy vs. depression.

Reply: We implemented strict inclusion and exclusion criteria. Please also note that the cohorts were increased, now both reaching at least 100 individuals, and the number of individuals in the compared groups is now more similar. We re-analyzed all corresponding data. Furthermore, we now also report potential confounders such as serum inflammatory factors, thyroid hormones and sex hormone levels in serum samples and found that there were no significant differences between the two groups.

- Also, curve fits for some of the regression are really questionable (e.g., 2c, 4f, 4g, e2d, e5b) doubting a linear relationship and again omitting any relevant confounders (i.e., 2c).

Reply: We now performed several multivariate correlation analyses and covariate correction (see Fig. 1d, Extended Data Fig. 1b,d,e,g), which all identified *S. aureus* as the only, or most, correlated species with the phenotypes in question. Furthermore, prompted by another reviewer, we applied published interpretation of degrees of correlation (poor, fair, etc.) to the discussion of the correlation analyses in the text.

- It is unclear whether, albeit statistically significant, the differences in sex hormone levels are biologically relevant. This would have to be backed up by literature or experimentally. This especially in the light that a same trend in the same range is seen in blood levels (e5).

Reply: While our statistical analysis revealed significant variations, we acknowledge the need to carefully evaluate their physiological implications. To address this concern, we conducted an expanded serum analysis with increased sample size. This extended investigation demonstrated that serum sex hormone concentrations no longer showed apparent intergroup differences, suggesting the initial observation might have resulted from limited statistical power in the preliminary serum cohort. See Extended Data Fig. 7a.

- The authors do not show, that *S. aureus* decolonization leads to a reversal of the "depressive

phenotype". Given the claim that *S. aureus* decolonization might be a measure to control prevalence of depression (see abstract and discussion), this would be an absolute must.

Reply: This study contains a large set of both human and animal data; and this revision now even includes a very large, additional series of experiments, based on extensive work we performed in the last 6 months. We are proud that we could respond to every single criticism before the resubmission deadline. However, we believe this specific request to be far beyond the scope of the present manuscript. It would require a dedicated clinical trial that would represent a lengthy stand-alone study. We hope the reviewer agrees with us that this cannot be accomplished as part of the present study. We agree, however, that translational claims without the data from such a trial are premature and therefore toned them down in the discussion and deleted them altogether from the abstract in the revised manuscript.

- It would be assumed that the level of *S. aureus* colonization density would then correlate with hormone changes and this depression. This is lacking here. Relative abundance is not a good a measure for this.

Reply: We agree and now measured absolute abundance using qPCR. All corresponding analyses were changed, based on the absolute abundance data.

- 17 β -HSDs catalyze reactions in steroidogenesis and steroid metabolism such as the interconversion of DHEA and androstenediol, estradiol and estrone as well as testosterone and androstenedione. There are different subtypes with slightly different major functions. As there were other bacteria that had quite a high level of estradiol metabolism I wonder whether for example *C. propinquum* would have similar effects.

Reply: Our data show indeed that such activity may be present to a lower degree in other bacteria as well. However, although some nasal bacteria might also have such degrading ability, these bacteria did not show increased abundance in depressed patients. We included a sentence in the discussion presenting this line of thought.

Minor:

- please show a "table 1" with basic demographic information on the study participants

Reply: We extended the Extended Data Table 1 with a large set of additional information, including on confounding variables. We could not include this as a main table due to the maximally allowed number for presentation items. (In our revision, due to the large number of additional data we produced, already exceed the number of allowed Extended Data items and after consultation with the editor, might need to move more data to Supplementary Data.)

- Figure 1: how is *S. aureus* identification done? Is FL-16S really good enough to discriminate *S. aureus* from other Staphylococci?

Reply: Yes. We used amplification of the entire 16S rRNA (not only parts, as usually done). This is an established procedure that allows distinguishing staphylococcal species. We now specified this in the main text and provide citations.

- Figure 2b: relative abundance might be suboptimal as the presence of *S. aureus* itself might be associated with a change in overall bacterial load. Was an overall quantification done? Was there a correlation between random colony picking?

Reply: We now performed absolute quantification via qPCR and recalculated all associated analyses.

- please include information on the controls used in amplicon seq workflows (negative and pos controls, mock communities?)

Reply: We now offer negative control data (Supplementary Data 1). Furthermore, we removed contaminated sequences and reanalyzed the data. Moreover, in the new analyses we corrected for the batch effect.

Reviewer #4 (Remarks to the Author):

In the manuscript entitled, “Nasal Staphylococcus aureus promotes depression via sex hormone degradation”, the authors do a very nice job describing their work. This is an exciting study with novel results that break open an understudied avenue of research in microbial-host interactions. With a few additional experiments and text edits, this work will be an advance in the field.

Reply: We are immensely grateful to the reviewer for their appreciation of our work! We hope we could address remaining issues with the large set of experiments and analyses we performed for this revision.

How many donors were used to generate mice with healthy vs depressed nasal microbiomes?

The methods states that the donors were randomly selected. Does this mean that each datapoint in the behavior experiment panels represents a different mouse each colonized by a distinct donor? Then at this point were the mice single-housed? If so, please clarify details in the text and methods.

-If this is not the case, and only a single or small number of donors were used to colonize multiple mice, this will need to be clearly explained and defended, due to it’s problematic nature experimentally and statistically. Additional donors will need to be used to colonize mice and confirm a broad phenotype across recipient mice to make this data more meaningful.

If the donors were selected randomly, please denote in the data which of the donors in the graphs were ones that had SA in the nasal microbiome? Please answer whether the transfer of a healthy microbiome that has staph aureus sufficient to cause anxiety- and depressive-like phenotypes in mice.

Reply: We appreciate the reviewers concerns and apologize for not having described the experimental details of the transplant experiment better in our initial submission. We now specified the details in the methods (and some in the figure legend). We also highlighted the

transplants that contained *S. aureus* (as per culture analysis) in the figure, as requested. (Fig. 2b)

Furthermore, we supplemented this experiment with a multivariate analysis (Fig. 2c).

As for the question of the reviewer whether a microbiome transplant from a healthy donor containing *S. aureus* can produce depression-like behavior in mice, this is difficult to answer. While there were single data points that may be interpreted this way, we believe one cannot draw meaningful conclusions from single data points and we refrained from speculating more on this question, as our experiment was not set up to answer it. What our data demonstrate is that *S. aureus*-containing transplants confer higher probability of anxiety/depressive-like behaviors and we certainly are aware of the multifactorial basis of depression.

Please add total locomotion/exploratory behavior, eg distance travelled for all OFT test behavior panels (in extended data figures) to serve as a control for health and movement of the animals spending less time in the center of the arena.

Reply: These data are now offered in Supplementary Data 2.

Does intranasal administration of the pure sex hormones decrease OFT, FST and TST phenotypes? Please include this experiment, which will be informative as to the mechanism and location that this relationship along the nose-brain axis is crucial.

Reply: Thank you for this suggestion! We now performed this experiment (using the mouse-adapted strain ST88, which seemed to be preferred by another reviewer). Estradiol or testosterone applied to the noses of mice had significant impacts on OFT, FST and TST phenotypes (see Fig. 6g-j).

We note that to perform this experiment, because a single instillation hardly recapitulates a natural situation and only leads to a momentaneous presence of sex hormones, we had to develop a hydrogel delivery model. This represented a quite laborious endeavor by itself, requiring a series of pilot and control experiments. Only some of those are included in the paper for brevity (Extended Data Fig. 12f,g).

For instance, does the staph enzyme degrade the sex hormone locally in the nose? Is there indication that the HSD12 enzyme is secreted?

Reply: From our new data, it follows that *S. aureus* degrades the sex hormones in the nose.

The HSD12 enzyme is not predicted to be secreted from its amino acid sequence, but it is often reported that cytoplasmic enzymes “moonlight” as secreted enzymes, being released potentially by cell lysis but also other mechanisms. It could also be possible that sex hormones owing to their hydrophobic nature pass through the bacterial membrane or that there are yet unidentified transporters. We find this question interesting and may address it in future work but found this a bit too speculative to include in the discussion.

Line 144 The term “metabolomics sequencing determination of metabolites” is an unusual way to describe metabolomics. Consider rephrasing.

Reply: We apologize; this was a mistake and was corrected.

How many metabolite peaks were detected? How many were identified? It is mentioned that “analyses only included pathways related to steroid biosynthesis and metabolism”, but this is not very meaningful if only a small handful of metabolites were identified and analyzed.

Reply: We reanalyzed those data, including further controls, and now better described the procedure in methods. Through MS/MS-based metabolomic profiling, we detected 1,193 metabolite features at MS2 level, of which 421 were successfully mapped to KEGG database entries. To enhance analytical precision, subsequent pathway analysis focused exclusively on KEGG-annotated metabolites. This refined approach revealed several dysregulated metabolic pathways, with steroid hormone biosynthesis emerging as the most significantly altered pathway.

Consider adding in the text that the correlation in males in panel 4g looks to be driven solely by about 4 healthy individuals.

Reply: Instead of commenting on specific data points, we now – prompted by the suggestion of other reviewers – used a published scale of interpretation of correlation data and always mentioned the degree of correlation (poor, fair, etc.) in the text. We hope this is agreeable.

Response to reviewers

Reviewers Comments:

Reviewer #1 (Remarks to the Author):

Thank you for the revised manuscript. All our questions were well addressed.

Reviewer #2 (Remarks to the Author):

The authors comprehensively addressed all questions raised by the reviewer. The additional animal experiments have resolved my worries that the observed depressive behavior is due to unnoticed *S. aureus* infections (sickness behavior).

Statistics: The authors have now included an accurate description of the error bars (SD) in all figure legends. Correlation data are now interpreted with more caution.

If available, please provide the nasal bacterial load for WT vs. Hsd12 mutant in the colonization model (to exclude that the mutant is per se attenuated).

Reply: We appreciate your comment and would like to note that the result in question has already been included in (what is now) Extended Data Fig. 9f of the manuscript.

Authors might want to swap Fig 3 and 4 so that their order matches their new order of reference in the results section.

Reply: Thank you for your comment. In the current manuscript, Fig. 3 summarizes the global metabolomic profiling analysis, while Fig. 4 provides experimental validation data that directly builds upon the findings from Fig. 3. This sequential presentation (from broad-scale analysis to targeted validation) aligns with the logical flow of the Results section and the corresponding text references. We believe the existing order effectively supports the narrative.

Adjust citation format in lines 903-918.

Reply: We didn't find any anomalies in the citation format. Possibly this was a pdf conversion error.

Reviewer #3 (Remarks to the Author):

The authors have revised the manuscript and extended sample sizes and provided reanalyses and additional experimental data. As initially stated this is a highly interesting manuscript with very solid experimental approaches and high novelty but still has some critical limitations. As also outlined the main problems are discussion of limitations with the human data-set and -analyses as well reversal of phenotype by removal of *S. aureus*.

The title now accurately reflects what was done and shown in the manuscript which is highly appreciated. Nevertheless, some parts still slightly overselling presented findings and for the sake of consistency please also make association and causation (your mouse experiments) clear in the abstract. You don't establish proof that *S. aureus* in humans causes depression.

Reply: We changed the abstract to better explain the sources of evidence of our study: "In this study, comparison of healthy and depressed patient data and mouse studies revealed..."

It is also acknowledged that the authors have expanded their cohort - however, limitations remain and this has to at least be more clearly discussed in the limitations section. Your cohort is not only "rather small" (as stated) but basically very healthy young (except the depression) people from one ethnicity that surprisingly neither anyone smokes or drinks alcohol or has any history of depression in the family. This indicates a highly specific cohort and limits generalizability. Additionally, confounders for depressive disorders are manifold and include:

Demographic & socioeconomic factors such as age, gender, race/ethnicity, income, education, employment status, marital status

clinical & psychiatric comorbidities such as chronic physical illnesses (e.g., diabetes, CVD), pain conditions and other mental disorders (anxiety disorders, PTSD, substance use disorders)

Lifestyle & behavioral factors such as physical activity levels, dietary patterns, sleep habits, smoking, alcohol use, etc

Environmental & social factors such as childhood adversity or trauma, social isolation vs. support, loneliness

Genetic & biological predispositions such as genetic polymorphisms (e.g., BDNF, serotonin transporter variants) or cortisol levels

Methodological issues especially selection bias and confounding by indication in observational studies

While you now indicate some of those in the baseline table, key factors such as education and other (see above) are lacking. And still, your key analysis is based on a Fisher/Chi-Square between depressive and non-depressive patients and a correlation

which (questionably) correlates abundance and PHQ 9 (Fig. 1). The absolute minimum is to extend a critical discussion of missing adjustment for unmeasured confounders that further limits the conclusions of human data. Normally you'd want to present some uni- vs. multivariate regression models comparing crude vs. adjusted ORs. Alternatively, as suggested by other reviewers, a validation cohort would be an option. Please expand the limitations.

Reply: We agree our cohort (ages 18-44, Han Chinese, no comorbidities and no familial psychiatric history) represents a specific subpopulation. However, this homogeneity reduced confounding factors. Strict exclusion criteria minimized interference from aging, chronic diseases, and medications.

To address the concerns, we now conducted structured telephone follow-ups to ascertain key sociodemographic parameters (educational attainment, household income bracket, marital status), psychosocial determinants (family relationship quality, adverse childhood experiences, availability of supportive friends), and sleep quality. However, dietary intake profiles and physical activity at sampling time were irretrievable. Serum cortisol concentrations were quantified through LC-MS/MS analysis in aliquoted specimens cryopreserved at -80°C.

1. Multivariate Logistic Regression:

To test the association of depression status with alpha diversity we used multivariate logistic regression models using depression status as the outcome and alpha diversity as independent variables adjusting for several covariates including sex, age, BMI, education, income and psychosocial and biochemical covariates.

- Model: Group (Depression) ~ Shannon diversity + Sex + Age + BMI + Education (newly collected) + Income (newly collected) + Marital status (newly collected) + Family relationship (newly collected) + Adverse childhood experience (newly collected) + Available supportive friends (newly collected) + Sleep disorders (newly collected) + Serum cortisol (newly collected) + FT3 + FT4 + TSH + IL-1 β + IL-6 + TNF- α

- Result: Shannon diversity still remained significant (adjusted OR=0.165, p=0.017) after adjustment (Extended Data Table 1), indicating nasal microbiota was associated with depression.

All these new results are now included in the manuscript.

2. Sensitivity analysis for unmeasured confounders (using E-value):

The E-value was 4.365 for Shannon diversity -Depression association (adjusted OR=0.165, p=0.017), indicating that an unmeasured confounder would need to increase depression risk and Shannon diversity by >4.365-fold to nullify our result.

3. Expanded limitations:

We tried to follow the reviewer's suggestions as much as we could regarding the extension of the discussion and limitations; however, we also had to shorten the text of our manuscript from about 4200 to less than 3500 words. This was extremely difficult.

We now included the following sentence in the discussion, but could only list what is in parentheses here, in the rebuttal, for that reason.

"Our cohort of young adults without comorbidities cannot represent populations with socioeconomic adversity and adolescent or elderly depression. Although we matched key confounders (*age, BMI, sex, smoking, drinking, comorbidities and medications*), unmeasured factors (*e.g., genetic risk, physical activity levels and dietary patterns*) may contribute to residual confounding. "

Besides, detailed methodological considerations regarding the correlation analysis between microbial abundance and PHQ-9 scores are comprehensively addressed in our response below.

We hope these clarifications addressed your concerns. Thank you for improving our study's rigor.

You cite Kuehner (<https://pubmed.ncbi.nlm.nih.gov/27856392/>) for the claim that the impact of sex hormones on depression is reflected by the fact that depression is more common in women than men. However, this review discusses gender-related depression phenotypes in relation to different exposures where sex hormone alterations is one of them. Please consider rephrasing this sentence to accurately reflect the role of hormones in the pathophysiology (i.e., postpubertal phenotypes (e.g, early-onset has no sex difference)).

Reply: We had to delete this sentence in the introduction for word limit reasons. We now

only state in the discussion:” there are distinct sex-dependent differences in the incidence of depression”. We regret that a more detailed description of the sex-dependent pathophysiology of depression was not possible within those limits.

I was uncertain to find what you added in the multivariate analyses with MaAsLin2 except for age, sex and batch. Can you indicate what was used in which analysis in the legends?

Reply: We appreciate the reviewer's careful evaluation. In response to this comment, we have now explicitly detailed all covariates included in the MaAsLin2 multivariate analyses across relevant figure legends (Fig. 1d, Extended Data Fig. 1e/g) and the former Extended Data Fig. 8, which is now Supplementary Data 2 - we had to move a considerable amount of material to supplementary due to the journal's limit of 10 Extended Data items). Specifically, in addition to the core demographic adjustments for age, sex, and technical batch effects, we now incorporated expanded covariates capturing socioeconomic status (BMI, education level, income), psychosocial factors (family relationship self-evaluation, adverse childhood experiences), and social support metrics (availability of supportive friends). Thank you for helping strengthen the methodological rigor of this work.

Also, you might want to include some of the work that has looked at *S. aureus* and depression: While MRSA isolation in patients has been linked to depressive symptoms (likely due to stigma related factors); the correlation between carriage and depression itself is less strong (<https://pubmed.ncbi.nlm.nih.gov/11740872/>). In mice this work might be of interest: <https://pubs.acs.org/doi/10.1021/acs.est.4c09497> (Airborne Staphylococcus aureus Exposure Induces Depression-like Behaviors in Mice via Abnormal Neural Oscillation and Mitochondrial Dysfunction).

Reply: We appreciate the reviewer's suggestion and thoroughly evaluated whether to include these citations. Given that (i) we had to generally shorten rather than extend the introduction and discussion, where one might have mentioned these studies, and (ii) these two studies are not directly related to our findings, we decided not to include them. As for (ii), the first study focuses on hospitalization rather than *S. aureus* and the second looked at airborne, not colonizing *S. aureus*. We acknowledge that their discussion might have been of some interest, but for text limit reasons it was impossible to include what would have required several sentences.

Experimentally, given your claims based on the well-done set of experiments, there is

still a key experiment missing: does removal of *S. aureus* from mice nasal passage reverse the depressive phenotype. Although you show no effect with ko colonization, a reversal of depressive phenotype by removal of *S. aureus* would be a key experiment for the claims of this work. I've requested this experiment earlier (not in humans) - this would add substantial strength for your mechanistic claims. You can use narrow spectrum antibiotics for this such as vancomycin or mupirocin.

Reply: We sincerely thank the reviewer for the valuable suggestion to include a *S. aureus* colonization group followed by mupirocin decolonization. We fully agree that this design would strengthen the causal link between nasal colonization and behavioral alterations.

However, while this is certainly an interesting experiment to perform in the future, we followed the editor's instructions that no further experimental work needs to be performed for the present study.

Also, I still am not satisfied by some of the correlation analyses (e.g., 1f, 3g) - how is this a "relevant" correlation/relationship between load and score? This should be discussed - see above.

Reply: We appreciate the reviewer's insightful comment regarding the interpretation of correlation analyses between bacterial absolute abundance and depression/anxiety scores (e.g., Fig 1f, 3g). We have taken the following steps to address this concern:

1. Methodological rationale for absolute abundance analysis:

In studies focusing on pathogenic bacteria (e.g., *S. aureus* in our model), absolute abundance directly reflects microbial load, which may drive depression through sex hormone fluctuations via *hsd12*.

This approach aligns with several examples in the established literature (e.g., Cheng L et al. *Cell Host & Microbe* 2024; PMID: 38198925; He X et al. *Nature Medicine* 2024; PMID: 39779925; Liang W et al. *Cell Host & Microbe* 2023; PMID: 37207649; Opron K et al. *NPJ Biofilms Microbiomes* 2021; PMID: 33547327. on pathogen load-behavior correlations and Wang P et al. *Nature Metabolism* 2023; PMID: 37872351 on metabolites-behavior correlations). We copied the respective figures and panels below.

(PMID: 39779925)

(PMID: 39779925)

(PMID: 33547327)

(PMID: 37207649)

(PMID: 37872351)

In these literature sources, correlations exist between microbial abundance or host-microbiota cross-regulated metabolites and disease severity scores, reflecting the role of bacterial load in disease pathogenesis. The absolute abundance metric used in our study provides a more accurate representation of bacterial niche occupancy compared to relative abundance, as also recommended by other reviewers.

2. Further validated biological mechanisms using animal experiments

While the effect size of *S. aureus* from human data is modest, its consistent association with depression scores across genders, combined with mechanistic evidence from animal studies, suggests its potential role as a microbial risk factor.

We hope these clarifications underscore the relevance of our findings. Thank you for improving our study's rigor.

Reviewer #4 (Remarks to the Author):

I appreciate the revised experiments and improvements to the manuscript. The additional experiments support the findings and clarify necessary components. My concerns are resolved.